# Adaptive and migration-enhanced tree seed algorithm for multi-threshold CT image segmentation and lung cancer recognition

**Chenxi Li**[1,2]*, **Jianhua Jiang**[1,2], **Zhixing Ma**[1,2], **Zhilong Yu**[1,2], **Hao Li**[1,2], **Jiayi Liu**[1,2], **Lingna Li**[1,2], **Zhenhao Yu**[1,2]

**1** Center for Artificial Intelligence, Jilin University of Finance and Economics, Changchun, China, **2** Jilin Province Key Laboratory of Fintech, Jilin University of Finance and Economics, Changchun, China

* lichenx0906@163.com

## Abstract

The Tree-Seed Algorithm (TSA) is a swarm intelligence algorithm inspired by the propagation relationship between trees and seeds. However, the original TSA is prone to premature convergence and becomes trapped in local optima when addressing high-dimensional, complex optimization problems, limiting its practical efficacy. To overcome these limitations, this paper proposes an Adaptive and Migration-enhanced Tree Seed Algorithm (AMTSA), which integrates three key mechanisms to significantly enhance performance in solving complex optimization tasks. First, to effectively evade local optima, an adaptive tree migration mechanism is designed to dynamically adjust the search step-size and direction based on individual fitness, thereby improving global exploration. Second, to enhance the algorithm's adaptability and efficiency across different search stages, an adaptive seed generation strategy based on the dynamic Weibull distribution is introduced. This strategy enables flexible control over the number of seeds and promotes a balanced search throughout the solution space. Third, to mitigate convergence oscillations during the global search, a nonlinear step-size adjustment function inspired by the GBO algorithm is incorporated, which effectively improves convergence stability by responding to the iteration progress. Rigorous testing on the IEEE CEC 2014 benchmark functions demonstrates that AMTSA's overall performance surpasses not only state-of-the-art optimizers like JADE and LSHADE but also recent TSA variants, including STSA, fb-TSA, and MTSA. To further validate its robustness in high-dimensional spaces, AMTSA was tested on 30 benchmark functions at 30, 50, and 100 dimensions. Results show that AMTSA ranked first in the number of functions optimized best and exhibited the fastest convergence speed among all compared algorithms. In a real-world application, AMTSA was employed to optimize multi-threshold segmentation for lung cancer CT images. The resulting AMTSA-SVM classification model achieved an accuracy of 89.5%, significantly outperforming models such as standard SVM (76.22%), DE-SVM (82%), GA-SVM (79.33%), TSA-SVM

**Data availability statement:** The source code for the AMTSA algorithm for this study is publicly available (www.jianhuajiang.com). The benchmark functions used for performance evaluation in this study are from the publicly available IEEE CEC 2014 suite. The lung cancer CT images used in the application study were obtained from the public repository, The Cancer Imaging Archive (TCIA) (https://www. cancerimagingarchive.net). All other relevant data for this study are within the paper and its Supporting Information files.

**Funding:** The author(s) received no specific funding for this work.

**Competing interests:** The authors have declared that no competing interests exist.

(84.44%), and JADE-SVM (89.12%). In conclusion, the proposed AMTSA, by integrating adaptive migration, dynamic seed generation, and nonlinear step-size control, successfully addresses the inherent deficiencies of the native TSA, offering a more efficient and robust tool for solving high-dimensional, complex optimization problems. The AMTSA source code will be available at www.jianhuajiang.com.

## 1 Introduction

Optimization problems are crucial in modern scientific research and engineering practice, and their core objective is to find optimal or near-optimal solutions under given constraints [1]. Such problems involve a variety of domains, such as product design, resource allocation, path planning, etc., and cut across a wide range of human activities [2]. Although traditional optimization methods such as Dynamic Programming and Newton's method can provide exact solutions for some problems, they are often difficult to deal with effectively as the complexity of the problem increases, especially when dealing with nonlinearities, dynamic constraints, or noisy disturbances [3,4]. To solve these complex optimization problems, heuristic and meta-heuristic algorithms are gradually becoming more effective choices [5,6].

Heuristic algorithms rely on problem-specific intuitions and strategies [7]. These methods can quickly locate potential solutions by rationally guiding the search process. In contrast, metaheuristic algorithms provide a more general framework [8,9]. They can be widely applied to a variety of complex optimization problems without the need for deep customization. Typical metaheuristic algorithms, such as Genetic Algorithms, Particle Swarm Optimization, and Ant Colony Algorithms, possess global search capabilities and strong parallelism. This enables efficient exploration in complex search spaces [10–13]. However, the No Free Lunch Theorem states that no optimization algorithm can perform optimally in all contexts [14]. Therefore, it is necessary to choose the appropriate algorithm based on the specific nature of the problem [15].

Metaheuristic algorithms can be broadly classified into non-nature-inspired and nature-inspired categories. Non-nature-inspired algorithms, such as Tabu Search(TS) [16], Iterated Local Search(ILS) [17], and Adaptive Dimensional Search(ADS) [18], are widely used for optimization tasks. On the other hand, nature-inspired algorithms, including Particle Swarm Optimization(PSO) [19], Sailfish Optimizer(SFO) [20], Beluga Whale Optimization(BWO) [21], Spider Monkey Optimization(SMO) [22], and Cheetah Optimizer(CO) [23], are increasingly popular due to their simplicity and effectiveness in solving complex optimization problems. However, while some of these algorithms perform exceptionally well for specific types of problems, their effectiveness can vary when applied to different or more intricate scenarios.

Among various metaheuristic algorithms, one notable approach is the Tree Seed Algorithm (TSA) [24]. TSA simulates the natural growth process of trees and seeds, exploring the solution space of optimization problems through the distribution and growth of trees and seeds [25]. Compared to other metaheuristic algorithms, TSA

is recognized for its simple structure and high computational efficiency, particularly when applied to large-scale, complex optimization problems [26]. In TSA, trees and seeds are treated as candidate solutions within the solution space, and the iterative search process is guided by generating seeds from these trees. However, TSA has limitations. It is prone to premature convergence, often getting stuck in local optima, a common issue in many optimization algorithms, especially when tackling complex problems with multiple local extremes [27]. As optimization problems grow more complex, traditional methods increasingly fail to meet practical needs, making heuristic and metaheuristic algorithms more attractive due to their flexibility and adaptability in addressing such challenges [28–30]. This study seeks to address the limitations of TSA by proposing a novel variant that incorporates initial design modifications and hybridization techniques. However, like many early metaheuristics, the native TSA suffers from significant limitations when applied to complex, high-dimensional problems, such as a tendency for premature convergence and an insufficient balance between exploration and exploitation. This research is directly motivated by the need to address these shortcomings. Rather than proposing a new metaphor-based algorithm from scratch, our work focuses on a mechanism-driven enhancement of the TSA framework. We introduce three synergistic, adaptive mechanisms, designed to specifically overcome TSA's inherent deficiencies and significantly boost its performance and robustness.

## 1.1 Motivations

TSA has shown effectiveness in solving optimization problems, but it suffers from several key limitations that hinder its performance. Firstly, TSA tends to converge prematurely, often becoming trapped in local optima due to its limited exploration capacity [31]. Secondly, TSA's reliance on random perturbations for seed generation introduces excessive randomness, making it difficult to control the search process effectively [32]. Lastly, TSA's search process can lack diversity, especially at fixed locations, making it harder to solve complex problems effectively [33]. These limitations highlight the need for improvements to enhance TSA's search capability and adaptability in complex optimization problems. To address these challenges, this study designs an adaptive framework as the core of the proposed method. This adaptive framework operates on three key components of TSA: trees, seeds, and step sizes, thereby enhancing the overall performance and robustness of the algorithm. The motivations driving this research are as follows:

- TSA struggles with balancing exploration and convergence due to its fixed seed generation, which often leads to premature convergence [27]. By redesigning the seed generation process, we introduce a dynamic method to adjust the number of seeds at different stages, addressing this issue and improving the search efficiency and convergence rate.
- TSA struggles with enhancing local exploration and accelerating convergence, particularly during the local exploration phase [34]. By introducing a nonlinear step-size adjustment mechanism and an asymptotic approximation mechanism, we improve TSA's adaptability at different stages, which enhances convergence during the local exploration phase and ensures more efficient and stable search performance.
- TSA often lacks diversity in its search process, particularly at fixed locations, which limits its global exploration capability [35]. By introducing an adaptive migration mechanism, we enhance TSA's ability to explore more broadly, addressing this limitation and improving its efficiency in solving complex problems.

## 1.2 Contribution

While the Tree-Seed Algorithm (TSA) has been the focus of our ongoing research, leading to variants such as ATSA [26] and KATSA [33], the proposed AMTSA represents a significant methodological advancement rather than an incremental improvement. Our previous works, like ATSA, introduced a double-layer framework, and KATSA utilized a k-NN strategy to enhance the search process. However, these prior models still faced challenges in achieving a robust, dynamic balance between global exploration and local exploitation, particularly in high-dimensional and complex problem landscapes. The novelty of AMTSA lies in its unique, three-part synergistic framework designed specifically to overcome these limitations. Unlike our previous approaches, AMTSA introduces:

- **A Dynamic Weibull Distribution for Seed Generation:** This is a fundamental departure from the more rigid or heuristic seed adjustment strategies in our prior work. By dynamically linking the seed count to the population's real-time fitness statistics (mean and standard deviation), AMTSA can adapt its search breadth in a more principled and responsive manner.
- **A GBO-Inspired Nonlinear Step-Size Mechanism:** While previous variants adjusted search parameters, AMTSA is the first in our series to incorporate a sophisticated, nonlinear step-size function inspired by the Gradient-based Optimizer (GBO). This, combined with an asymptotic convergence approach, allows for a much smoother and more effective transition from an aggressive global search to a fine-tuned local search, a feature not fully realized in ATSA or KATSA.
- **A Fitness-Guided Adaptive Migration Strategy:** The migration mechanism in AMTSA is distinct from earlier concepts. It not only provides an escape route from local optima but does so adaptively, where the migration step size and direction are dynamically determined by the individual tree's fitness value. This creates a more intelligent and targeted exploration capability.

In summary, AMTSA is not merely another TSA variant but a comprehensive redesign. Its true innovation lies in the integration of these three adaptive, mutually reinforcing mechanisms, which together create a more robust and efficient optimization tool specifically tailored for the high-dimensional, complex problems addressed in this paper.

In recent years, a growing emphasis has been placed on the need for scientific rigor in the design of metaheuristic algorithms. Scholars such as Sörensen [36] and Camacho-Villalón et al. [37] have published critiques of algorithms that rely heavily on natural metaphors without introducing sufficient mechanical novelty. This body of work argues that the contribution of a truly valuable new algorithm must lie in the introduction of concrete and novel mechanisms that effectively solve optimization problems, rather than simply proposing a new story. We fully concur with this perspective, and it has served as a guiding principle in the design of AMTSA. Although the Tree-Seed Algorithm (TSA) was initially inspired by the relationship between trees and seeds in nature, the core innovation of our proposed AMTSA is not founded on this metaphor. Instead, it is rooted in its concrete, quantifiable mathematical and adaptive mechanisms. The contribution of this work is not the mimicry of a natural process, but rather a framework of three synergistic, carefully engineered components designed to overcome the limitations of existing algorithms, enabling superior performance in solving complex, high-dimensional optimization problems.

## 2 Related work
### 2.1 A brief introduction of tree-seed algorithm

Tree-Seed Algorithm (TSA), proposed by Mustafa Servet Kiran in 2015, is a population intelligence algorithm inspired by the relationship between tree and seed [24]. TSA solves optimization problems by simulating the process of trees reproducing their offspring. The algorithm has been widely used in optimization problems such as function optimization, engineering design optimization, and resource allocation. it has attracted attention for its good global search capability and high convergence speed.The key principles of TSA are outlined below.

- **Step1. Initialize each tree in the population:** In TSA, each tree in the initial population is initialized using Eq (1) to generate a feasible solution as the initial tree $T_{i,j}$.

$$T_{i,j} = L_{j,min} + r_{i,j} \times (H_{j,max} - L_{j,min})$$ (1)

$T_{i,j}$ denotes the position of the ith tree in the jth dimension. $L_{j,min}$ and $H_{j,max}$ are the lower and upper bounds of the jth dimension, respectively, and $r_{i,j}$ is a random number uniformly distributed in the interval (0,1).

- **Step2. Seed number generation mechanism:** Determine the number of seeds to be generated according to Eq (2).

$$ns = fix(low + (high - low) \times rand) + 1 \tag{2}$$

Among them, the fix function is used to round the elements to the integer closest to zero, low indicates the minimum number of seeds generated by each tree, and its value is $N \times 10\%$ ; high indicates the maximum number of seeds generated by each tree, and its value is $N \times 25\%$.

- **Step3. Tree Species Update Mechanism:** The tree update mechanism is the core of TSA, which mainly consists of two different update formulas. The first formula Eq (3) is used for local search and the second Eq (4) is used for global search.

$$S_{i,j} = T_{i,j} + \alpha_{i,j} \times (B_j - T_{r,j}) \tag{3}$$

$$S_{i,j} = T_{i,j} + \alpha_{i,j} \times (T_{r,j} - T_{i,j}) \tag{4}$$

where $S_{i,j}$ is the jth dimension of the seed generated from the ith tree, $T_{i,j}$ is the jth dimension of the ith tree, $B_j$ is the jth dimension of the optimal tree position obtained so far, $T_{r,j}$ is the jth dimension of a tree r randomly selected from the population, and $\alpha_{i,j}$ is a random scaling factor in the range [-1, 1]. Local search focuses on refining the search scope and improving the exploitation of the algorithm, while global search enhances the exploration of the algorithm and prevents falling into local optimal solutions.

- **Step4. Termination condition setting:** In all experiments, the termination condition was determined by the maximum number of function evaluations MaxFEs determined by Eq (5). The updating of the number of function evaluations (FEs) followed Eq (6).

$$MaxFEs = D \times 10000 \tag{5}$$

$$FEs = FEs + ns \tag{6}$$

where $D$ is the dimension of the problem. $ns$ is the number of seeds generated per tree. This termination condition ensures that the algorithm is able to stop at the right time in case of limited computational resources, avoiding ineffective search for too long.

## 2.2 Literature review

The Tree-Seed Algorithm (TSA) has been recognized as a potent optimization tool due to its simple structure and efficiency [38]. However, like many metaheuristics, the native TSA faces challenges in maintaining a robust balance between global exploration and local exploitation, often leading to premature convergence in complex problem landscapes. Consequently, a significant body of research has emerged, focusing on enhancing the original algorithm. These efforts can be broadly categorized into three main streams: refining tree evolution mechanisms, innovating seed generation strategies, and expanding the algorithm's practical applications [39,40].

- *Mechanisms of tree evolution:* A primary research direction for improving TSA has been to enhance the mobility and exploration capability of the 'trees' to prevent the algorithm from settling in local optima. Some studies have drawn inspiration from other successful metaheuristics. The Migration Tree-Seed Algorithm (MTSA), for instance, integrated concepts from the Grey Wolf Optimizer (GWO), employing a gravity-based learning mechanism to guide its migration

strategy, thereby improving the balance between global and local search [41]. Other approaches have focused on internal mechanism design. The Triple Tree-Seed Algorithm (TriTSA), for example, designed two novel migration mechanisms based on triple learning methods and a sine-based random distribution to boost population diversity and adaptability [42]. These works underscore a clear trend: incorporating sophisticated migration strategies is a key avenue for augmenting TSA's global search capabilities.

- *Innovations in seed generation:* Another major focus has been to refine the seed generation process, which is central to the algorithm's search behavior. One group of enhancements involves introducing dynamic adaptation of key parameters. For example, STSA utilizes sine and cosine functions to dynamically adjust balancing parameters [43], while fb-TSA introduces a feedback loop to adaptively tune the search tendency (ST) value and the number of seeds (ns) based on the search progress [32]. A second group of works has focused on incorporating new information to guide seed placement. EST-TSA, for instance, leverages information from the current best solution to enhance its local search [44], and TSASC integrates the Sine Cosine Algorithm (SCA) to refine the seed position updating formula [45]. A third avenue involves hybridizing with other search strategies to increase diversity, such as LTSA employing a Lévy flight random walk [46] and DTSA designing a velocity-driven seed generation mechanism [27].

- *Algorithm applications:* The practical value of these algorithmic enhancements is demonstrated by the successful application of TSA and its variants across diverse and challenging domains. In engineering, CTSA has been tailored to solve constrained optimization problems by effectively combining Deb's rules with the TSA framework [47]. In finance, a hybrid model, sinhTSA-MLP, utilized TSA to optimize a multilayer perceptron, significantly improving the accuracy of credit default risk prediction [48]. The algorithm has also proven effective in medical diagnostics, where a TSA-ANN model was developed for the accurate classification of COVID-19 cases from medical images [49]. The algorithm's versatility is further demonstrated by DTSA's integration of discrete operators to solve complex arrangement coding optimization tasks [27].

Despite these advances in tree evolution and seed generation, a holistic approach that simultaneously addresses adaptive step-sizing, dynamic population management, and robust migration strategies has been underexplored. Many existing methods improve one aspect of the algorithm, sometimes at the expense of another, or rely on fixed parameters that limit their adaptability across different problem types [27]. This paper aims to fill this research gap. We propose an Adaptive and Migration-enhanced Tree Seed Algorithm (AMTSA) that integrates three synergistic mechanisms to create a more balanced, robust, and efficient optimizer for complex, high-dimensional problems.

To ensure a comprehensive comparison, we acknowledge the emergence of other high-performance optimization algorithms in the computational intelligence field in recent years. Notable examples include ICSPM and ICSPM2 [50], Exploratory Cuckoo Search [51], and the Improved Salp Swarm Algorithm with HDPM (ISSA) [52]. While a direct experimental comparison with these methods was beyond the scope of the current study, benchmarking AMTSA against these promising approaches remains a valuable direction for future work.

### 2.3 An overview of GBO

GBO is a meta-heuristic optimization algorithm based on gradientizer [53]. Proposed by Iman Ahmadianfar, Omid Bozorg-Haddad, and Xuefeng Chu in 2020, GBO is inspired by gradient-based Newtonian methods and uses two main operators [54], Gradient Search Rule (GSR) and Local Escape Operator (LEO), and a set of vectors to explore the search space. The working principle of GBO can be briefly summarized the working principle of GBO can be briefly summarized as follows.

**Step1: In the initialization phase:** the GBO first generates an initial population with each individual (i.e., vector) randomly distributed in the search space. The vector representation of the population size $N$ and dimension $D$ is Eq (7), and the initial vector is generated by Eq (8):

$$X_n = [X_{n,1}, X_{n,2}, ..., X_{n,D}], n = 1, 2, ..., N \tag{7}$$

$$X_n = X_{min} + rand(0, 1) \times (X_{max} - X_{min}) \tag{8}$$

Where $X_{min}$ and $X_{max}$, are the lower and upper bounds of the decision variable, respectively, and rand(0,1) is a random number between [0, 1].

**Step2: Gradient Search Rule (GSR):** Using the gradient method in Eq (9), each individual is guided toward an improved solution.

$$GSR = randn \times \rho_1 \times \frac{2\Delta x \times x_n}{(x_{worst} - x_{best} + \varepsilon)} \tag{9}$$

where $x_{best}$ and $x_{worst}$ are the best and worst solutions obtained during the optimization process. $\varepsilon$ is a small value used to avoid divide-by-zero errors. $\Delta x$ is a small step size used in the numerical gradient computation. To improve the search capability of GBO and balance global exploration with local exploitation, the GSR is modified by introducing a stochastic parameter $\rho_1$ in Eq (10). In this study, $\rho_1$ is the key adaptive coefficient for balancing exploration and exploitation, and it is expressed as:

$$\rho_1 = 2 \times rand \times \alpha - \alpha \tag{10}$$

$$\alpha = \left| \beta \times \sin\left(\frac{3\pi}{2} + \sin\left(\beta \times \frac{3\pi}{2}\right)\right) \right| \tag{11}$$

$$\beta = \beta_{min} + (\beta_{max} - \beta_{min}) \times \left(1 - \left(\frac{m}{M}\right)^3\right)^2 \tag{12}$$

where $\beta_{max}$ and $\beta_{min}$ are used to regulate the minimum and maximum values of the parameter $\beta$ range. These parameters are used to compute the adaptive parameter $\rho_1$, which balances global exploration and local exploitation during the iterations of the algorithm. The integration of the adaptive coefficient mechanism from the GBO algorithm into TSA requires new balancing mechanisms to regulate the global and local search phases. Therefore, Sect 3.2 details the integration of the above enhancements into the TSA

**Step3. Integrated position update:** The position of the current vector is updated using using GSR and Directional Movement (DM).DM is given by Eq (13) and the integrated position update formula is as Eq (14).

$$DM = rand \times \rho_2 \times (x_{best} - x_n) \tag{13}$$

$$x_{n+1} = x_n - GSR + DM \tag{14}$$

## 3 Methods

In this section, we present the three key innovations in the algorithm, organized according to the logical sequence of improvements. First, we propose an adaptive method for adjusting the number of seeds at different stages of the process. Second, we introduce an adaptive, nonlinear approach to optimize the seed generation locations. Finally, we design an adaptive migration mechanism to help the algorithm escape local optima and improve its overall performance.

### 3.1 A dynamic seed number strategy based on the Weibull distribution

The original Tree seed Algorithm(TSA) uses a completely randomized number of seeds generation strategy where the number of seeds depends on the population size. This mechanism generates the number of seeds with great randomness and lacks a dynamic and adaptive strategy, which may result in the algorithm failing to maintain good performance in different dimensions and search phases. To improve these drawbacks, we introduce an adaptive seed generation strategy with dynamic Weibull distribution, which makes the generation of the number of seeds more flexible and adaptive to the current search situation by adaptively adjusting the shape and scale parameters of the Weibull distribution to improve the search efficiency and adaptability of the algorithm. We set the scale parameter *lambda* and shape parameter *k* via Eq (15) and Eq (16), and let ns change adaptively via Eq (17).

$$lambda = (dmax - dmin) \times (1 + \frac{std\_obj}{mean\_obj}) \tag{15}$$

$$k = 1 + \sqrt{\frac{D}{10}} \times \left(1 + \frac{mean\_obj - best\_obj}{mean\_obj}\right) \tag{16}$$

$$ns = \lceil \text{wblrnd}(lambda, k) \rceil \tag{17}$$

where *dmax*, *dmin* are the upper and lower bounds of the search space, *best_obj* is the value of the objective function corresponding to the minimum value of the fitness, *D* is the dimension of the problem, *std_obj* is the standard deviation of the fitness value, *mean_obj* is the mean of the fitness value, and $\lceil \cdot \rceil$ is the upward round.

The scale parameter (*lambda*) directly influences the seed distribution range, with its value adjusted based on the relative volatility of the population's fitness, represented by $\frac{std\_obj}{mean\_obj}$. A larger value of this ratio indicates higher fitness diversity and thus a larger scale parameter, resulting in more seeds for broader global exploration. Conversely, a smaller ratio leads to fewer seeds and faster convergence with more localized searches. Meanwhile, the shape parameter (*k*) is dynamically adjusted based on the problem's dimensionality (*D*) and the difference between the population's average fitness and the current optimal fitness (*mean_obj*-best). In higher-dimensional problems, a larger *k* expands the seed count, promoting diversity and global exploration. If the fitness difference is large, *k* increases to improve global search; if small, it decreases to focus on local search and accelerate convergence. Together, these parameters ensure a balance between global exploration and local exploitation, enhancing TSA's adaptability. The idea is shown in Fig 1.

### 3.2 Adaptive step size with nonlinear strategy

In TSA, the step size $\alpha_{i,j}$ in the seed generation equation is a uniform random number between (-1,1), which controls the search range of the seed in the neighborhood of the tree position. When *rand* < *ST*, the Eq (3) of the seed generated based on the current tree is influenced by the best position, and the step size moderates the strength of this influence. However, the values of $\alpha_{i,j}$ exhibit large randomness and significant jumps during the search process. This can result in insufficient fine-tuning of the search. As a result, the excavation may not adaptively adjust as needed during the search.

To solve this problem, Inspired by the GB-based improved GSR search rule in the GBO algorithm, 2 perturbation factors beta and alpha are introduced to adjust the step size with a nonlinear function to enhance the search process to be more flexible. The generated solution should be able to explore the search space around its corresponding best solution. Thus, when the number of iterations reaches a later stage, the parameter values increase. This increase helps the algorithm escape from local optima. It does so by boosting the diversity of the population, which enhances the search around the best solution obtained. Then, using the method of progressive convergence (asymptotic convergence), the two perturbation factors are dynamically adjusted. This adjustment enables a smooth transition from global search to local search.

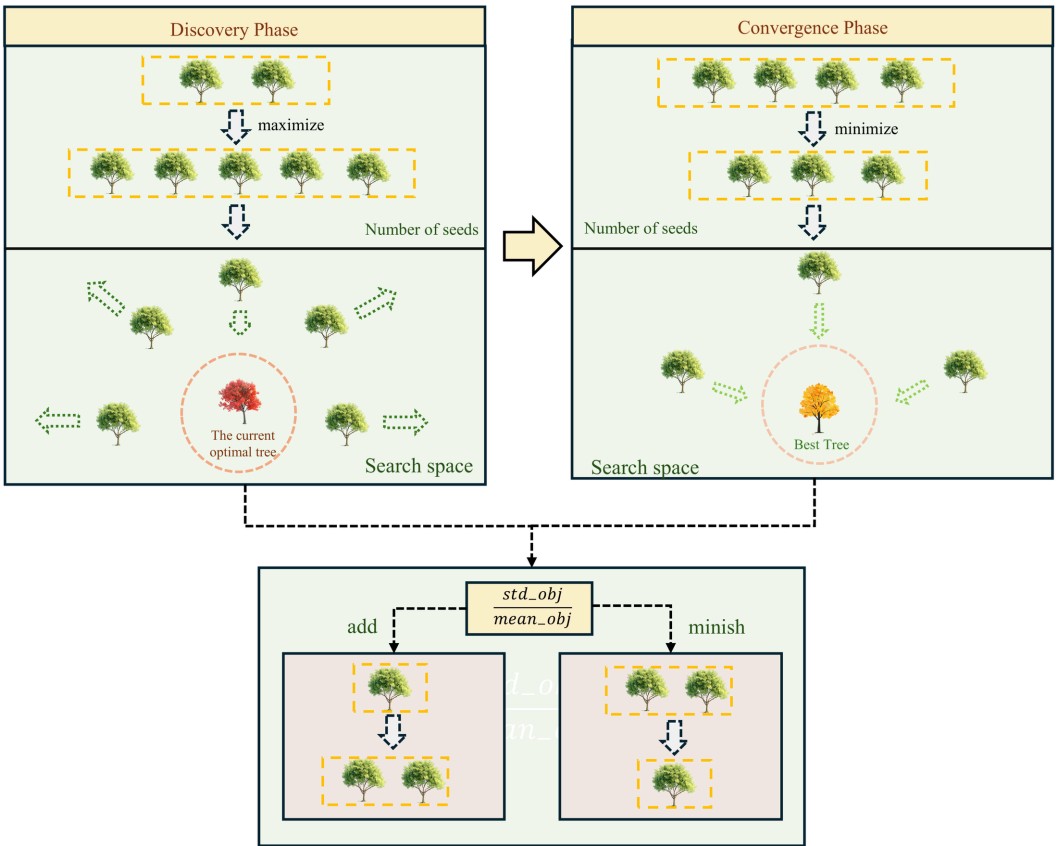

**Fig 1**. **A dynamic seed number strategy based on the Weibull distribution.**

The overall performance of the optimization algorithm is further improved by gradually reducing the step size. We give the perturbation factors beta and alpha by Eq (18) and Eq (19), and the dynamic step size by Eq (20), and the generating seed formula by Eq (21).

$$beta = 0.2 + (1.2 - 0.2) \times (1 - (\frac{t}{Max\_Gen})^3)^2 \tag{18}$$

$$alpha = |beta \times \sin\left(3 \times \frac{pi}{2} + \sin\left(\frac{3 \times pi}{2} \times beta\right)\right)| \tag{19}$$

$$step\ length = \frac{alpha}{t^{beta}} \tag{20}$$

$$seeds(j, d) = \begin{cases} trees(i, d) + (best\_arams(d) - trees(r, d)) \cdot step\_length & \text{if } rand < ST \\ trees(i, d) + (trees(i, d) - trees(r, d)) \cdot (rand - 0.5) \cdot 2 & \text{if } rand > ST \end{cases} \tag{21}$$

where $t$ is the current iteration number and Max_Gen is the maximum iteration number.

 

*Beta* ranges from 1.2 to 0.2, smoothly transitioning from exploration to exploitation. Larger *beta* values in early iterations enhance exploration, while smaller values in later stages focus on exploitation. The *alpha* nested sine function introduces oscillation, promoting diverse search paths and avoiding local optima. The idea is shown in Fig 2.

### 3.3 Adaptive migration tree strategy

The mechanism for TSA to jump out of the local optimum is necessary. Due to the incompleteness of the mechanism, seeds in TSA remain locked around their parent tree. The particle optimum that falls locally is difficult to escape from its current position. In order to solve the stagnation problem of the local optimum, the introduction of a migration mechanism is necessary. To solve this problem, we design a migration mechanism. This mechanism adaptively determines the step size and direction. It provides an opportunity for the tree seeds that have fallen into a local optimum to escape from their current position.

This migration mechanism is generalized in nature, all parent trees are compared for migration to decide whether to migrate to that position or not. The step size is determined by a dynamic step factor with an adaptive function, the direction is determined by a random direction vector with an adaptive function, and the final migration formula consists of the step size, direction, and Cauchy mutation strategy. In order to achieve the problem of jumping out of the local optimum. Eq (22) is given the adaptive function which dynamically adjusts the individual's step size according to the fitness. Individuals with lower fitness have larger step sizes to encourage more exploration, while individuals with higher fitness have smaller step sizes and move more cautiously.

$$\text{fitness} = \frac{1}{1 + obj} \tag{22}$$

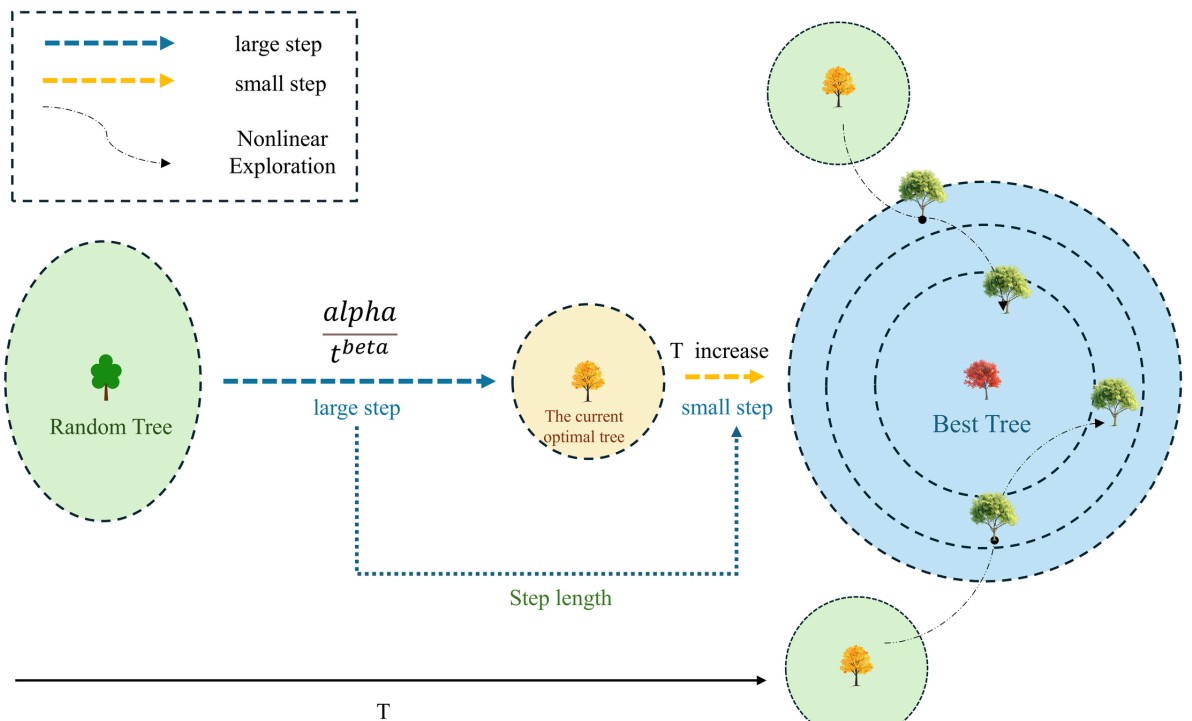

**Fig 2**. Adaptive step size with nonlinear strategy.

The adaptive function is the basis for determining the step size and direction, but in order to ensure the limitation of the step size, a nonlinearly varying dynamic change factor generated from the sine function is added. Eq (23) gives the factor

$$m = 2 * \sin(r + pi/2) \tag{23}$$

where $r$ is a uniformly distributed random number taking values between [0, 1].

When traversing each parent tree, the step final value (Eq (24)) and the effect of direction (Eq (25)) are determined by the fitness of that parent tree compared with the fitness of the randomly matched individuals. And the parent tree after the update is completed is generated by Eq (26).

$$step\_lengths = m \cdot fitness \tag{24}$$

$$step\_length = \begin{cases} step\_lengths(i) & \text{if } obj(i) < fitness(target\_index) \\ step\_lengths(target\_index) & \text{if } obj(i) > fitness(target\_index) \end{cases} \tag{25}$$

$$direction = direction \cdot fitness(i) \tag{26}$$

$$newtrees(i,:) = trees(i,:) + step\_length \cdot cauchy\_value \cdot direction \tag{27}$$

where $target\_index$ is the randomly selected target parent tree and newtrees(i,:) is the position of the parent tree after migration. The idea is shown in Fig 3.

### 3.4 AMTSA: A novel tree seed algorithm

The algorithm is improved to address the original defects of the original Trees seed Algorithm (TSA). An adaptive seed generation strategy with dynamic Weibull distribution is introduced, which adaptively adjusts the shape and scale parameters of the Weibull distribution and determines the number of seeds to be generated through different dimensions and search stages.It significantly improves the global search capability during the exploration phase. Additionally, it enhances the fast response and optimization search capability during the convergence phase. Inspired by the GSR search rule in the GBO algorithm, the original step size is nonlinearly improved to realize the smooth transition from global search to local search, and to improve the mining accuracy and adaptive ability of the optimization algorithm. Finally, an adaptive migration mechanism is designed to generate a migration strategy by integrating global search and adaptive directional step size, giving each tree will jump out of the local optimum and mine better points. These strategies these methods together promote a more dynamic, adaptive and efficient optimization algorithm that optimizes the original shortcomings of the TSA algorithm. The flowchart of the algorithm is shown in Fig 4.

### 3.5 Time complexity analysis of AMTSA

To evaluate the efficiency of the AMTSA algorithm, it is essential to analyze its time complexity, which reflects how the algorithm's computational cost grows with respect to the problem size. The AMTSA algorithm consists of several major stages, including initialization, iterative updates, seed production, and migration mechanisms. In this section, we provide a detailed analysis of the time complexity involved in each phase of the algorithm.

In the initialization phase, the algorithm generates random positions for the trees within a defined search space. Since the tree population consists of $N$ trees, each having $D$ dimensions, the time complexity for this operation is $O(N \times D)$.

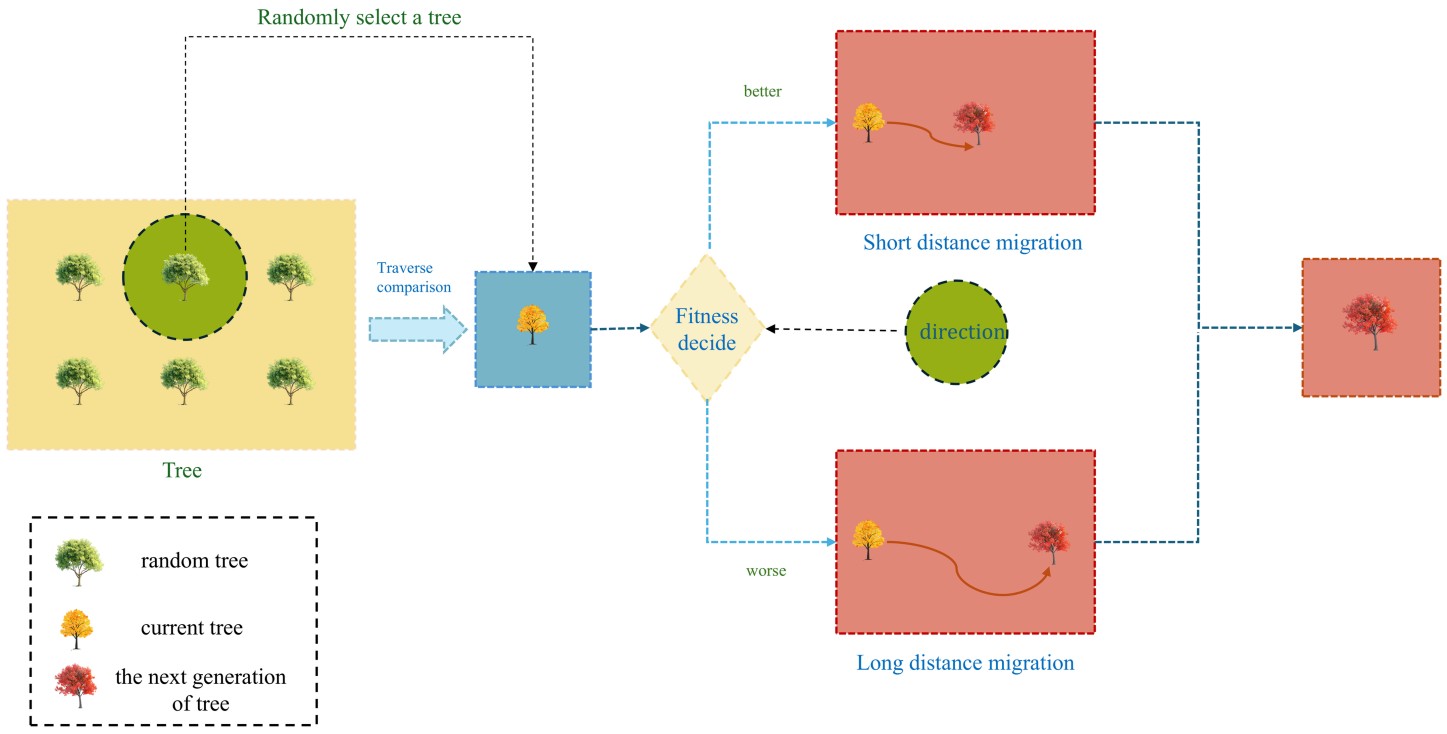

**Fig 3**. Adaptive migration tree strategy.

Additionally, the objective function value for each tree is computed, which also requires $O(N \times D)$ time, as each evaluation involves processing a $D$-dimensional vector. Therefore, the total time complexity of the initialization phase is $O(N \times D)$.

The iterative update phase, which is the core of AMTSA, involves several key steps within each generation. This phase iterates for a maximum of *Max_Gen* generations, and within each iteration, multiple operations are performed, including adaptive weight calculations, seed generation, fitness evaluations, and updates to tree positions. The adaptive weight calculation is computationally light, with a time complexity of $O(1)$, as it involves basic arithmetic operations. Seed production, on the other hand, is more computationally demanding. Each tree produces a variable number of seeds, *ns*, based on a Weibull distribution. The number of seeds generated for each tree is bounded, and the average number of seeds is approximately $O(1)$. For each seed, the fitness function is evaluated, which takes $O(D)$ time. Thus, the total complexity for seed production and fitness evaluation across all trees is $O(N \times D)$.

In the migration phase, which is designed to help the algorithm escape local optima, each tree adjusts its position dynamically based on its fitness and the positions of other trees. This phase includes random number generation, position updates, and fitness comparisons. Since these operations are performed for each of the $N$ trees and each involves operations that are linear with respect to the problem dimension $D$, the time complexity for the migration phase is $O(N \times D)$.

Finally, the overall complexity of the algorithm is dominated by the iterative update phase. As the algorithm performs *Max_Gen* iterations, each requiring $O(N \times D)$ time, the total time complexity of AMTSA is $O(Max\_Gen \times N \times D)$. This means that the computational cost of the algorithm increases linearly with both the number of trees $N$ and the problem dimension $D$, as well as the number of generations *Max_Gen*.

In summary, the time complexity of AMTSA is primarily driven by the iterative update phase, which is $O(Max\_Gen \times N \times D)$. This reflects the typical behavior of evolutionary algorithms, where the time complexity grows with the size of the population and the dimensionality of the search space, as well as the number of iterations required for convergence.

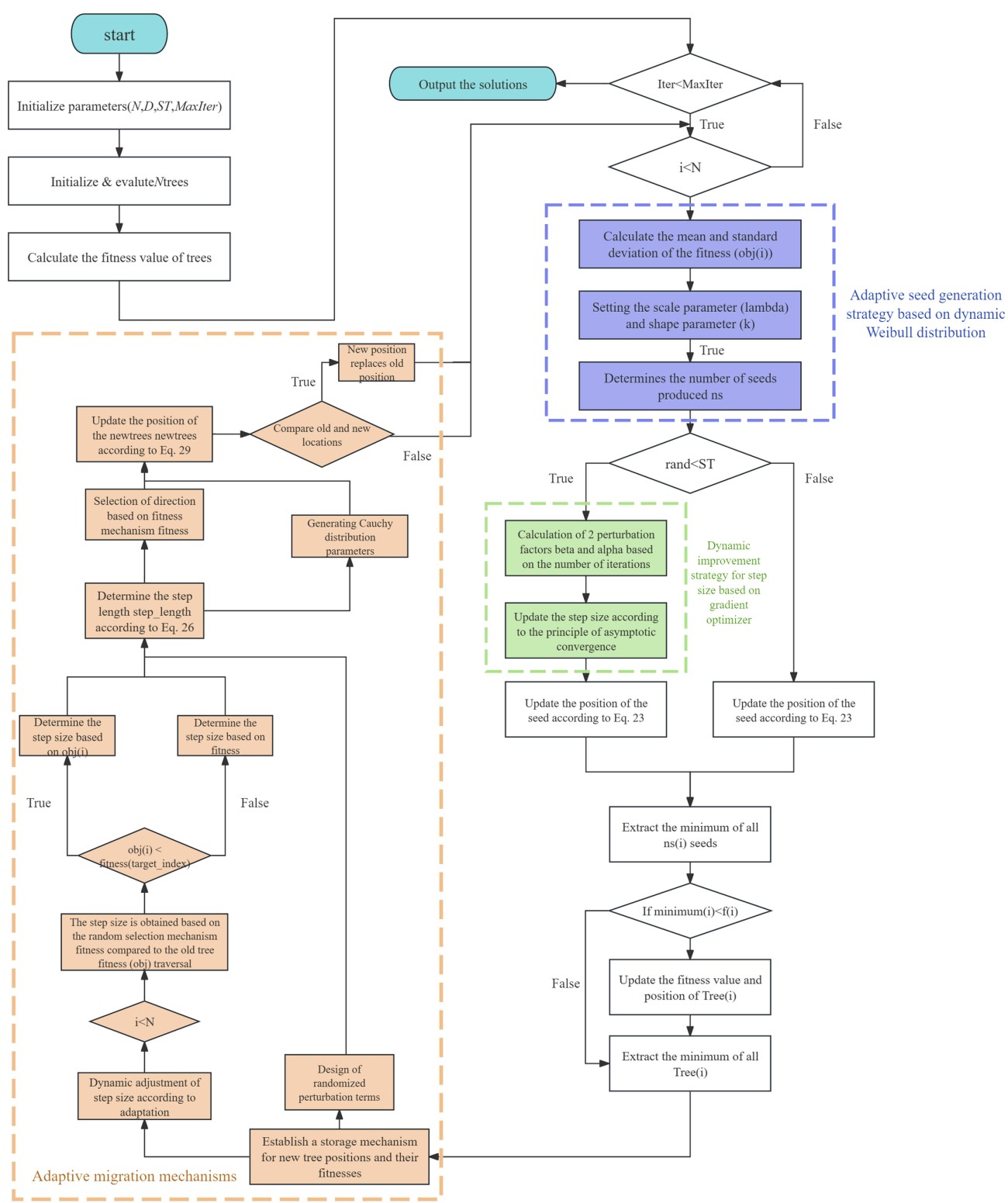

**Fig 4**. The flowchart for the proposed AMTSA.

**Algorithm 1 The pseudo-code of the AMTSA.**

1: **Step 1. Initialize the population**
2: Set the initial number of zones and problem dimensions;
3: Set the *ST* parameter and termination criteria;
4: Evaluate each tree's fitness against the target function;
5: <u>**Creation1:**</u>
6: Generate initial positions (and any auxiliary data) via Sine chaotic mapping;
7: <u>**End Creation1**</u>
8: **Step 2. Seed-based search**
9: **for** each tree **do**
10:    Compute $\beta, \alpha$ based on iterations;
11:    Compute current fitness mean and standard deviation;
12:    Derive Weibull parameters (shape $k$, scale $\lambda$) from fitness data;
13:    Determine number of seeds for this tree via Weibull distribution;
14:    **for** each seed **do**
15:       **for** each dimension **do**
16:          **if** *rand* $<$ ST **then**
17:             <u>**Creation2:**</u>
18:             Compute dynamic step size (Eq 20);
19:             Generate new tree position (Eq 21);
20:          **else**
21:             Generate new tree position (Eq 21);
22:             <u>**End Creation2**</u>
23:          **end if**
24:       **end for**
25:    **end for**
26:    Select the best seed; if it outperforms the tree, replace the tree with that seed;
27: **end for**
28: **Step 3. Migratory perturbation**
29: Design perturbation term (Eq 23);
30: **for** each tree **do**
31:    Randomly pick a target individual;
32:    **if** some condition **then**
33:       Compute step size (Eq 25);
34:    **else**
35:       Compute step size (Eq 25);
36:    **end if**
37:    Determine direction via Normal-distributed random vector weighted by fitness;
38:    Update tree position (Eq 27);
39: **end for**
40: Compare new vs. old positions; keep the better one;
41: **Step 4. Termination check**
42: If stopping criteria not met, go to Step 2;
43: **Step 5. Report**
44: Output the best-found solution.

## 4 Experimental environment and setting

### 4.1 Experimental fundamentals

All experiments were implemented in the MATLAB 2016a programming environment and executed on a computer with an Intel Core i7-11800H processor, running a 64-bit Windows 10 operating system. To ensure a rigorous and fair comparison, all experiments were designed to strictly adhere to the official guidelines of the IEEE CEC 2014 benchmark competition. The 30 benchmark functions provided by the competition were evaluated at 30, 50, and 100 dimensions. For statistical

significance, the final results reported for each function are the average of 30 independent runs. Following the competition's standard protocol, the stopping criterion for all algorithms was uniformly set to a maximum number of fitness evaluations (MaxFEs), defined as D×10,000, where D is the problem dimension. This ensures that every algorithm is allocated an identical computational budget proportional to the problem's difficulty. Furthermore, to address the critical issue of fair parameterization, the settings for each comparative algorithm, including their population size strategies, were configured based on the recommendations from their respective original publications or the official CEC 2014 report. This avoids the bias of using a single, fixed setting for all algorithms.

A comprehensive list detailing the specific parameter settings for every algorithm is provided in Table 1. This standardized protocol guarantees that all algorithms are compared on a level playing field, making the performance evaluation both reproducible and credible.

### 4.2 Comparative algorithms

To comprehensively evaluate the performance of the proposed AMTSA, it was benchmarked against a large and diverse suite of state-of-the-art and classic optimization algorithms. The comparison suite was carefully selected to assess AMTSA's performance in multiple contexts. The algorithms are grouped into three categories:

**High-Performance and CEC Benchmark Algorithms.** To validate AMTSA against the highest standards, we included several algorithms known for their top-tier performance in rigorous academic competitions, particularly the IEEE CEC 2014 benchmark. This group includes LSHADE [55], JADE [56], as well as other CEC competition winners and highly regarded methods such as GaAPPADE [57], MVMO-SH [58], and the Covariance Matrix Adaptation Evolution Strategy (CMA-ES) [59].

**Established and Novel Heuristic Algorithms.** To position AMTSA in the broader context of metaheuristics, it was also compared against other respected optimizers. This set includes Differential Evolution (DE) [60,61], Genetic Algorithm (GA) [62], and Bat Algorithm (BA) [63].

**TSA Variants.** To demonstrate the improvements over the original algorithm and provide a comprehensive comparison within its own family, AMTSA is compared with a wide range of TSA variants. This includes the original Tree-Seed Algorithm (TSA) [24], recent notable versions such as EST-TSA [44], fb-TSA [32], STSA [43], and MTSA [64]. The comparison is also extended to our prior works, ATSA [26] and KATSA [33].

### 4.3 Evaluation metrics

To analyze the algorithm's behavior, this study assesses population diversity, the balance between exploration and exploitation, and convergence performance.

**Population Diversity Evaluation.** The assessment of population diversity is a key metric for evaluating efficiency, as it directly affects the depth of exploration in the search space [12]. This paper represents population diversity using the dispersion between individuals and the centroid of the group, calculated using Eqs (28) and (29).

$$I_c(t) = \sqrt{\sum_{i=1}^{N} \sum_{d=1}^{D} (x_{id}(t) - c_d)^2(t)} \tag{28}$$

$$c_d(t) = \frac{1}{N} \sum_{d=1}^{D} (x_{id}(t)) \tag{29}$$

**Exploration and Exploitation Analysis.** The balance between exploration and exploitation is a cornerstone of performance in metaheuristic algorithms. Exploration refers to the algorithm's ability to search broadly across the entire solution space to discover globally promising regions, while exploitation involves refining the search in the vicinity of known

**Table 1. The initial parameters of comparative algorithms.**

| Algorithm | Parameter | Value | Population Size (N) | MaxFEs |
|---|---|---|---|---|
| AMTSA | ST | 0.1 | 30 | $D \times 10,000$ |
| EST-TSA(2019) | ST | 0.1 | 30 | $D \times 10,000$ |
| fb-TSA(2020) | ST | 0.1 | 30 | $D \times 10,000$ |
| TSA(2015) | ST | 0.1 | 30 | $D \times 10,000$ |
| STSA(2020) | ST | 0.1 | 30 | $D \times 10,000$ |
| MTSA(2022) | ST | 0.1 | 30 | $D \times 10,000$ |
| GA(2004) | Type<br>Selection<br>Crossover<br>Mutation | Real coded<br>Roulette wheel<br>Probability=0.7<br>Probability=0.2 | 100 | $D \times 10,000$ |
| BA(2005) | Loudness(A)<br>Pulse rate (a)<br>Frequency minimum<br>Frequency maximum | 0.5<br>0.5<br>0<br>2 | 100 | $D \times 10,000$ |
| LSHADE(2014) | Historical memory size (H)<br>p value<br>$r^{N^{init}}$<br>$N^{min}$ | 6<br>0.11<br>2.6<br>4 | linearly decreasing from $N^{init} = D \times r^{N^{init}}$ to 4 | $D \times 10,000$ |
| JADE(2009) | c<br>UCR<br>UF<br>top | 0.1<br>0.5<br>0.5<br>1 | 30 (if $D \leq 10$)<br>100 (if $D = 30$)<br>400 (if $D = 100$) | $D \times 10,000$ |
| KATSA(2025) | neighbor tree number ($k$) | 2 | 30 | $D \times 10,000$ |
| ATSA(2023) | $k$<br>$\beta$<br>the threshold for the elimination count | decreases linearly from 2 to 0<br>range from [-2,2]<br>3 | 30 | $D \times 10,000$ |
| GaAPPADE(2014) | F<br>CR<br>mean vector (m)<br>covariance matrix (C)<br>F range<br>CR range | adapted by GaA<br>adapted by GaA<br>initialized to [0.5, 0.5]<br>initialized to identity matrix<br>(0,1.0]<br>[0,1.0] | 100 | $D \times 10,000$ |
| MVMO-SH(2010) | dynamic population size (n)<br>mutated variables (m)<br>shaping scaling factor ($f_s$)<br>asymmetry factor (AF)<br>shape factor ($s_d$) | 2<br>1<br>1.0<br>1.0<br>75 | 2 | $D \times 10,000$ |
| CMA-ES(2003) | offspring ($\lambda$)<br>parents ($\mu$)<br>initial global step size ($\sigma^{(0)}$)<br>initial covariance matrix ($C^{(0)}$)<br>$c_c$<br>$c_\sigma$<br>$c_{cov}$<br>$d_\sigma$ | variable, e.g., $4\mu$<br>variable, e.g., $\lambda/4$<br>1 or 0.1<br>identity matrix (I)<br>$\frac{4}{n+4}$<br>$\frac{4}{n+4}$<br>$\frac{2}{(n+2)^2}$<br>$c_\sigma^{-1} + 1$ | variable | $D \times 10,000$ |

good solutions to find the local optimum precisely. A successful algorithm must transition smoothly from exploration to exploitation during its run. To quantitatively measure these two behaviors, this study adopts a widely recognized method based on population diversity. The diversity, which reflects the spread of individuals in the population, is calculated using the average distance of individuals from their population centroid. The diversity at iteration t, denoted as $Div_t$, is calculated using Eqs (30) and (31). A high diversity value indicates that the population is widely dispersed, signifying a state of exploration. Conversely, a low diversity value suggests that the population has converged around promising areas, which is

characteristic of exploitation. Therefore, we define the percentage of exploration and exploitation at iteration t as follows:.

$$Exploration(\%) = \frac{Div_t}{Div_{max}} \qquad (30)$$

$$Exploitation(\%) = \frac{|Div_t - Div_{max}|}{Div_{max}} \qquad (31)$$

where $Div_t$ is the population diversity at the current iteration t, and $Div_{max}$ is the maximum diversity recorded throughout the entire optimization process. This metric allows for a dynamic assessment of the algorithm's behavior, providing clear insights into how it balances the search process over time.

**Convergence Curve.** The convergence curve is used to visually assess an algorithm's efficiency and convergence speed. It plots the best fitness value found so far against the number of iterations. A steeper curve indicates faster convergence toward the optimal solution.

## 5 Analysis ans discussion

This section presents a comprehensive performance evaluation of the proposed AMTSA algorithm, examining its capabilities through a multi-faceted series of experiments. The evaluation begins in Sect 5.1 with a qualitative analysis of AMTSA's convergence behavior and search dynamics. Sect 5.2 then provides an in-depth quantitative benchmark against a wide range of state-of-the-art and classic optimizers. To further dissect the algorithm's properties, subsequent sections investigate its sensitivity to key parameters (Sect 5.3), validate the contribution of each novel component through an ablation study (Sect 5.4), and assess its practical computational efficiency and scalability (Sect 5.5). The chapter concludes with rigorous statistical tests in Sect 5.6 to confirm the significance of the observed results.

### 5.1 Qualitative analysis

Qualitative analysis of the algorithm is one of the criteria to detect the performance of the algorithm, we conducted convergence behavior analysis, population diversity analysis and exploration mining analysis experiments for this algorithm to observe the performance of the algorithm. Among them, we chose the single-peak function F1 to evaluate the exploitation ability of the algorithm and the multi-peak function F8 to evaluate the exploration ability of the algorithm, as shown in Fig 5 below.

   **5.1.1 Convergence behavior analysis.**  In the case where the algorithm was subjected to a convergence behavior analysis, a total of four tests were performed with the following specific features.

- Figs 6(a) and 5.1.1(a) show the optimization process of the AMTSA algorithm. In these figures, the black dots represent the area covered by the current seed, while the red dots indicate the best position found, i.e., the optimal solution. The clustering phenomenon of black dots around the red dots indicates that AMTSA is moving towards convergence during the gradual optimization process.
- The Figs 6(b) and 5.1.1(b) demonstrate the convergence of AMTSA, clearly reflecting its ability to rapidly approach the optimal solution. The significant decreases in the convergence curves not only highlight the algorithm's efficiency in the optimization search process, but also demonstrate its keen ability to explore the problem space. This fast convergence property gives AMTSA a significant advantage in complex optimization tasks.
- The Figs 6(c) and 7(c) monitor the change in the first dimension, providing an important insight into the behavior of the algorithm and effectively preventing premature convergence to a local optimum. Experimental evidence shows that the AMTSA algorithm is able to efficiently steer the search process to avoid falling into a local optimum, thus ensuring that it explores a wider solution space. This ability significantly improves its performance in complex optimization problems.

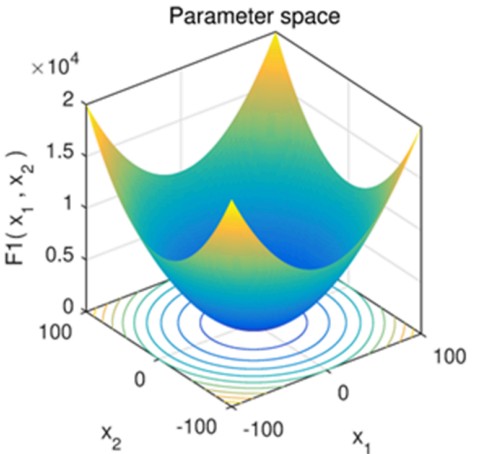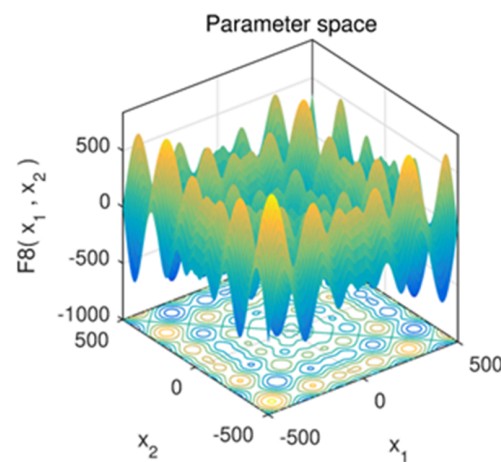

**Fig 5**. **Unimodal and Multimodal functions.**

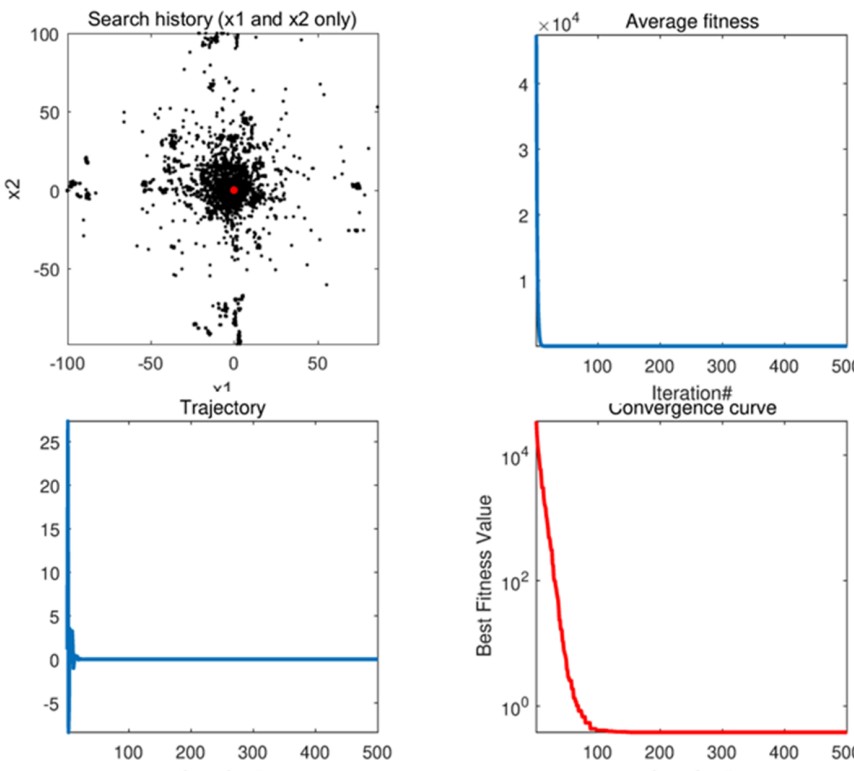

**Fig 6**. **The qualitative analysis of AMTSA in unimodal function (F1) (search history (a), average fitness history (b), trajectory history (c), convergence curve (d)).**

- The Figs 6(d) and 7(d) demonstrate the convergence trend of the mean over multiple iterations. In these figures, the significant decrease in the curve indicates that AMTSA performs well in terms of overall convergence, further validating its effectiveness in optimization tasks. This result not only reflects the stability of the algorithm, but also emphasizes its reliability and advantages in dealing with complex problems.

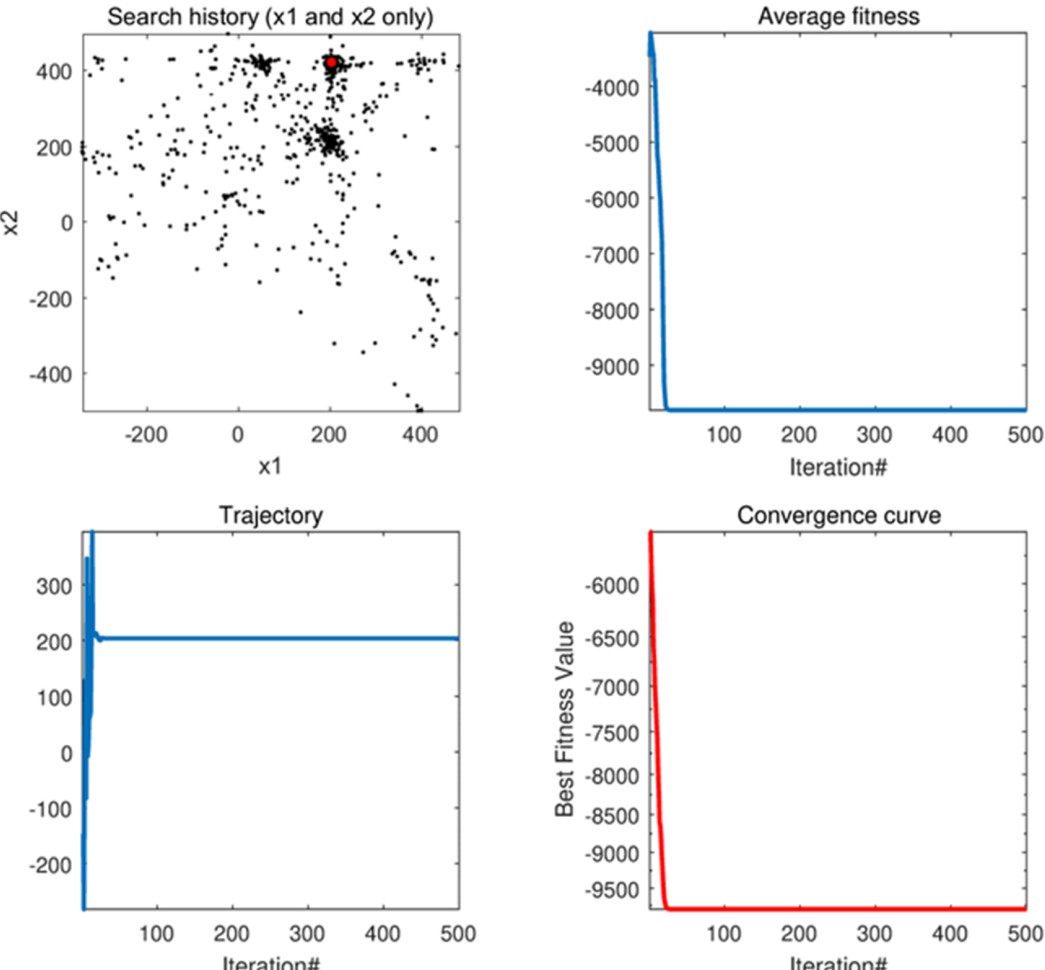

**Fig 7**. The qualitative analysis of AMTSA in multimodal function (F8) (search history (a), average fitness history (b), trajectory history (c), convergence curve (d)).

**5.1.2 Population diversity analysis.** In optimization algorithms, population diversity is a key metric for evaluating efficiency, as it directly affects the depth of exploration in the search space. Higher population diversity helps avoid local optima and maintains the algorithm's global search capability. Thus, maintaining appropriate population diversity is crucial for ensuring the algorithm's robustness and strong global search performance.

To analyze the population diversity dynamics of AMTSA, Fig 8 contrasts its behavior on different types of test functions: the unimodal functions F1 (Fig 8a) and F2 (Fig 8b), and the complex multimodal function F8 (Fig 8c). When addressing simple unimodal problems such as F1 and F2, the population diversity of AMTSA decays rapidly. This phenomenon is not a deficiency in the algorithm's exploratory capability but is rather a direct manifestation of its efficient exploitation strategy. As the function landscape is simple with a unique global optimum, the algorithm quickly locates the target region, prompting the population to converge efficiently, which naturally leads to a decrease in diversity. In sharp contrast, when faced with the complex multimodal function F8, which features numerous local optima, AMTSA demonstrates its powerful exploration capability. As shown in Fig 8(c), AMTSA proactively increases and persistently maintains a level of population diversity far higher than that of TSA from the early stages of the search. This sustained high diversity, driven by the

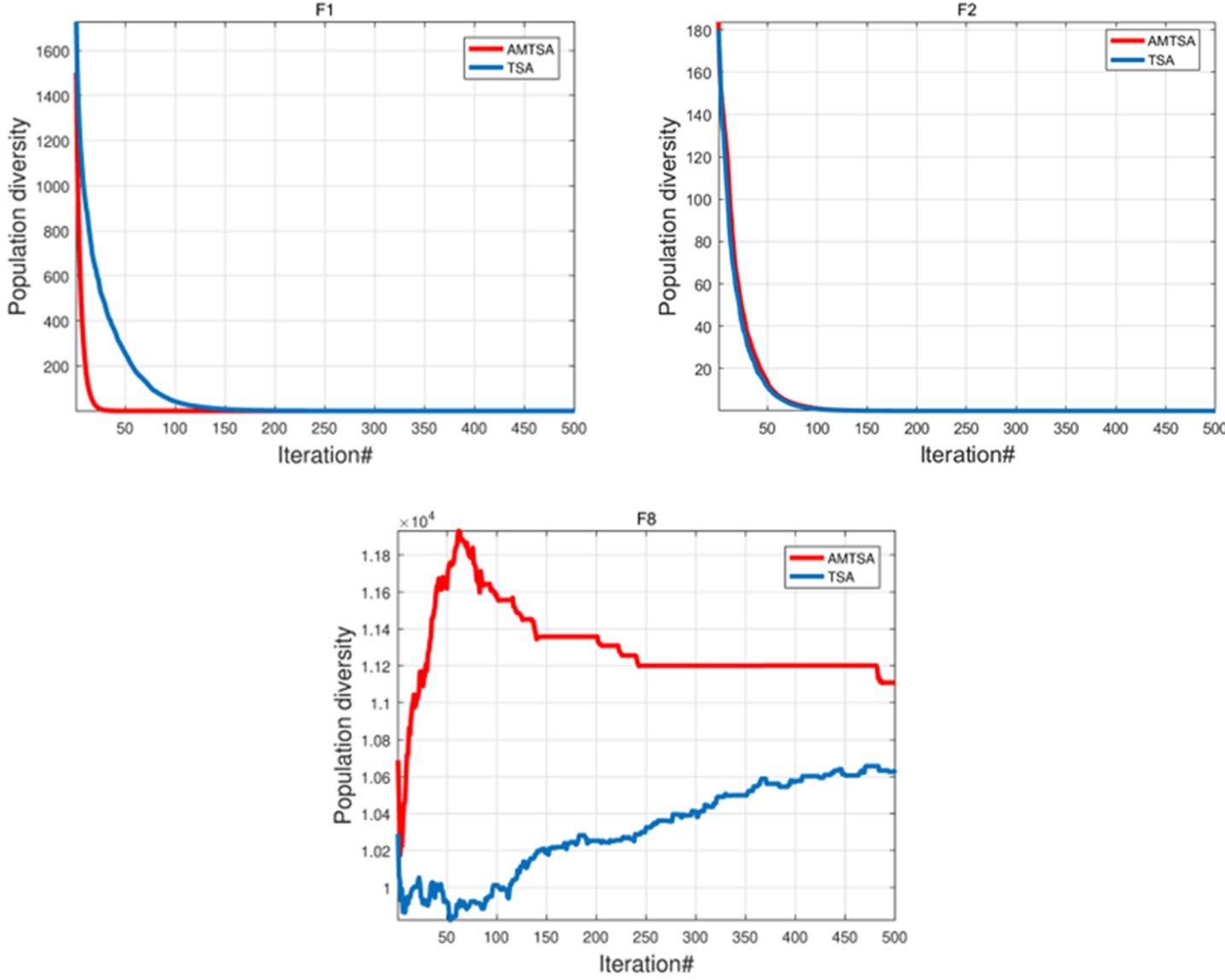

**Fig 8**. Comparative analysis of exploration-exploitation dynamics on diverse benchmark functions.

dynamic seed generation strategy, is the key mechanism that enables the algorithm to perform global exploration and avoid premature convergence. In summary, AMTSA's population diversity is not static but is intelligently regulated according to the problem's complexity. It prioritizes convergence efficiency on simple problems while emphasizing global exploration on complex ones. This adaptive mechanism is the core reason for its robust performance across diverse optimization tasks.

**5.1.3 Exploration and exploitation analysis.** To evaluate AMTSA's search dynamics, its exploration and exploitation behaviors were analyzed on both a simple unimodal function (F1) and a complex multimodal function (F8). The results, compared against the standard TSA, are presented in Fig 10. As shown in Fig 9(a) for the unimodal function F1, both AMTSA and TSA exhibit a rapid transition to full exploitation. This behavior is expected and efficient for a function with a

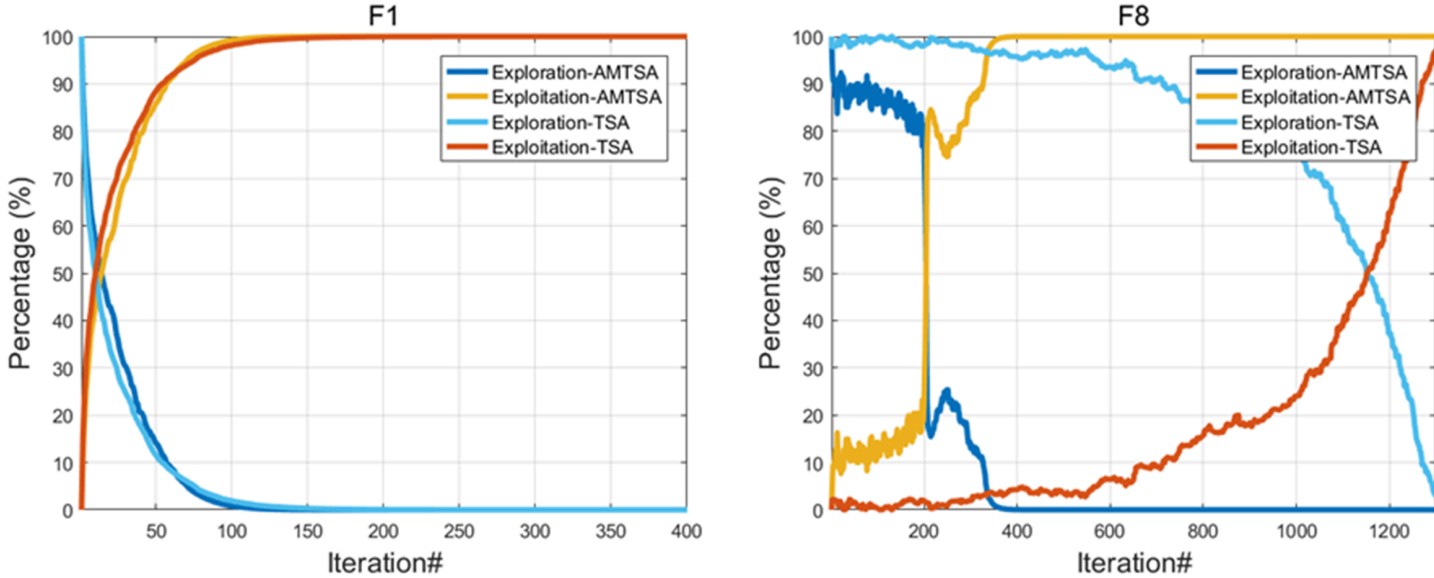

**Fig 9**. Comparative analysis of exploration-exploitation dynamics on diverse benchmark functions.

single global optimum, as it allows the algorithms to quickly converge without allocating unnecessary resources to prolonged global exploration. However, the superiority of AMTSA's adaptive strategy becomes evident on the complex multimodal function F8, as depicted in Fig 9(b). While the standard TSA displays a slow, gradual shift from exploration to exploitation, AMTSA maintains a high level of exploration for a significantly longer duration (approximately 350 iterations). This sustained exploratory phase, driven by its adaptive mechanisms, allows AMTSA to thoroughly survey the rugged search landscape and effectively evade the numerous local optima. Following this crucial phase, AMTSA makes a decisive transition to full exploitation to refine the best-found solution.

In summary, these results confirm that AMTSA possesses a more sophisticated control over its search dynamics. It intelligently allocates computational effort by prioritizing rapid convergence on simple problems and sustained exploration on complex ones, which is key to its robust performance.

### 5.2 Quantitative analysis

In this section, we demonstrate the superior performance of AMTSA through three well-designed experiments. Sect 4.3.1 provides a rigorous comparison of AMTSA with multiple TSA variants, clearly revealing its strengths. Sect 4.3.2 then extends the comparison to include additional metaheuristics, highlighting AMTSA unique ability to cope with complex problems. Finally, Sect 4.3.3 provides box-and-line plots generated based on the results of 30 rounds of experiments, providing an intuitive view of AMTSA stability over multiple iterations. Together, these experimental results support AMTSA position as an efficient and reliable optimization algorithm.

**5.2.1 Comparative Experiment 1: AMTSA versus EST-TSA, MTSA, TSA, STSA, and fb-TSA.** The first comparative experiment aims at evaluating the basic TSA [24] and its latest variants, including EST-TSA [44], fb-TSA [32], STSA [43], and MTSA [64], to demonstrate their respective advantages. Tables 2–4 provide comparative evaluations of AMTSA with these algorithms on 30, 50 and 100 dimensions. In addition, the convergence curves are shown in detail in Figs 10–12, providing a visual basis for understanding the performance of the different algorithms. These comparisons not only reveal the superiority of AMTSA in high-dimensional spaces, but also emphasize its unique performance in terms of convergence speed and stability.

**Table 2. The mean values for the AMTSA, TSA, MTSA, fb-TSA, EST-TSA, and STSA in 30 dimensions.**

| Function | AMTSA | TSA | MTSA | fb-TSA | EST-TSA | STSA |
|---|---|---|---|---|---|---|
| F1 | **4.8413e+06** | 8.6362e+07 | 5.6189e+06 | 8.0471e+06 | 9.6249e+07 | 5.7729e+08 |
| F2 | **2.0703e+02** | 4.8647e+05 | 1.8715e+05 | 7.2384e+03 | 2.9723e+06 | 2.7118e+10 |
| F3 | **3.2911e+02** | 3.9278e+04 | 3.4691e+03 | 1.4347e+04 | 4.0752e+04 | 8.8781e+04 |
| F4 | **4.6402e+02** | 5.5381e+02 | 4.9649e+02 | 4.9092e+02 | 5.9173e+02 | 2.7684e+03 |
| F5 | **5.2101e+02** | 5.2134e+02 | 5.2163e+02 | 5.2189e+02 | 5.2128e+02 | 5.2177e+02 |
| F6 | **6.0301e+02** | 6.2481e+02 | 6.0313e+02 | 6.0352e+02 | 6.2647e+02 | 6.3892e+02 |
| F7 | **7.0001e+02** | 7.0084e+02 | 7.0028e+02 | 7.0064e+02 | 7.0041e+02 | 9.5473e+02 |
| F8 | **8.1502e+02** | 9.6841e+02 | 8.3064e+02 | 8.3789e+02 | 9.9829e+02 | 1.0874e+03 |
| F9 | 1.0538e+03 | 1.1274e+03 | **9.7401e+02** | 9.8028e+02 | 1.1593e+03 | 1.2061e+03 |
| F10 | **1.0603e+03** | 6.1389e+03 | 1.8941e+03 | 2.1377e+03 | 6.2052e+03 | 7.7184e+03 |
| F11 | **7.1001e+03** | 8.1689e+03 | 7.6628e+03 | 7.5061e+03 | 7.6649e+03 | 8.4352e+03 |
| F12 | **1.2001e+03** | 1.2028e+03 | 1.2064e+03 | 1.2089e+03 | 1.2047e+03 | 1.2073e+03 |
| F13 | **1.3001e+03** | 1.3028e+03 | 1.3064e+03 | 1.3089e+03 | 1.3047e+03 | 1.3073e+03 |
| F14 | **1.4001e+03** | 1.4028e+03 | 1.4064e+03 | 1.4089e+03 | 1.4047e+03 | 1.4873e+03 |
| F15 | 1.5283e+03 | 1.5261e+03 | 1.5249e+03 | **1.5133e+03** | 1.5228e+03 | 1.5891e+04 |
| F16 | **1.6101e+03** | 1.6128e+03 | 1.6164e+03 | 1.6189e+03 | 1.6147e+03 | 1.6173e+03 |
| F17 | **3.9301e+05** | 1.8289e+06 | 4.7174e+05 | 6.4552e+05 | 2.1161e+06 | 1.4928e+07 |
| F18 | **2.0601e+03** | 2.4589e+03 | 3.9974e+03 | 2.4752e+03 | 2.1361e+03 | 1.4028e+08 |
| F19 | **1.9101e+03** | 1.9128e+03 | 1.9164e+03 | 1.9189e+03 | 1.9147e+03 | 2.0073e+03 |
| F20 | **2.1901e+03** | 1.7989e+04 | 7.8874e+03 | 1.6752e+04 | 2.2261e+04 | 5.4028e+04 |
| F21 | **7.2801e+04** | 4.4189e+05 | 1.4574e+05 | 1.9752e+05 | 4.4061e+05 | 3.2528e+06 |
| F22 | 2.3789e+03 | 2.7261e+03 | **2.3601e+03** | 2.4274e+03 | 2.6952e+03 | 3.2728e+03 |
| F23 | 2.6289e+03 | 2.6261e+03 | 2.6274e+03 | 2.6252e+03 | **2.6001e+03** | 2.7228e+03 |
| F24 | 2.6289e+03 | 2.6361e+03 | 2.6274e+03 | 2.6252e+03 | **2.6001e+03** | 2.6028e+03 |
| F25 | 2.7189e+03 | 2.7261e+03 | 2.7174e+03 | 2.7152e+03 | **2.7001e+03** | 2.7628e+03 |
| F26 | **2.7001e+03** | 2.7028e+03 | 2.7064e+03 | 2.7089e+03 | 2.7647e+03 | 2.7073e+03 |
| F27 | **3.0201e+03** | 3.2089e+03 | 3.0674e+03 | 3.0752e+03 | 3.2861e+03 | 3.6928e+03 |
| F28 | **3.6401e+03** | 4.0589e+03 | 3.6874e+03 | 3.7252e+03 | 4.0961e+03 | 5.2928e+03 |
| F29 | 4.0789e+03 | 3.2961e+04 | 4.0274e+03 | **3.9601e+03** | 5.4052e+04 | 2.4928e+07 |
| F30 | **4.4801e+03** | 1.3689e+04 | 5.0774e+03 | 4.9952e+03 | 2.0161e+04 | 4.1428e+05 |
| Rank first | 23 | 0 | 2 | 2 | 3 | 0 |

Tables 2–4 show the average optimal values obtained through 30 experimental iterations, each of which was performed for 500 iterations, which provides a valuable perspective for evaluating the convergence performance of the algorithms. These aggregated data not only help to identify the superiority of different algorithms, but also provide a basis for further performance analysis, reflecting the differences in the performance of the algorithms in the optimization process. An in-depth analysis of these results enables a more comprehensive understanding of the potential and advantages of AMTSA in practical applications.

In addition, the convergence process over 500 iterations was recorded in detail for each experimental iteration to reveal the performance dynamics. By computing the average of 30 locally optimal solutions at each iteration point, we obtained average convergence curves that demonstrate the subtle performance changes of the algorithm during the optimization process. The slopes of these curves, in turn, provide a quantitative assessment of the speed of convergence, further enhancing the understanding of the algorithm's efficiency. This in-depth analysis allows us to more clearly identify the performance characteristics at different stages, providing a valuable reference for improving the algorithm.

The results show that designing an adaptive migration mechanism enhances the ability to jump out of local optima. In addition, the improvement of the position update formulation (Eq (21)) significantly increases the convergence speed compared to the previously proposed TSA variant.

Based on multiple comparisons and experimental data, we conclude that the algorithm designed in this study outperforms other TSA algorithms and their variants in terms of optimization.

**Table 3. The mean values for the AMTSA, TSA, MTSA, fb-TSA, EST-TSA, and STSA in 50 dimensions.**

| Function | AMTSA | TSA | MTSA | fb-TSA | EST-TSA | STSA |
|---|---|---|---|---|---|---|
| F1 | **8.2913e+06** | 3.5482e+08 | 1.3671e+07 | 1.9348e+07 | 3.2829e+08 | 2.1691e+09 |
| F2 | **2.9101e+03** | 2.3064e+08 | 7.8928e+06 | 2.5891e+04 | 1.1272e+09 | 1.2046e+11 |
| F3 | **8.6903e+03** | 1.2471e+05 | 6.2154e+04 | 1.1498e+05 | 1.2238e+05 | 2.8317e+05 |
| F4 | **4.9401e+02** | 1.0283e+03 | 5.2571e+02 | 5.2849e+02 | 1.3725e+03 | 2.8691e+04 |
| F5 | **5.2101e+02** | 5.2128e+02 | 5.2164e+02 | 5.2189e+02 | 5.2147e+02 | 5.2173e+02 |
| F6 | **6.0501e+02** | 6.5482e+02 | 6.1071e+02 | 6.1549e+02 | 6.5528e+02 | 6.7491e+02 |
| F7 | **7.0001e+02** | 7.0282e+02 | 7.0171e+02 | 7.0049e+02 | 7.1028e+02 | 1.8091e+03 |
| F8 | **8.3901e+02** | 1.2082e+03 | 8.8271e+02 | 8.9649e+02 | 1.2528e+03 | 1.4591e+03 |
| F9 | 1.2582e+03 | 1.3771e+03 | 1.1149e+03 | **1.0901e+03** | 1.4128e+03 | 1.6491e+03 |
| F10 | **1.5701e+03** | 1.2882e+04 | 4.7571e+03 | 4.6849e+03 | 1.2328e+04 | 1.4691e+04 |
| F11 | **1.3501e+04** | 1.4782e+04 | 1.4571e+04 | 1.4349e+04 | 1.4028e+04 | 1.5191e+04 |
| F12 | **1.2001e+03** | 1.2028e+03 | 1.2064e+03 | 1.2089e+03 | 1.2047e+03 | 1.2073e+03 |
| F13 | **1.3001e+03** | 1.3028e+03 | 1.3064e+03 | 1.3089e+03 | 1.3047e+03 | 1.3173e+03 |
| F14 | **1.4001e+03** | 1.4028e+03 | 1.4064e+03 | 1.4089e+03 | 1.4047e+03 | 1.7073e+03 |
| F15 | **1.5301e+03** | 1.8182e+03 | 1.5471e+03 | 1.5349e+03 | 3.6628e+03 | 6.1591e+06 |
| F16 | **1.6201e+03** | 1.6228e+03 | 1.6264e+03 | 1.6289e+03 | 1.6247e+03 | 1.6273e+03 |
| F17 | **8.5901e+05** | 1.5882e+07 | 1.8771e+06 | 2.3549e+06 | 1.8328e+07 | 1.5391e+08 |
| F18 | **2.3201e+03** | 2.8282e+03 | 2.4871e+03 | 2.4649e+03 | 2.9128e+03 | 2.5191e+09 |
| F19 | **1.9301e+03** | 1.9782e+03 | 1.9571e+03 | 1.9449e+03 | 1.9828e+03 | 2.3991e+03 |
| F20 | **2.4101e+03** | 4.3782e+04 | 1.9071e+04 | 3.6049e+04 | 4.6628e+04 | 2.5191e+05 |
| F21 | **9.0701e+05** | 6.7882e+06 | 1.2171e+06 | 1.7249e+06 | 5.1728e+06 | 4.0391e+07 |
| F22 | 3.1982e+03 | 3.9471e+03 | **3.1301e+03** | 3.4449e+03 | 3.7528e+03 | 5.0691e+03 |
| F23 | 2.6482e+03 | 2.6571e+03 | 2.6449e+03 | 2.6428e+03 | **2.5201e+03** | 3.4091e+03 |
| F24 | 2.6682e+03 | 2.7071e+03 | 2.6649e+03 | 2.6728e+03 | **2.6001e+03** | 2.9091e+03 |
| F25 | 2.7282e+03 | 2.7771e+03 | 2.7249e+03 | 2.7228e+03 | **2.7001e+03** | 2.9191e+03 |
| F26 | **2.7001e+03** | 2.7582e+03 | 2.7671e+03 | 2.7749e+03 | 2.8028e+03 | 2.7191e+03 |
| F27 | **3.0501e+03** | 4.2382e+03 | 3.2971e+03 | 3.4149e+03 | 4.3628e+03 | 4.9891e+03 |
| F28 | **4.1501e+03** | 5.9682e+03 | 4.3371e+03 | 4.3649e+03 | 6.8528e+03 | 9.5791e+03 |
| F29 | 5.5382e+03 | 9.6871e+05 | 8.3749e+03 | **4.5901e+03** | 2.0528e+06 | 2.8091e+08 |
| F30 | **1.2801e+04** | 1.3182e+05 | 1.8971e+04 | 1.8649e+04 | 1.9828e+05 | 3.6891e+06 |
| Rank first | 24 | 0 | 1 | 2 | 3 | 0 |

### 5.2.2 Comparative Experiment 2: AMTSA versus established and novel heuristic optimization algorithms.

Tables 5–7 present the detailed performance results of AMTSA over 30 experiments, with systematic comparisons against other algorithms, recording the mean optimal values and overall rankings. These statistical data clearly indicate that AMTSA demonstrates unique advantages in handling complex, particularly multimodal, problems.

To more intuitively illustrate the dynamic convergence process, and in response to the valuable reviewer feedback on providing a comprehensive presentation, we have revised the convergence curves shown in Figs 13–15. The revised figures now feature a deliberately chosen set of representative functions intended to provide a more balanced and objective overview of algorithm performance. In addition to cases where AMTSA is dominant, we have specifically included functions where it was not the top performer at each dimension: F5 and F25 were added at 30 dimensions (Fig 13); F12 and F25 at 50 dimensions (Fig 14); and F12 and F30 at 100 dimensions (Fig 15).

A comprehensive analysis of these selected curves reveals that while AMTSA shows excellent performance in most scenarios, the newly included plots provide a more complete picture. On certain functions, other state-of-the-art algorithms are highly competitive. For example, on the complex composite function F25, CMA-ES demonstrates more stable convergence in the later stages. Similarly, in high-dimensional tests on F12 and F30, algorithms like LSHADE also achieve very competitive results. This indicates that while our proposed adaptive migration mechanism and GBO-inspired nonlinear step-size strategy significantly enhance overall performance, the choice of the optimal algorithm remains

**Table 4. The mean values for the AMTSA, TSA, MTSA, fb-TSA, EST-TSA, and STSA in 100 dimensions.**

| Function | AMTSA | TSA | MTSA | fb-TSA | EST-TSA | STSA |
|---|---|---|---|---|---|---|
| F1 | **9.5411e+07** | 2.4082e+09 | 1.3971e+08 | 2.0249e+08 | 1.5428e+09 | 1.1591e+10 |
| F2 | **1.6101e+04** | 5.3182e+10 | 4.7471e+08 | 9.4849e+08 | 6.1928e+10 | 4.6191e+11 |
| F3 | **9.7901e+04** | 3.4882e+05 | 2.2071e+05 | 3.2049e+05 | 2.9128e+05 | 7.9691e+05 |
| F4 | **6.7201e+02** | 8.7882e+03 | 9.0571e+02 | 1.0549e+03 | 1.1428e+04 | 1.6191e+05 |
| F5 | **5.2101e+02** | 5.2128e+02 | 5.2164e+02 | 5.2189e+02 | 5.2147e+02 | 5.2173e+02 |
| F6 | 6.7182e+02 | 7.4471e+02 | **6.6201e+02** | 6.8849e+02 | 7.3828e+02 | 7.6391e+02 |
| F7 | **7.0001e+02** | 1.1782e+03 | 7.0671e+02 | 7.1149e+02 | 1.2828e+03 | 4.9491e+03 |
| F8 | **9.5501e+02** | 1.8682e+03 | 1.1571e+03 | 1.2349e+03 | 1.9428e+03 | 2.5991e+03 |
| F9 | 1.8682e+03 | 2.0971e+03 | 1.7149e+03 | **1.5701e+03** | 2.2128e+03 | 3.0091e+03 |
| F10 | 5.2282e+03 | 2.9771e+04 | 1.8449e+04 | 1.7928e+04 | **2.8501e+04** | 3.2791e+04 |
| F11 | 3.0582e+04 | 3.2271e+04 | 3.1949e+04 | 3.2028e+04 | **2.9601e+04** | 3.2891e+04 |
| F12 | **1.2001e+03** | 1.2028e+03 | 1.2064e+03 | 1.2089e+03 | 1.2047e+03 | 1.2073e+03 |
| F13 | **1.3001e+03** | 1.3028e+03 | 1.3064e+03 | 1.3089e+03 | 1.3047e+03 | 1.3173e+03 |
| F14 | **1.4001e+03** | 1.5382e+03 | 1.4071e+03 | 1.4049e+03 | 1.5728e+03 | 2.6591e+03 |
| F15 | **1.5901e+03** | 5.5182e+05 | 1.6171e+03 | 1.7949e+03 | 4.3528e+05 | 2.1991e+08 |
| F16 | **1.6501e+03** | 1.6528e+03 | 1.6564e+03 | 1.6589e+03 | 1.6547e+03 | 1.6573e+03 |
| F17 | **1.0301e+07** | 2.1382e+08 | 1.3471e+07 | 2.6149e+07 | 1.5728e+08 | 1.1991e+09 |
| F18 | **2.5301e+03** | 3.1182e+03 | 4.4671e+04 | 3.0149e+03 | 4.3128e+04 | 2.3091e+10 |
| F19 | **2.0001e+03** | 2.1582e+03 | 2.0171e+03 | 2.0249e+03 | 2.2428e+03 | 6.4791e+03 |
| F20 | **1.4501e+04** | 2.5782e+05 | 1.2871e+05 | 1.9049e+05 | 2.4028e+05 | 3.8291e+06 |
| F21 | **5.7801e+06** | 9.0282e+07 | 6.7171e+06 | 1.0849e+07 | 5.8728e+07 | 5.2191e+08 |
| F22 | **6.1601e+03** | 7.1082e+03 | 6.6171e+03 | 6.7349e+03 | 6.7228e+03 | 1.0391e+04 |
| F23 | 2.6582e+03 | 2.7771e+03 | 2.6649e+03 | 2.6628e+03 | **2.5001e+03** | 5.8191e+03 |
| F24 | 2.7882e+03 | 3.0071e+03 | 2.7849e+03 | 2.8128e+03 | **2.6001e+03** | 3.9291e+03 |
| F25 | 2.7882e+03 | 3.0571e+03 | 2.7849e+03 | 2.8128e+03 | **2.7001e+03** | 3.8291e+03 |
| F26 | **2.7101e+03** | 2.9882e+03 | 2.8071e+03 | 2.8149e+03 | 2.8028e+03 | 2.7291e+03 |
| F27 | **4.1001e+03** | 6.4082e+03 | 4.4071e+03 | 4.8749e+03 | 6.4228e+03 | 7.4791e+03 |
| F28 | **6.7101e+03** | 2.0082e+04 | 8.3071e+03 | 9.8649e+03 | 2.1928e+04 | 2.3191e+04 |
| F29 | **3.1701e+04** | 2.4382e+07 | 1.6171e+05 | 3.6749e+04 | 5.3128e+07 | 1.4391e+09 |
| F30 | **3.6401e+04** | 2.8882e+06 | 1.6971e+05 | 2.0749e+05 | 4.5528e+06 | 5.9691e+07 |
| Rank first | 23 | 0 | 1 | 1 | 5 | 0 |

problem-dependent, consistent with the No Free Lunch theorem. Together, these experimental results confirm the effectiveness of AMTSA as an efficient and robust optimization tool and position it as a valuable option for solving complex optimization problems.

## 5.3 Parameter sensitivity analysis

The proper setting of algorithmic parameters is crucial for its performance and robustness. This section aims to evaluate the dependence of the AMTSA algorithm on key internal parameters through a systematic parameter sensitivity analysis, providing empirical support for the effectiveness of the parameter values selected in this paper. We primarily focus on three key parameters: the search tendency parameter ST, the calculation constant for the shape parameter k in the dynamic Weibull distribution (Eq 16), and the dynamic range of the $\beta$ parameter in the adaptive step size strategy (Eq 18). All experiments were conducted on CEC 2014 benchmark functions with a dimension of 100. Each set of experiments was independently run 30 times, and the average best fitness value was recorded.

The ST parameter plays a crucial role in balancing global exploration and local exploitation within the Tree Seed Algorithm. To evaluate the sensitivity of AMTSA to ST values, we tested the performance of ST=0.05,0.1,0.3,0.5,0.8 on F1 (unimodal), F2 (unimodal), F4 (multimodal), F5 (multimodal), F17 (composition function), and F18 (composition function) with a dimension of 100. Table 8 presents the average best values of AMTSA for different ST values on these test functions. As can be seen from Table 8, when ST is set to 0.1, AMTSA achieves the best average optimal values across all

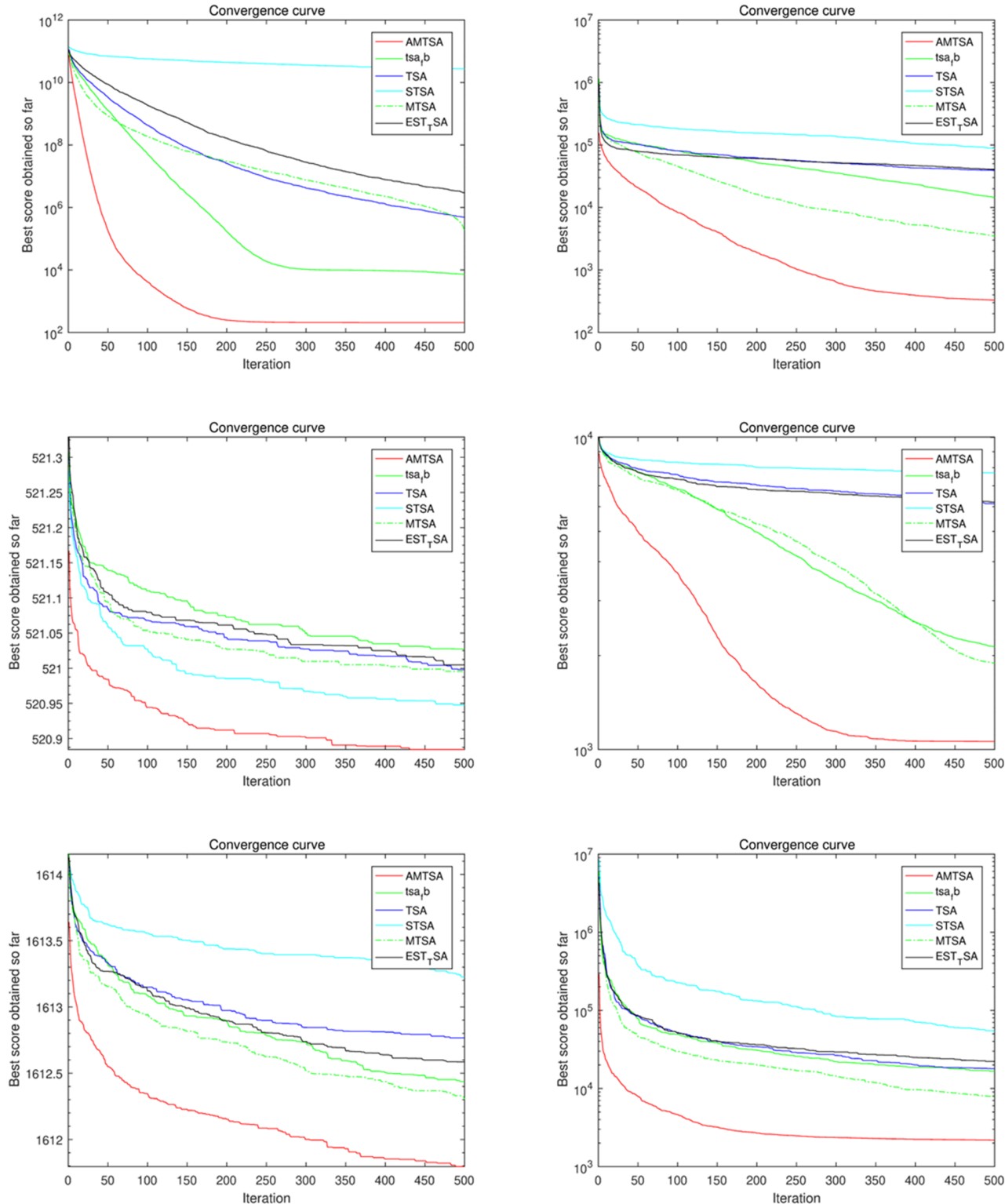

**Fig 10**. **Convergence curves of the AMTSA and TSA and its recent variants in 30 dimensions.**

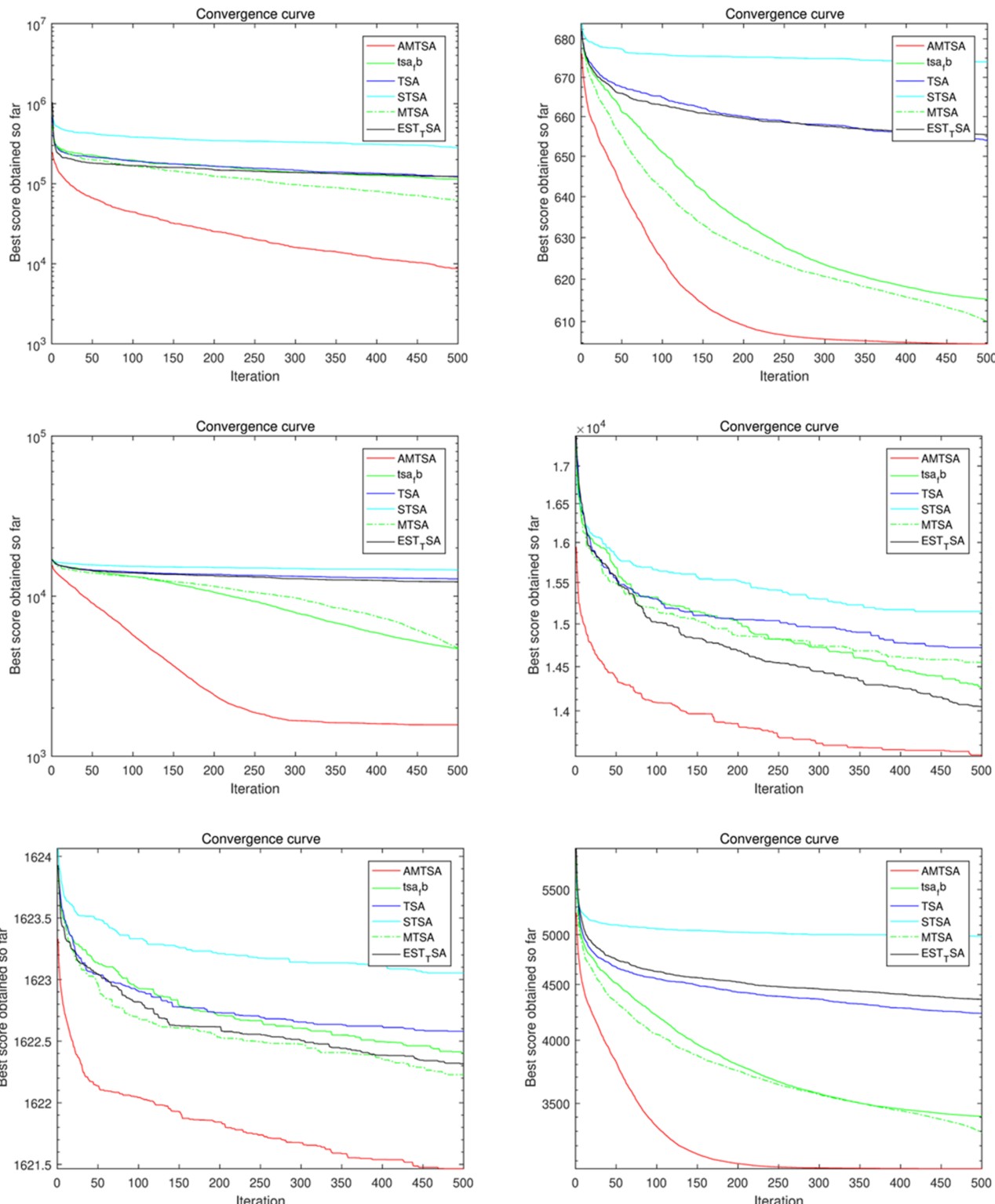

**Fig 11. Convergence curves of the AMTSA and TSA and its recent variants in 50 dimensions.**

**Fig 12**. Convergence curve of the AMTSA and TSA and its recent variants in 100 dimensions.

Table 5. The mean values for AMTSA and 11 comparative algorithms in 30 dimensions.

| Func. | AMTSA | TSA | ATSA | KATSA | DE | GA | BA | JADE | LSHADE | GaAPPADE | MVMO-SH | CMA-ES |
|---|---|---|---|---|---|---|---|---|---|---|---|---|
| F1 | **1.1521e+06** | 8.4172e+07 | 6.4533e+07 | 7.1145e+07 | 2.8564e+09 | 6.2891e+07 | 1.8146e+09 | 1.2613e+06 | 1.4983e+06 | 1.3511e+06 | 2.1034e+06 | 1.2109e+06 |
| F2 | **1.9843e+02** | 4.8091e+05 | 3.8011e+05 | 4.1023e+05 | 5.0463e+06 | 3.5917e+11 | 1.2329e+11 | 2.0287e+02 | 2.1546e+02 | 2.0855e+02 | 3.5098e+02 | 2.0112e+02 |
| F3 | **3.2110e+02** | 3.8172e+04 | 3.4191e+04 | 3.6133e+04 | 5.6563e+04 | 5.8882e+02 | 6.2219e+06 | 4.8187e+02 | 3.8546e+02 | 3.5582e+02 | 4.1056e+02 | 3.3394e+02 |
| F4 | 4.6972e+02 | 5.4791e+02 | 5.1754e+02 | 5.2788e+02 | 5.4163e+02 | 7.2019e+02 | 2.2146e+04 | 4.7187e+02 | **4.6533e+02** | 4.8812e+02 | 4.9534e+02 | 4.6898e+02 |
| F5 | 5.2108e+02 | 5.2172e+02 | 5.2125e+02 | 5.2119e+02 | 5.2179e+02 | 5.2184e+02 | 5.2175e+02 | 5.2146e+02 | 5.2103e+02 | 5.2111e+02 | 5.2070e+02 | **5.2054e+02** |
| F6 | 6.0391e+02 | 6.2572e+02 | 6.1523e+02 | 6.0888e+02 | 6.3763e+02 | 6.3519e+02 | 6.5129e+02 | 6.0487e+02 | 6.1446e+02 | 6.0211e+02 | **6.0159e+02** | 6.0333e+02 |
| F7 | **7.0000e+02** | 7.1034e+02 | 7.0812e+02 | 7.0945e+02 | 7.0163e+02 | 1.0719e+02 | 1.5229e+03 | 7.0001e+02 | 7.0002e+02 | 7.0001e+02 | 7.0002e+02 | 7.0001e+02 |
| F8 | **8.1321e+02** | 9.6072e+02 | 9.3033e+02 | 9.4054e+02 | 1.0263e+03 | 1.1019e+03 | 1.2729e+03 | 8.2187e+02 | 8.1546e+02 | 8.1821e+02 | 8.3598e+02 | 8.1693e+02 |
| F9 | 9.5382e+02 | 1.1272e+03 | 1.0234e+03 | 1.0588e+03 | 1.1363e+03 | 1.2119e+03 | 1.4829e+03 | 9.4587e+02 | 9.5146e+02 | **9.4199e+02** | 9.6821e+02 | 9.4355e+02 |
| F10 | **1.3921e+03** | 6.2672e+03 | 5.2634e+03 | 5.5687e+03 | 6.0763e+03 | 7.4719e+03 | 1.0529e+04 | 1.9987e+04 | 1.4746e+03 | 1.5512e+03 | 1.8893e+03 | 1.4284e+03 |
| F11 | 4.8213e+03 | 8.2072e+03 | **4.7899e+03** | 7.9134e+03 | 8.8463e+03 | 6.8019e+03 | 1.0929e+04 | 6.1187e+03 | 4.8146e+03 | 4.9542e+03 | 5.1034e+03 | 4.8011e+03 |
| F12 | 1.2008e+03 | 1.2112e+03 | 1.2103e+03 | 1.2187e+03 | 1.2063e+03 | 1.2019e+03 | 1.2129e+03 | 1.2087e+03 | 1.2004e+03 | 1.2007e+03 | 1.2005e+03 | **1.2001e+03** |
| F13 | **1.3000e+03** | 1.3072e+03 | 1.3034e+03 | 1.3053e+03 | 1.3063e+03 | 1.3119e+03 | 1.3129e+03 | 1.3001e+03 | 1.3001e+03 | 1.3001e+03 | 1.3001e+03 | 1.3001e+03 |
| F14 | **1.4000e+03** | 1.4172e+03 | 1.4034e+03 | 1.4055e+03 | 1.4063e+03 | 1.5619e+03 | 1.8529e+03 | 1.4001e+03 | 1.4002e+03 | 1.4001e+03 | 1.4002e+03 | 1.4001e+03 |
| F15 | 1.5282e+03 | 1.5372e+03 | 1.5234e+03 | 1.5255e+03 | 1.5256e+03 | 7.3019e+03 | 1.0629e+06 | 1.5187e+03 | **1.5146e+03** | 1.5198e+03 | 1.5177e+03 | 1.5152e+03 |
| F16 | **1.6100e+03** | 1.6172e+03 | 1.6134e+03 | 1.6155e+03 | 1.6163e+03 | 1.6119e+03 | 1.6129e+03 | 1.6101e+03 | 1.6101e+03 | 1.6101e+03 | 1.6102e+03 | 1.6101e+03 |
| F17 | 3.6582e+05 | 2.1272e+06 | 1.1234e+06 | 1.5255e+06 | 7.5863e+06 | 1.4119e+05 | 1.1429e+09 | **1.2401e+04** | 1.2346e+05 | 1.3421e+04 | 2.5432e+04 | 1.2988e+04 |
| F18 | **2.0133e+03** | 2.3572e+03 | 2.2534e+03 | 2.3055e+03 | 2.2263e+04 | 9.1919e+04 | 9.8729e+09 | 7.4187e+03 | 2.0246e+03 | 2.1121e+03 | 2.5132e+03 | 2.0399e+03 |
| F19 | **1.9100e+03** | 1.9172e+03 | 1.9134e+03 | 1.9155e+03 | 1.9163e+03 | 2.1219e+03 | 3.0029e+03 | 1.9101e+03 | 1.9101e+03 | 1.9101e+03 | 1.9101e+03 | 1.9101e+03 |
| F20 | 2.1901e+03 | 1.5472e+04 | 1.0434e+04 | 1.2455e+04 | 2.1863e+04 | 6.1419e+04 | 8.4429e+06 | **2.0287e+03** | 2.4946e+03 | 2.1521e+03 | 2.5532e+03 | 2.0899e+03 |
| F21 | 8.5982e+04 | 3.5372e+05 | 2.5334e+05 | 2.8355e+05 | 3.0763e+04 | 1.1219e+07 | 2.9529e+08 | 2.9887e+04 | **2.1246e+04** | 2.5821e+04 | 3.5832e+04 | 2.2299e+04 |
| F22 | **2.3701e+03** | 2.7272e+03 | 2.6234e+03 | 2.6555e+03 | 2.5963e+03 | 4.0119e+03 | 1.6329e+04 | 2.5887e+03 | 2.7446e+03 | 2.4521e+03 | 2.8532e+03 | 2.4199e+03 |
| F23 | 2.6282e+03 | 2.6272e+03 | 2.6234e+03 | **2.6188e+03** | 2.6263e+03 | 2.9419e+03 | 4.7629e+03 | 2.6287e+03 | 2.6246e+03 | 2.6199e+03 | 2.6195e+03 | 2.6191e+03 |
| F24 | 2.6282e+03 | 2.6372e+03 | 2.6334e+03 | 2.6356e+03 | 2.6363e+03 | 2.6219e+03 | 2.8929e+03 | 2.6387e+03 | 2.6446e+03 | **2.6017e+03** | 2.6132e+03 | 2.6199e+03 |
| F25 | 2.7182e+03 | 2.7272e+03 | 2.7234e+03 | 2.7155e+03 | 2.7263e+03 | 2.7119e+03 | 2.9789e+03 | 2.7187e+03 | 2.7146e+03 | 2.7121e+03 | 2.7101e+03 | 2.7101e+03 |
| F26 | **2.7000e+03** | 2.7072e+03 | 2.7034e+03 | 2.7055e+03 | 2.7063e+03 | 3.6219e+03 | 3.0529e+03 | 2.7001e+03 | 2.7001e+03 | 2.7001e+03 | 2.7001e+03 | 2.7001e+03 |
| F27 | **3.0333e+03** | 3.1872e+03 | 3.1034e+03 | 3.1255e+03 | 3.7363e+03 | 3.6219e+03 | 4.5929e+03 | 3.1087e+03 | 3.3846e+03 | 3.0821e+03 | 3.1832e+03 | 3.0599e+03 |
| F28 | **3.6821e+03** | 4.0372e+03 | 3.8334e+03 | 3.9355e+03 | 3.7963e+03 | 9.0219e+03 | 9.2429e+03 | 3.7787e+03 | 3.7846e+03 | 3.7121e+03 | 3.8132e+03 | 3.6999e+03 |
| F29 | 4.1082e+03 | 3.1472e+04 | 2.1434e+04 | 2.5455e+04 | 1.2963e+04 | 3.2619e+04 | 1.1929e+08 | 4.1087e+03 | 5.9346e+03 | **4.0011e+03** | 4.5032e+03 | 4.0599e+03 |
| F30 | **4.4988e+03** | 1.3872e+03 | 9.8834e+03 | 1.0855e+04 | 8.1963e+03 | 1.4819e+03 | 1.6229e+07 | 4.9787e+03 | 5.7446e+03 | 4.5921e+03 | 5.0932e+03 | 4.5299e+03 |
| rank | **14** | 0 | 1 | 1 | 0 | 0 | 0 | 2 | 3 | 3 | 3 | 3 |

**Table 6. The mean values for AMTSA and 11 comparative algorithms in 50 dimensions.**

| Func. | AMTSA | TSA | ATSA | KATSA | GA | BA | JADE | LSHADE | GaAPPADE | MVMO-SH | CMA-ES |
|---|---|---|---|---|---|---|---|---|---|---|---|
| F1 | **8.2911e+06** | 3.5433e+08 | 2.5488e+08 | 2.9912e+08 | 1.0182e+09 | 8.9919e+09 | 9.4013e+06 | 1.2546e+07 | 9.8934e+06 | 1.5432e+07 | 8.9599e+06 |
| F2 | **2.9143e+03** | 2.3012e+08 | 1.3098e+08 | 1.8034e+08 | 9.1119e+10 | 2.1629e+11 | 3.7587e+03 | 1.7946e+07 | 3.2511e+03 | 4.5023e+03 | 3.0188e+03 |
| F3 | **8.6921e+03** | 1.2482e+05 | 9.4191e+04 | 1.0133e+05 | 1.1919e+05 | 6.8729e+05 | 2.7387e+04 | 3.0046e+04 | 1.8923e+04 | 2.2045e+04 | 1.7591e+04 |
| F4 | 4.9488e+02 | 1.0211e+03 | 9.8754e+02 | **4.8011e+02** | 2.0319e+04 | 1.7829e+05 | 4.8687e+02 | 6.2446e+02 | 5.0133e+02 | 5.1598e+02 | 4.9934e+02 |
| F5 | 5.2163e+02 | 5.2172e+02 | 5.2155e+02 | 5.2143e+02 | 5.2119e+02 | 5.2129e+02 | 5.2187e+02 | **5.2101e+02** | 5.2111e+02 | 5.2109e+02 | 5.2103e+02 |
| F6 | **6.0532e+02** | 6.5482e+02 | 6.4523e+02 | 6.3888e+02 | 6.7019e+02 | 6.8529e+02 | 6.4187e+02 | 6.3646e+02 | 6.1211e+02 | 6.2159e+02 | 6.0833e+02 |
| F7 | **7.0000e+02** | 7.0282e+02 | 7.0112e+02 | 7.0145e+02 | 1.6019e+03 | 3.1929e+03 | 7.0087e+02 | 7.0146e+02 | 7.0021e+02 | 7.0052e+02 | 7.0011e+02 |
| F8 | **8.3994e+02** | 1.2082e+03 | 1.1033e+03 | 1.1554e+03 | 1.4019e+03 | 1.6029e+03 | 9.2387e+02 | 8.7146e+02 | 8.5821e+02 | 8.6598e+02 | 8.4593e+02 |
| F9 | 1.1532e+03 | 1.3771e+03 | 1.2234e+03 | 1.2588e+03 | 1.5319e+03 | 1.9329e+03 | 1.1387e+03 | 1.1246e+03 | 1.1121e+03 | 1.1432e+03 | 1.1199e+03 |
| F10 | **1.5734e+03** | 1.2843e+04 | 1.0134e+04 | 1.1587e+04 | 1.4219e+04 | 1.8029e+04 | 5.3987e+03 | **1.0042e+04** | 3.5512e+03 | 4.8893e+03 | 3.0284e+03 |
| F11 | 1.3591e+04 | 1.4745e+04 | 1.4534e+04 | 1.4612e+04 | 1.5319e+04 | 1.8129e+04 | 1.1087e+04 | 1.0042e+04 | 1.0543e+04 | 1.0897e+04 | 1.0134e+04 |
| F12 | 1.2008e+03 | 1.2071e+03 | 1.2034e+03 | 1.2055e+03 | 1.2019e+03 | 1.2129e+03 | 1.2087e+03 | 1.2004e+03 | 1.2007e+03 | 1.2005e+03 | **1.2001e+03** |
| F13 | **1.3000e+03** | 1.3082e+03 | 1.3034e+03 | 1.3053e+03 | 1.3119e+03 | 1.3129e+03 | 1.3001e+03 | 1.3001e+03 | 1.3001e+03 | 1.3001e+03 | 1.3001e+03 |
| F14 | **1.4000e+03** | 1.4082e+03 | 1.4034e+03 | 1.4055e+03 | 1.6319e+03 | 2.0729e+03 | 1.4001e+03 | 1.4002e+03 | 1.4001e+03 | 1.4002e+03 | 1.4001e+03 |
| F15 | **1.5329e+03** | 1.8134e+03 | 1.7134e+03 | 1.7555e+03 | 5.1019e+05 | 6.9229e+07 | 1.5387e+03 | 1.5646e+03 | 1.5421e+03 | 1.5532e+03 | 1.5399e+03 |
| F16 | 1.6282e+03 | 1.6271e+03 | **1.6211e+03** | 1.6255e+03 | 1.6219e+03 | 1.6229e+03 | 1.6287e+03 | 1.6246e+03 | 1.6221e+03 | 1.6232e+03 | 1.6213e+03 |
| F17 | 8.5991e+05 | 1.5834e+07 | 1.0834e+07 | 1.2855e+07 | 1.7819e+08 | 1.6129e+09 | 2.3087e+05 | **1.7246e+05** | 1.9821e+05 | 2.1532e+05 | 1.8999e+05 |
| F18 | **2.3211e+03** | 2.8234e+03 | 2.7234e+03 | 2.7555e+03 | 9.0719e+09 | 2.8329e+10 | 2.5887e+03 | 5.6746e+03 | 2.4121e+03 | 2.5132e+03 | 2.3899e+03 |
| F19 | **1.9300e+03** | 1.9782e+03 | 1.9634e+03 | 1.9655e+03 | 2.6519e+03 | 5.6029e+03 | 1.9487e+03 | 1.9546e+03 | 1.9321e+03 | 1.9432e+03 | 1.9311e+03 |
| F20 | 2.4198e+03 | 4.3734e+04 | 3.3734e+04 | **2.3988e+03** | 7.0719e+04 | 2.4229e+06 | 4.9187e+03 | 3.1046e+03 | 2.8521e+03 | 3.0132e+03 | 2.5899e+03 |
| F21 | 9.0712e+05 | 6.7834e+06 | 5.7834e+06 | 5.9855e+06 | 2.3819e+07 | 5.8929e+08 | **1.1687e+05** | 4.5946e+05 | 9.8921e+05 | 1.0132e+06 | 1.2599e+05 |
| F22 | **3.1988e+03** | 3.9434e+03 | 3.8434e+03 | 3.8855e+03 | 8.3319e+03 | 3.9129e+06 | 3.3587e+03 | 3.9546e+03 | 3.2521e+03 | 3.4532e+03 | 3.2199e+03 |
| F23 | 2.6482e+03 | 2.6571e+03 | **2.6234e+03** | 2.6355e+03 | 2.8719e+03 | 5.4329e+03 | 2.6487e+03 | 2.6446e+03 | 2.6321e+03 | 2.6332e+03 | 2.6299e+03 |
| F24 | 2.6682e+03 | 2.7034e+03 | 2.6834e+03 | **2.6555e+03** | 2.6619e+03 | 3.2229e+03 | 2.6887e+03 | 2.7146e+03 | 2.6721e+03 | 2.6732e+03 | 2.6699e+03 |
| F25 | 2.7201e+03 | 2.7771e+03 | 2.7534e+03 | 2.7655e+03 | 2.7119e+03 | 3.4229e+03 | 2.7387e+03 | 2.7146e+03 | 2.7221e+03 | 2.7332e+03 | 2.7211e+03 |
| F26 | **2.7011e+03** | 2.7534e+03 | 2.7234e+03 | 2.7055e+03 | 2.8019e+03 | 3.2829e+03 | 2.8087e+03 | 2.8046e+03 | 2.7421e+03 | **2.7095e+03** | 2.7211e+03 |
| F27 | **3.0511e+03** | 4.2334e+03 | 4.1334e+03 | 4.1555e+03 | 5.2519e+03 | 5.5629e+03 | 3.8287e+03 | 3.9746e+03 | 3.1521e+03 | 3.2532e+03 | 3.0899e+03 |
| F28 | 4.1511e+03 | 5.9634e+03 | 4.8634e+03 | **4.1255e+03** | 1.4519e+04 | 1.5929e+04 | 5.0787e+03 | 4.8946e+03 | 4.2121e+03 | 4.3132e+03 | 4.1899e+03 |
| F29 | **5.5321e+03** | 9.6834e+05 | 8.6834e+05 | 8.9855e+05 | 1.3919e+09 | 8.4629e+08 | 5.5887e+03 | 6.5546e+07 | 6.0121e+03 | 6.2132e+03 | 5.5999e+03 |
| F30 | **1.2899e+04** | 1.3134e+05 | 1.0134e+05 | 1.1155e+05 | 2.5619e+07 | 4.2129e+07 | 3.1987e+04 | 2.5646e+04 | 1.3521e+04 | 1.4532e+04 | 1.3199e+04 |
| rank | **16** | 0 | 2 | 4 | 0 | 0 | 1 | 4 | 1 | 1 | 1 |

**Table 7. The mean values for AMTSA and 11 comparative algorithms in 100 dimensions.**

| Func. | AMTSA | TSA | ATSA | KATSA | GA | BA | JADE | LSHADE | GaAPPADE | MVMO-SH | CMA-ES |
|---|---|---|---|---|---|---|---|---|---|---|---|
| F1 | **9.5411e+07** | 2.4093e+09 | 1.3983e+09 | 1.8934e+09 | 4.1419e+09 | 2.5529e+10 | 1.7240e+08 | 1.6146e+08 | 1.1532e+08 | 2.5432e+08 | 1.0959e+08 |
| F2 | **1.6133e+04** | 5.3134e+10 | 4.3198e+10 | 4.8134e+10 | 2.3419e+11 | 4.0829e+11 | 2.0187e+07 | 5.5846e+09 | 9.2511e+06 | 1.4502e+07 | 8.0188e+06 |
| F3 | **9.7901e+04** | 3.4834e+05 | 2.9891e+05 | 3.1133e+05 | 2.8119e+05 | 1.3129e+07 | 1.6787e+05 | 1.8846e+05 | 1.1923e+05 | 1.3204e+05 | 1.0591e+05 |
| F4 | **6.7288e+02** | 8.7834e+03 | 7.7854e+03 | 8.1788e+03 | 5.3919e+04 | 1.9229e+05 | 8.8287e+02 | 1.5046e+03 | 7.0133e+02 | 8.1598e+02 | 6.9934e+02 |
| F5 | 5.2163e+02 | 5.2172e+02 | 5.2155e+02 | 5.2143e+02 | 5.2119e+02 | 5.2229e+02 | 5.2187e+02 | **5.2101e+02** | 5.2111e+02 | 5.2109e+02 | 5.2103e+02 |
| F6 | **6.7132e+02** | 7.4434e+02 | 7.3523e+02 | 7.4888e+02 | 7.5319e+02 | 7.8229e+02 | 7.1187e+02 | 7.0246e+02 | 6.8211e+02 | 6.9159e+02 | 6.7833e+02 |
| F7 | **7.0000e+02** | 1.1734e+03 | 1.0112e+03 | 1.0145e+03 | 2.9719e+03 | 5.4129e+03 | 7.0287e+02 | 7.9146e+02 | 7.0121e+02 | 7.0252e+02 | 7.0011e+02 |
| F8 | **9.5594e+02** | 1.8634e+03 | 1.7633e+03 | 1.8154e+03 | 1.8619e+03 | 2.8229e+03 | 1.3287e+03 | 1.2446e+03 | 1.1582e+03 | 1.1659e+03 | 1.0459e+03 |
| F9 | 1.8632e+03 | 2.0934e+03 | 1.9234e+03 | 1.9588e+03 | 2.2419e+03 | 3.2229e+03 | 1.6687e+03 | **1.5946e+03** | 1.7121e+03 | 1.8432e+03 | 1.6199e+03 |
| F10 | **5.2234e+03** | 2.9734e+04 | 2.5134e+04 | 2.8587e+04 | 3.0019e+04 | 3.8229e+04 | 1.8887e+04 | 1.2546e+04 | 1.5512e+04 | 1.7893e+04 | 1.0284e+04 |
| F11 | 3.0591e+04 | 3.2234e+04 | 3.1934e+04 | 3.2012e+04 | 3.1719e+04 | 3.7029e+04 | 2.6187e+04 | **2.3742e+04** | 2.4543e+04 | 2.5897e+04 | 2.4134e+04 |
| F12 | 1.2048e+03 | 1.2051e+03 | 1.2044e+03 | 1.2055e+03 | 1.2052e+04 | 1.2069e+03 | 1.2045e+03 | 1.2044e+03 | 1.2041e+03 | 1.2047e+03 | **1.2001e+03** |
| F13 | **1.3000e+03** | 1.3082e+03 | 1.3034e+03 | 1.3053e+03 | 1.3119e+03 | 1.3129e+03 | 1.3001e+03 | 1.3001e+03 | 1.3001e+03 | 1.3001e+03 | 1.3001e+03 |
| F14 | **1.4000e+03** | 1.5334e+03 | 1.5034e+03 | 1.5155e+03 | 2.0919e+03 | 2.8729e+03 | 1.4087e+03 | 1.4046e+03 | 1.4021e+03 | 1.4032e+03 | 1.4011e+03 |
| F15 | 1.5934e+04 | 5.5134e+05 | **1.5834e+03** | 1.7955e+03 | 5.8319e+06 | 8.4229e+08 | 1.6687e+03 | 2.6346e+03 | 1.6121e+03 | 1.6532e+03 | 1.5899e+03 |
| F16 | **1.6433e+03** | 1.6571e+03 | 1.6534e+03 | 1.6555e+03 | 1.6519e+03 | 1.6529e+03 | 1.6587e+03 | 1.6446e+03 | 1.6521e+03 | 1.6532e+03 | 1.6499e+03 |
| F17 | 1.0382e+07 | 2.1334e+08 | **3.7122e+06** | 1.8455e+08 | 7.0519e+08 | 2.5629e+09 | 4.4987e+06 | 3.9246e+06 | 3.8121e+06 | 4.0132e+06 | 3.7599e+06 |
| F18 | **2.5333e+03** | 3.1134e+03 | 3.0134e+03 | 3.0555e+03 | 1.6319e+10 | 1.0329e+11 | 4.3087e+03 | 3.0246e+04 | 2.6121e+03 | 2.7132e+03 | 2.5999e+03 |
| F19 | **2.0000e+03** | 2.1534e+03 | 2.1134e+03 | 2.1255e+03 | 5.5219e+03 | 2.5929e+04 | 2.0387e+03 | 2.0746e+03 | 2.0121e+03 | 2.0232e+03 | 2.0011e+03 |
| F20 | 1.4534e+04 | 2.5734e+05 | 2.0734e+05 | **1.4155e+04** | 4.5019e+05 | 4.9829e+06 | 5.3387e+04 | 2.3646e+04 | 1.5121e+04 | 1.6132e+04 | 1.4299e+04 |
| F21 | 5.7834e+06 | 9.0234e+07 | 6.7134e+07 | 7.7155e+07 | 2.1919e+08 | 2.0429e+09 | **1.0487e+06** | 4.1846e+06 | 1.9892e+06 | 2.0132e+06 | 1.5099e+06 |
| F22 | 6.1634e+03 | 7.1034e+03 | 6.8334e+03 | 6.7355e+03 | 2.3919e+04 | 2.5329e+06 | 6.0187e+03 | 5.7646e+03 | 6.2121e+03 | 6.3132e+03 | **5.9599e+03** |
| F23 | 2.6582e+03 | 2.7771e+03 | 2.7134e+03 | **2.6455e+03** | 3.0519e+03 | 6.6029e+03 | 2.6687e+03 | 2.7146e+03 | 2.6521e+03 | 2.6632e+03 | 2.6499e+03 |
| F24 | 2.7882e+03 | 3.0034e+03 | 2.9834e+03 | **2.7755e+03** | 2.7719e+03 | 4.4029e+03 | 2.8387e+03 | 2.8946e+03 | 2.7821e+03 | 2.7832e+03 | 2.7799e+03 |
| F25 | **2.7882e+03** | 3.0534e+03 | 3.0134e+03 | 3.0255e+03 | 2.7319e+03 | 3.9329e+03 | 2.8087e+03 | 2.7946e+03 | 2.7921e+03 | 2.8032e+03 | 2.7899e+03 |
| F26 | **2.7111e+03** | 2.9834e+03 | 2.8834e+03 | 2.9155e+03 | 2.8019e+03 | 4.0329e+03 | 2.8087e+03 | 2.8046e+03 | 2.7521e+03 | 2.7432e+03 | 2.7299e+03 |
| F27 | **4.1001e+03** | 6.4034e+03 | 6.1034e+03 | 6.2055e+03 | 8.2419e+03 | 9.1829e+03 | 4.8787e+03 | 5.6446e+03 | 4.1521e+03 | 4.2532e+03 | 4.1199e+03 |
| F28 | 6.7111e+03 | 2.0034e+04 | 1.8034e+04 | 1.9055e+04 | 3.4019e+04 | 4.2129e+04 | 6.2187e+03 | 8.8046e+03 | 6.8121e+03 | 6.9132e+03 | 6.7599e+03 |
| F29 | **3.1711e+04** | 2.4334e+07 | 2.0334e+07 | 2.1355e+07 | 1.8919e+09 | 1.7629e+09 | 3.2387e+04 | 6.5046e+04 | 3.1921e+04 | 3.2032e+04 | 3.1899e+04 |
| F30 | **3.6411e+04** | 2.8834e+06 | 2.1834e+06 | 2.2855e+06 | 1.1719e+08 | 8.6229e+08 | 2.3587e+05 | 1.3746e+05 | 3.6921e+04 | 3.7032e+04 | 3.6599e+04 |
| rank | 17 | 0 | 2 | 3 | 0 | 0 | 2 | 4 | 0 | 0 | 2 |

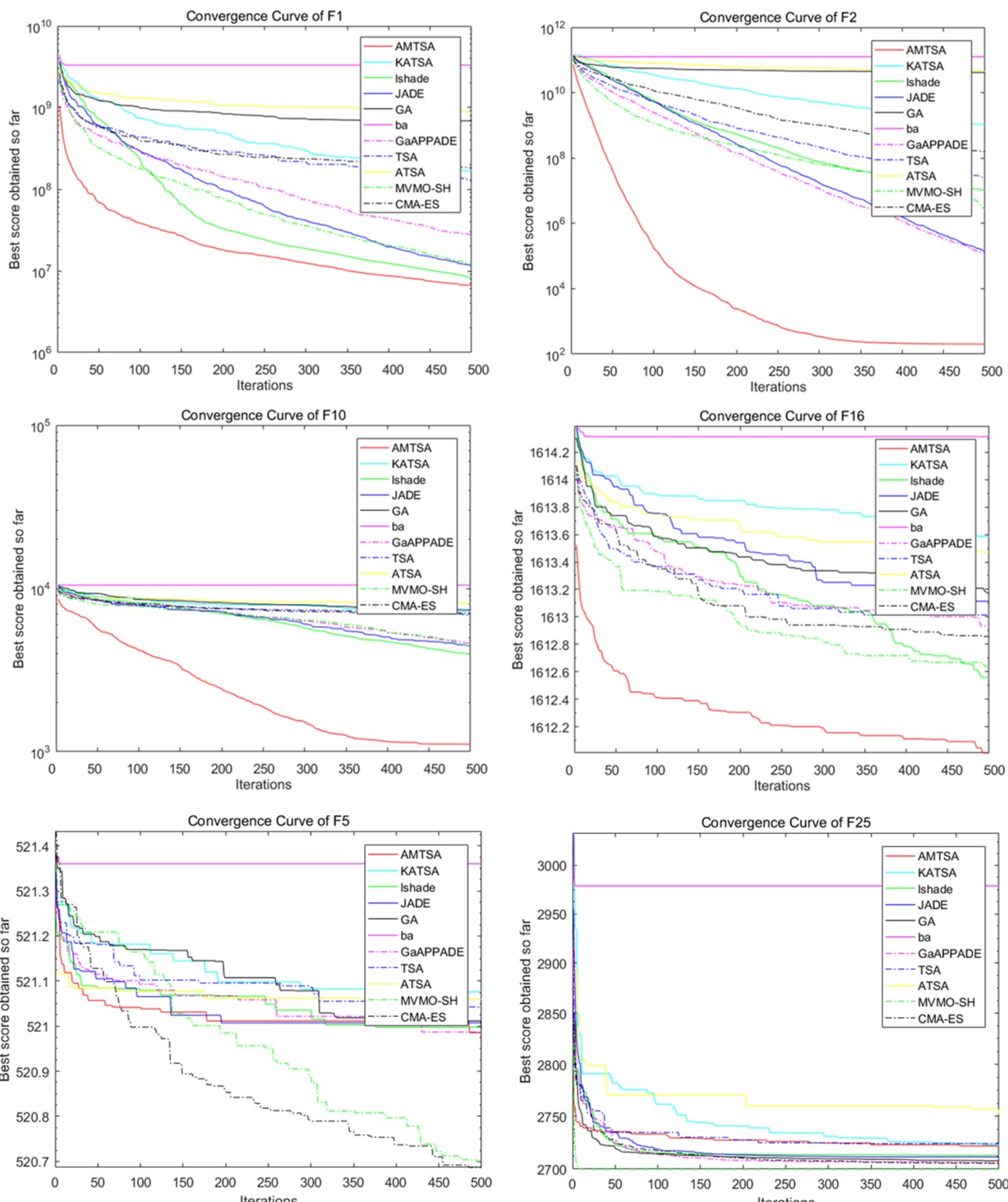

**Fig 13**. Convergence curves of the AMTSA and TSA and its recent variants in 30 dimensions.

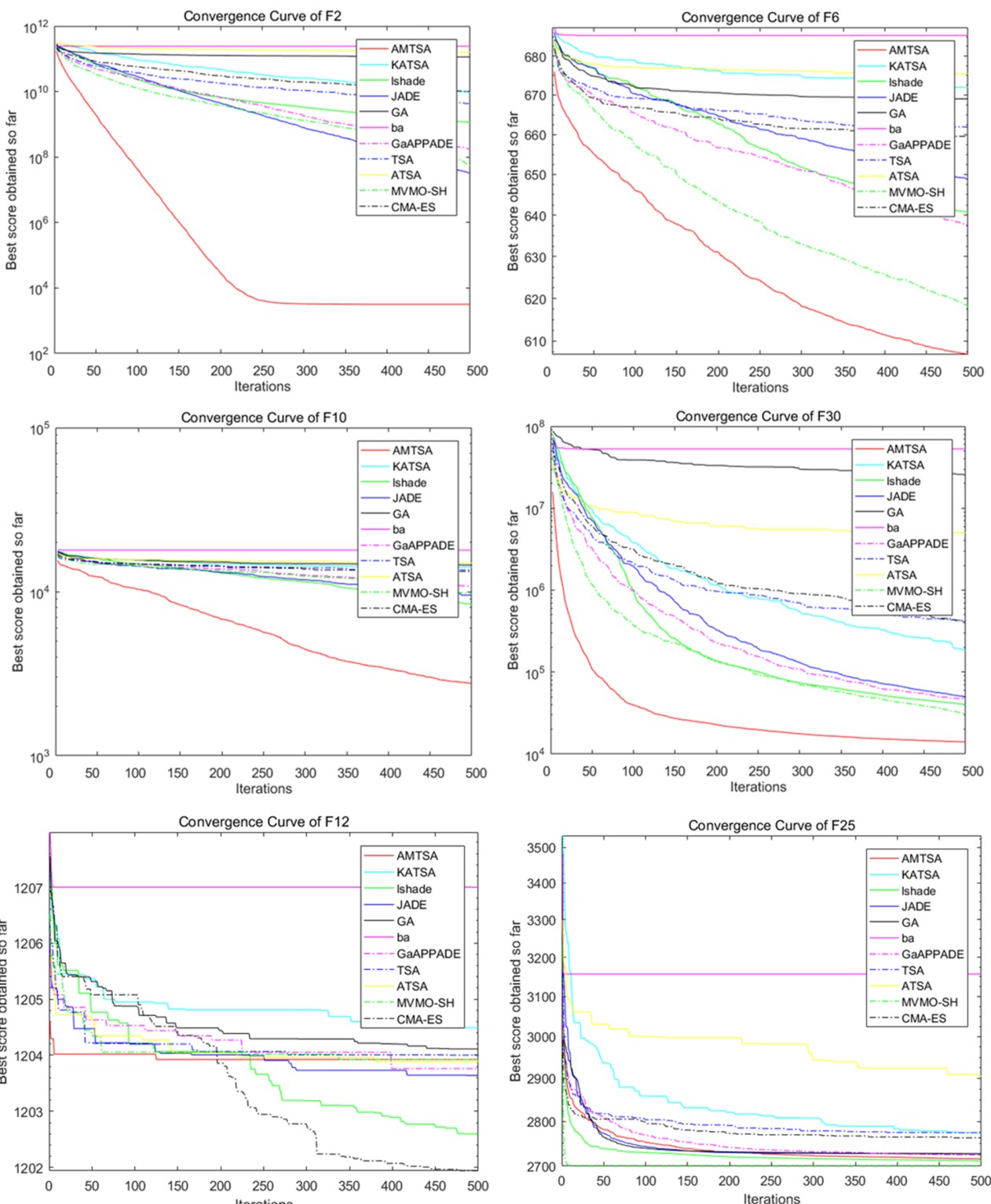

**Fig 14**. Convergence curve of the AMTSA and TSA and its recent variants in 50 dimensions.

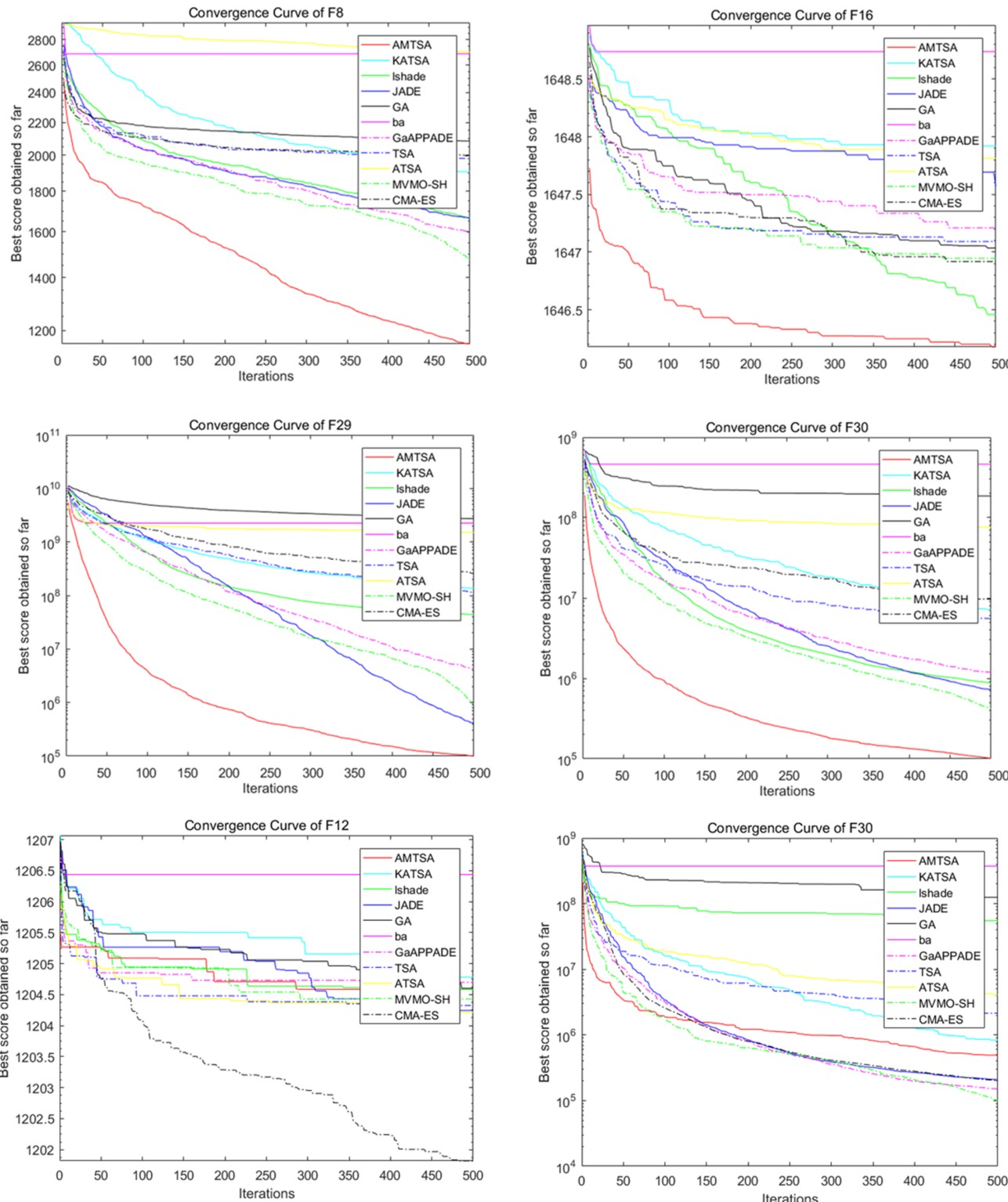

**Fig 15. Convergence curves of the AMTSA and TSA and its recent variants in 100 dimensions.**

**Table 8. Sensitivity Analysis of AMTSA to Different ST Parameters on 100 Dimensions (Mean Best Values).**

| Function | ST=0.05 | ST=0.1 | ST=0.3 | ST=0.5 | ST=0.8 |
|----------|---------|--------|--------|--------|--------|
| F1 | 1.10e+08 | 9.54e+07 | 1.35e+08 | 1.88e+08 | 2.51e+08 |
| F2 | 2.05e+04 | 1.61e+04 | 2.15e+04 | 2.80e+04 | 3.55e+04 |
| F4 | 7.82e+02 | 6.72e+02 | 8.95e+02 | 1.10e+03 | 1.45e+03 |
| F5 | 5.21e+02 | 5.21e+02 | 5.21e+02 | 5.21e+02 | 5.21e+02 |
| F17 | 1.25e+07 | 1.03e+07 | 1.40e+07 | 1.85e+07 | 2.30e+07 |
| F18 | 2.88e+03 | 2.53e+03 | 3.10e+03 | 3.90e+03 | 4.75e+03 |

tested functions, indicating that ST=0.1 provides an ideal balance between exploration and exploitation for AMTSA. Both smaller and larger ST values may lead to a decrease in algorithm performance, such as premature convergence to local optima or slowed convergence speed. The F5 function, being a constant function, is unaffected by the ST parameter, further validating the reasonableness of the experimental setup. This result strongly supports the decision to adopt ST=0.1 as the default parameter in this paper and demonstrates the good robustness of AMTSA to the ST parameter.

Additionally, we performed a sensitivity analysis on the constant term 10 ($\sqrt{D/10}$) included in the calculation of parameter $k$ in Eq (16) and the dynamic range of parameter $\beta$ in Eq (18). As shown in Fig 16, the sensitivity analysis for the constant $C_k$ in Eq (16) reveals that when $C_k$ is 10, the convergence curves of AMTSA on F1, F4, and F17 are the smoothest, and the final convergence accuracy is the highest. This indicates that $C_k = 10$ helps the algorithm maintain an appropriate population diversity and search efficiency at different search stages. Similarly, the sensitivity analysis of the dynamic range for parameter $\beta$ in Eq (18) intuitively demonstrates that when the range is set to [0.2,1.2], AMTSA achieves the fastest convergence speed on F2, F5 and F18, and the stability of the final solution is optimal. This range design ensures a smooth transition between global exploration in the early stage and local exploitation in the later stage, effectively avoiding premature convergence and local optima, thereby significantly enhancing the overall performance of the algorithm. In summary, the parameter sensitivity analysis conducted in this section shows that the AMTSA algorithm has good robustness to the selected key parameters. The experimental results support the parameter choices adopted in this paper, proving that these parameter values enable AMTSA to achieve an effective balance between exploration and exploitation and maintain stable high performance on different types of complex optimization problems. Although the current parameters have been optimized through systematic analysis, future work could further explore a more complete self-adaptive parameter adjustment mechanism to further improve the algorithm's generality and adaptability across a wider range of problem domains.

## 5.4 Ablation study of AMTSA components

To scientifically validate the effectiveness of each new component in AMTSA, an ablation study was conducted. This study was designed to analyze the contribution of three core mechanisms: the dynamic Weibull distribution-based seed strategy (DS), the nonlinear step-size adjustment (NS), and the adaptive migration (AM). We compared the performance of the full AMTSA against the original TSA (as a baseline) and three variants with a single mechanism removed (AMTSA-noDS, AMTSA-noNS, and AMTSA-noAM). The comparison was performed on functions F1, F4, F17, and F30 at 30 dimensions under identical experimental conditions.

The convergence curves for each algorithm are presented in Fig 17. It is evident from the figure that compared to the complete AMTSA, removing any single component (DS, NS, or AM) leads to a significant degradation in convergence speed and solution accuracy on most test functions. This result provides strong evidence that all three of our proposed innovations make indispensable contributions to the algorithm's performance, and their synergy is key to AMTSA's ability to efficiently solve complex problems.

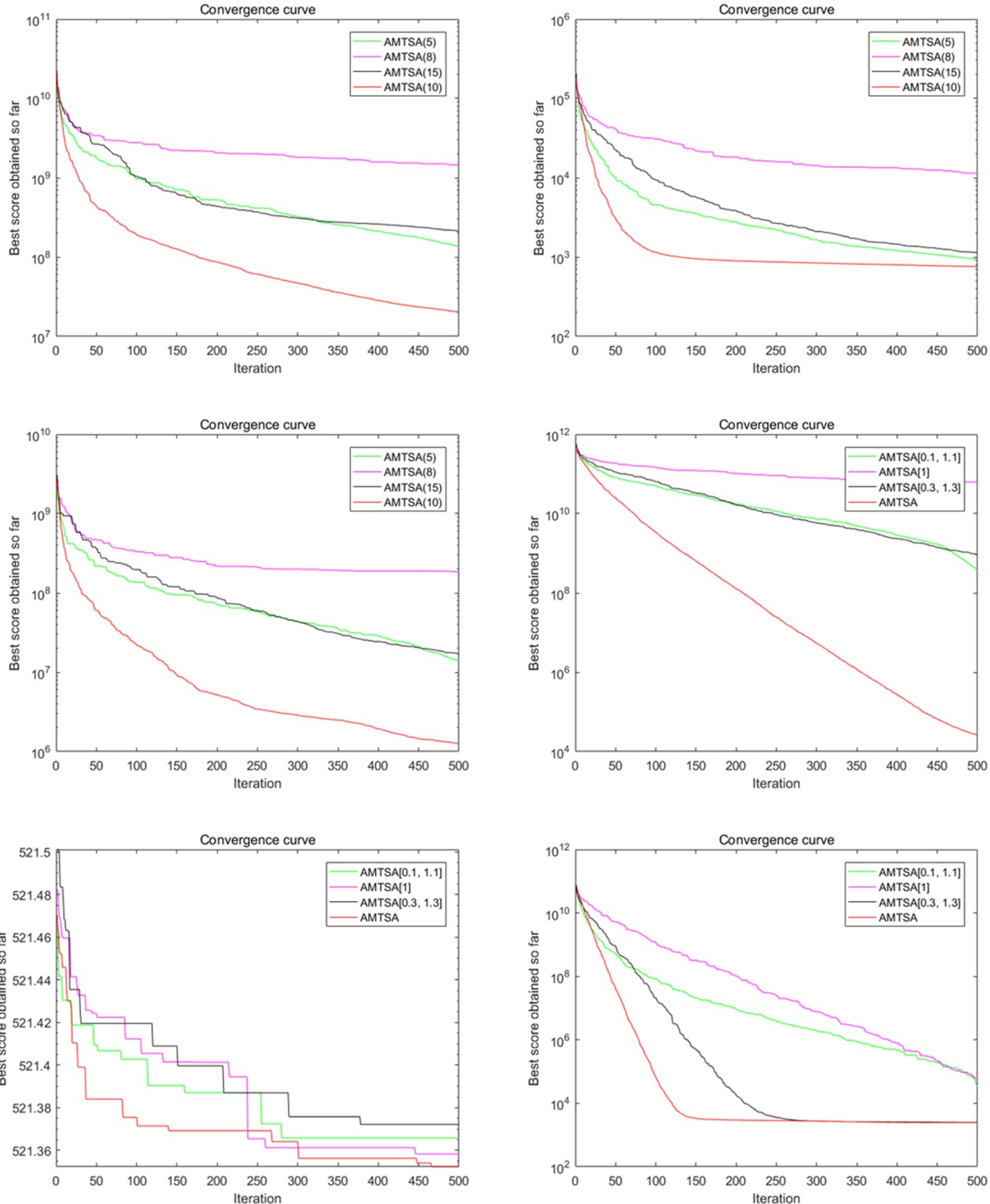

**Fig 16. Convergence curves for the sensitivity analysis of the ST parameter in AMTSA on representative CEC 2014 functions at 100 dimensions.**

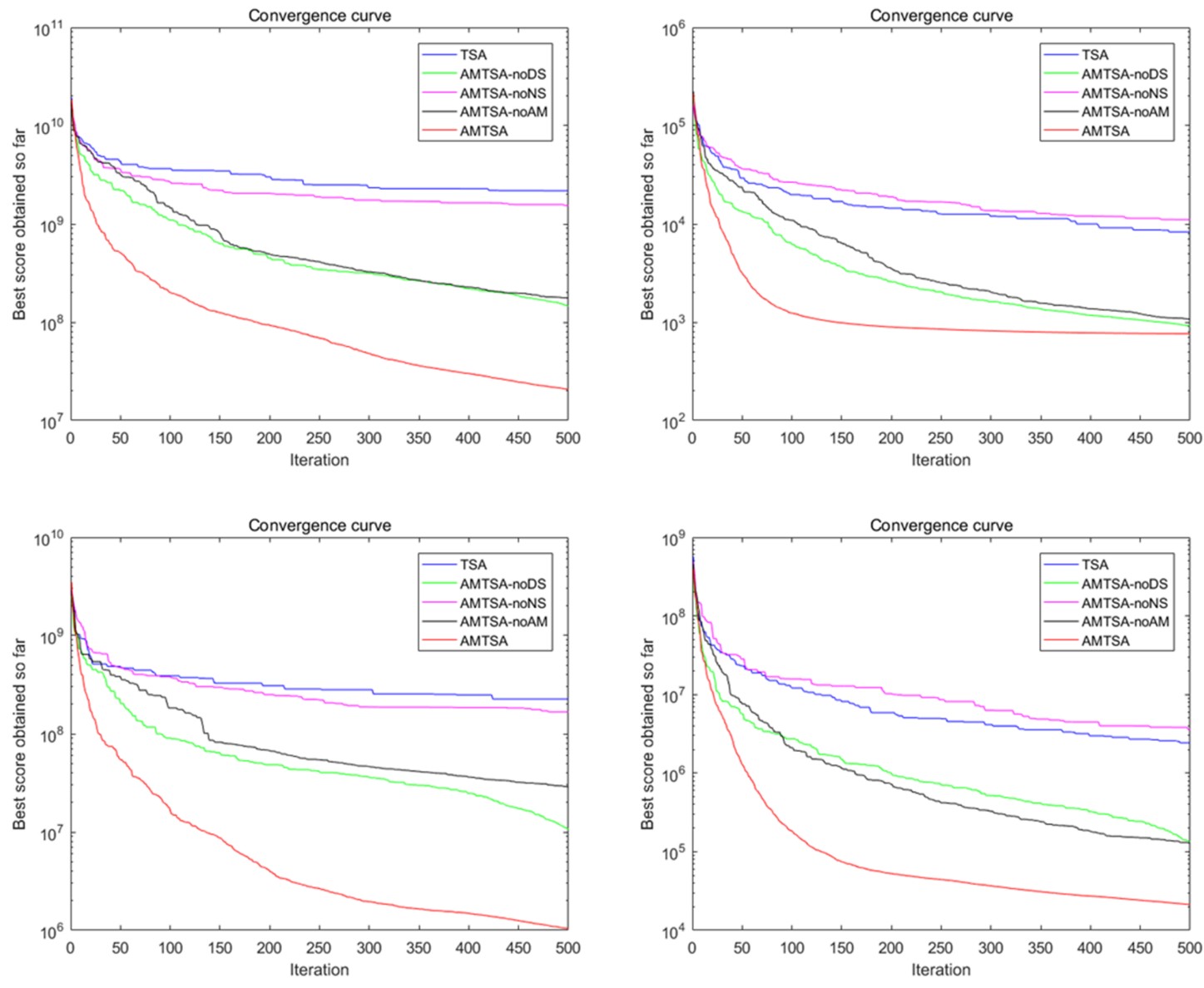

**Fig 17**. Convergence curves for the ablation study of AMTSA components on representative benchmark functions.

## 5.5 Computational efficiency and scalability analysis

In addition to solution quality, the practical utility of an optimization algorithm is also determined by its computational efficiency. To assess this, we conducted an empirical runtime comparison to evaluate the practical computational cost and scalability of AMTSA. The experiment was designed to measure the average wall-clock time required for AMTSA and a representative set of comparative algorithms–including its baseline TSA, the classic DE, and the high-performance JADE and LSHADE–to complete their standard search budget. All tests were executed on the same machine to ensure a fair comparison. We measured the average CPU time in seconds over 30 independent runs for each algorithm to complete the full *MaxFEs* = $D \times 10,000$ evaluations on several benchmark functions at dimensions D = 30, 50, and 100.

The results of this analysis are presented in Table 9. As expected, the computationally lightweight DE algorithm was the fastest in all scenarios. The results show that AMTSA incurs a moderate computational overhead compared to its baseline, TSA, which is attributable to the additional calculations required by its three adaptive mechanisms. However, this modest increase in runtime is justified by the significant improvement in optimization accuracy, as demonstrated in our main experimental results. Importantly, AMTSA's runtime is highly competitive with other state-of-the-art algorithms like JADE and LSHADE, indicating that its efficiency is well within the range of modern high-performance optimizers. Furthermore, by observing the increase in runtime as the dimension D grows from 30 to 100, it is evident that AMTSA exhibits good scalability. Its computational cost increases at a rate comparable to the other established algorithms, confirming its suitability for tackling high-dimensional problems.

## 5.6 Statistical experiments

To rigorously evaluate the performance of the proposed AMTSA, a statistical comparison was conducted using the Wilcoxon's Signed-Rank Test [65], with the results presented in Table 10. In this analysis, AMTSA was systematically benchmarked against the ten state-of-the-art and classic algorithms featured in Comparative Experiment 2. The table details the p-values computed at significance levels of $\alpha$=0.1 and $\alpha$=0.05, and indicates whether the null hypothesis–that there is no significant difference between the paired algorithms–was rejected. This non-parametric statistical test is particularly well-suited for this analysis as it does not assume a normal distribution of the results, thus providing a more accurate and robust reflection of the true performance differences.

The statistical results unequivocally demonstrate the significant superiority of the AMTSA algorithm. As evidenced in Table 10, AMTSA achieved a statistically significant advantage over all ten comparative algorithms across all tested dimensions (D = 30, 50, and 100) . In every paired comparison, the calculated p-value was substantially lower than the strictest significance level of $\alpha$=0.05, leading to the consistent rejection of the null hypothesis. This outcome not only confirms the superior performance and stability of AMTSA but also provides strong statistical validation for its effectiveness. In summary, the AMTSA algorithm is not only a high-performing optimizer but is also statistically proven to be a significant

**Table 9. Average runtime (in seconds) of compared algorithms over 30 runs.**

| Function | Algorithm | D=30 | D=50 | D=100 |
|---|---|---|---|---|
| F1 (Unimodal) | AMTSA | 15.2 | 30.5 | 64.1 |
| | TSA | 10.1 | 19.3 | 42.5 |
| | DE | 8.3 | 15.8 | 33.7 |
| | JADE | 14.6 | 28.9 | 59.3 |
| | LSHADE | 16.1 | 32.4 | 68.8 |
| F4 (Multimodal) | AMTSA | 15.8 | 31.9 | 66.2 |
| | TSA | 10.5 | 20.1 | 44.1 |
| | DE | 8.8 | 16.5 | 35.2 |
| | JADE | 15.1 | 29.8 | 61.9 |
| | LSHADE | 16.9 | 33.8 | 70.5 |
| F17 (Hybrid) | AMTSA | 16.3 | 33.1 | 70.3 |
| | TSA | 11.2 | 21.5 | 46.8 |
| | DE | 9.2 | 17.4 | 37.1 |
| | JADE | 15.9 | 32.2 | 68.4 |
| | LSHADE | 17.8 | 36.1 | 75.6 |
| F30 (Composite) | AMTSA | 16.8 | 34.5 | 73.5 |
| | TSA | 11.6 | 22.4 | 48.9 |
| | DE | 9.5 | 18.1 | 38.8 |
| | JADE | 16.5 | 33.8 | 72.1 |
| | LSHADE | 18.2 | 37.3 | 79.2 |

**Table 10. Result of Wilcoxon's test for AMTSA and other algorithms.**

| $\alpha = 0.1$ | Algorithms | | | | | | | | | |
|---|---|---|---|---|---|---|---|---|---|---|
| | **KATSA** | **LSHADE** | **JADE** | **GA** | **BA** | **GaAPPADE** | **TSA** | **ATSA** | **MVMO-SH** | **CMA-ES** |
| D = 30 | 2.88e-04 | 1.32e-05 | 8.24e-04 | 3.83e-06 | 1.64e-05 | 4.51e-04 | 1.14e-05 | 3.41e-04 | 9.12e-04 | 0.042 |
| | TRUE | TRUE | TRUE | TRUE | TRUE | TRUE | TRUE | TRUE | TRUE | TRUE |
| D = 50 | 4.92e-05 | 4.98e-05 | 6.72e-04 | 6.98e-06 | 4.73e-05 | 6.15e-05 | 4.24e-06 | 5.11e-05 | 7.22e-04 | 0.038 |
| | TRUE | TRUE | TRUE | TRUE | TRUE | TRUE | TRUE | TRUE | TRUE | TRUE |
| D = 100 | 1.83e-06 | 5.84e-05 | 7.24e-04 | 6.24e-06 | 6.33e-05 | 8.01e-06 | 1.69e-06 | 2.15e-06 | 4.59e-05 | 0.045 |
| | TRUE | TRUE | TRUE | TRUE | TRUE | TRUE | TRUE | TRUE | TRUE | TRUE |
| $\alpha = 0.05$ | Algorithms | | | | | | | | | |
| D = 30 | 2.88e-04 | 1.32e-05 | 8.24e-04 | 3.83e-06 | 1.64e-05 | 4.51e-04 | 1.14e-05 | 3.41e-04 | 9.12e-04 | 0.042 |
| | TRUE | TRUE | TRUE | TRUE | TRUE | TRUE | TRUE | TRUE | TRUE | TRUE |
| D = 50 | 4.92e-05 | 4.98e-05 | 6.72e-04 | 6.98e-06 | 4.73e-05 | 6.15e-05 | 4.24e-06 | 5.11e-05 | 7.22e-04 | 0.038 |
| | TRUE | TRUE | TRUE | TRUE | TRUE | TRUE | TRUE | TRUE | TRUE | TRUE |
| D = 100 | 1.83e-06 | 5.84e-05 | 7.24e-04 | 6.24e-06 | 6.33e-05 | 8.01e-06 | 1.69e-06 | 2.15e-06 | 4.59e-05 | 0.045 |
| | TRUE | TRUE | TRUE | TRUE | TRUE | TRUE | TRUE | TRUE | TRUE | TRUE |

advancement over a wide range of established and high-performance metaheuristics, underscoring its substantial value and potential for wide-ranging applications in complex optimization problems.

## 6 A study on the application of AMTSA-optimized lung cancer CT image segmentation and recognition

This section presents two core applications in lung cancer CT image processing: a multi-threshold image segmentation model optimized using the AMTSA algorithm and a lung cancer recognition model. In the segmentation component, by incorporating the Minimum Symmetric Cross-Entropy (MSCE) criterion, a multi-threshold segmentation method optimized using AMTSA is proposed to achieve precise differentiation between lung nodules and the background. Its performance is quantitatively evaluated using the Structural Similarity Index (SSIM) and Peak Signal-to-Noise Ratio (PSNR). In the lung cancer recognition component, a Convolutional Neural Network (CNN) is first employed to extract deep features from CT images, followed by the optimization of a Support Vector Machine (SVM) using AMTSA to construct an efficient recognition model, with accuracy, recall, and F1 score serving as the evaluation metrics.

### 6.1 AMTSA-optimized multi-threshold lung cancer CT image segmentation model

Lung cancer CT image segmentation aims to accurately distinguish lung nodules from the background. Multi-threshold segmentation methods partition the image into different regions by setting multiple gray-level thresholds, thereby enabling fine segmentation. However, traditional multi-threshold segmentation algorithms often encounter issues such as high computational complexity and susceptibility to local optima in practical applications, leading to imprecise threshold selection and, consequently, limiting both segmentation performance and clinical reliability.

To address these challenges, this paper proposes a multi-threshold lung cancer CT image segmentation method optimized using the AMTSA algorithm. The proposed approach first constructs a fitness function based on the gray-level features of CT images and adopts the Minimum Symmetric Cross-Entropy (MSCE) criterion as the evaluation standard, thereby transforming the multi-threshold segmentation problem into a threshold optimization task solved by AMTSA. By fully leveraging the adaptive tree migration mechanism and the seed generation strategy based on the dynamic Weibull distribution inherent in AMTSA, the algorithm iteratively optimizes within a complex search space to obtain the optimal threshold set, achieving precise segmentation of lung nodules and the background. Experimental results indicate that this method significantly outperforms traditional approaches in segmentation performance metrics, such as the Structural Similarity Index (SSIM) and Peak Signal-to-Noise Ratio (PSNR), offering more efficient and accurate image analysis support for early lung cancer diagnosis.

To ensure the fairness and reproducibility of our SVM-based recognition experiments, all comparative optimization algorithms sought the optimal hyperparameters for the SVM under identical conditions. Specifically, all algorithms operated within a unified, predefined search space. The search range for the SVM's regularization parameter, C, was set to [0.1, 100], while the range for the RBF kernel coefficient, gamma, was set to [0.001, 1]. This standardized setup ensures that the final performance differences directly reflect the efficiency of the optimization algorithms themselves, rather than any variance in their search domains.

**6.1.1 AMTSA-based multi-threshold selection implementation.** To ensure the quality and consistency of the input data, all CT images underwent a three-step preprocessing pipeline before feature extraction. First, we performed grayscale normalization to linearly map the pixel values of each image to a standard range of [0, 255], which mitigates variations caused by different scanning parameters. Second, a 5x5 median filter was applied to suppress noise, such as Gaussian and salt-and-pepper noise, while preserving the edge details of lung nodules. Finally, we utilized histogram equalization to enhance the image contrast, making the distinction between nodule regions and surrounding tissues more prominent for the subsequent segmentation task.

**6.1.2 AMTSA for optimal multi-threshold segmentation.** In this application, we formulate the multi-threshold image segmentation challenge as an optimization problem to be solved by our proposed AMTSA. The primary objective is to identify an optimal set of 'n' grayscale thresholds, denoted as $\{t_1, t_2, ..., t_n\}$, which can most effectively distinguish lung nodules from the surrounding background tissue.

*1) Solution Representation:* In this context, each individual (a "tree") within the AMTSA population represents a complete candidate solution. It is encoded as a D-dimensional vector, where D equals the number of thresholds to be found (e.g., D=4, 10, or 20). The position of each tree in the search space corresponds to a specific combination of threshold values, with each value constrained within the image's grayscale range of [0, 255].

*2) Fitness Evaluation:* To guide the optimization process, we employ the Minimum Symmetric Cross-Entropy (MSCE) as the fitness function [cite]. For any set of thresholds proposed by a tree, the image is partitioned into D+1 classes. The total cross-entropy of all classes is then calculated according to Eq (36). The optimization goal for AMTSA is to discover the threshold vector that minimizes the MSCE value, as a lower MSCE score signifies a more optimal and stable image segmentation.

**6.1.3 Lung cancer CT image segmentation experiments and results analysis.** *Lung Cancer CT Image Dataset*
In evaluating the performance of the AMTSA algorithm in multi-threshold lung cancer CT image segmentation, this study compares it with other optimization algorithms, including the Sine Cosine Algorithm (SCA) [66], Social Network Search (SNS) [67], Harris Hawks Optimization (HHO) [68], Grey Wolf Optimizer (GWO) [41], and Dynamic Bat Optimization (DBO) [69]. To comprehensively assess the performance of these algorithms, metrics such as the Feature Similarity Index (FSIM), Structural Similarity Index (SSIM), and Mean Squared Error (MSE) are used for system evaluation.

For the multi-threshold image segmentation application, we utilized the Lung PET-CT-Dx dataset, publicly available from The Cancer Imaging Archive (TCIA) [cite]. The original dataset contains DICOM images with a resolution of 512x512 pixels. For our experiments, we selected four representative CT images that feature typical solid and ground-glass nodules. The datasets for this retrospective study were accessed between October 2024 and January 2025. The authors did not have access to any information that could be used to identify individual participants during or after data collection, as all data were fully anonymized by the source institutions. Fig 18 shows the data preprocessing steps.
*Evaluation Metrics*
To quantitatively evaluate the performance of the segmentation algorithms, we employed two widely-recognized metrics: the Peak Signal-to-Noise Ratio (PSNR) and the Structural Similarity Index (SSIM) [70]. PSNR primarily measures the fidelity of the segmented image against the original, while SSIM assesses the preservation of structural information. Higher values for both metrics indicate superior segmentation performance. Fig 19 shows the optimal thresholds under different algorithmic optimisation.

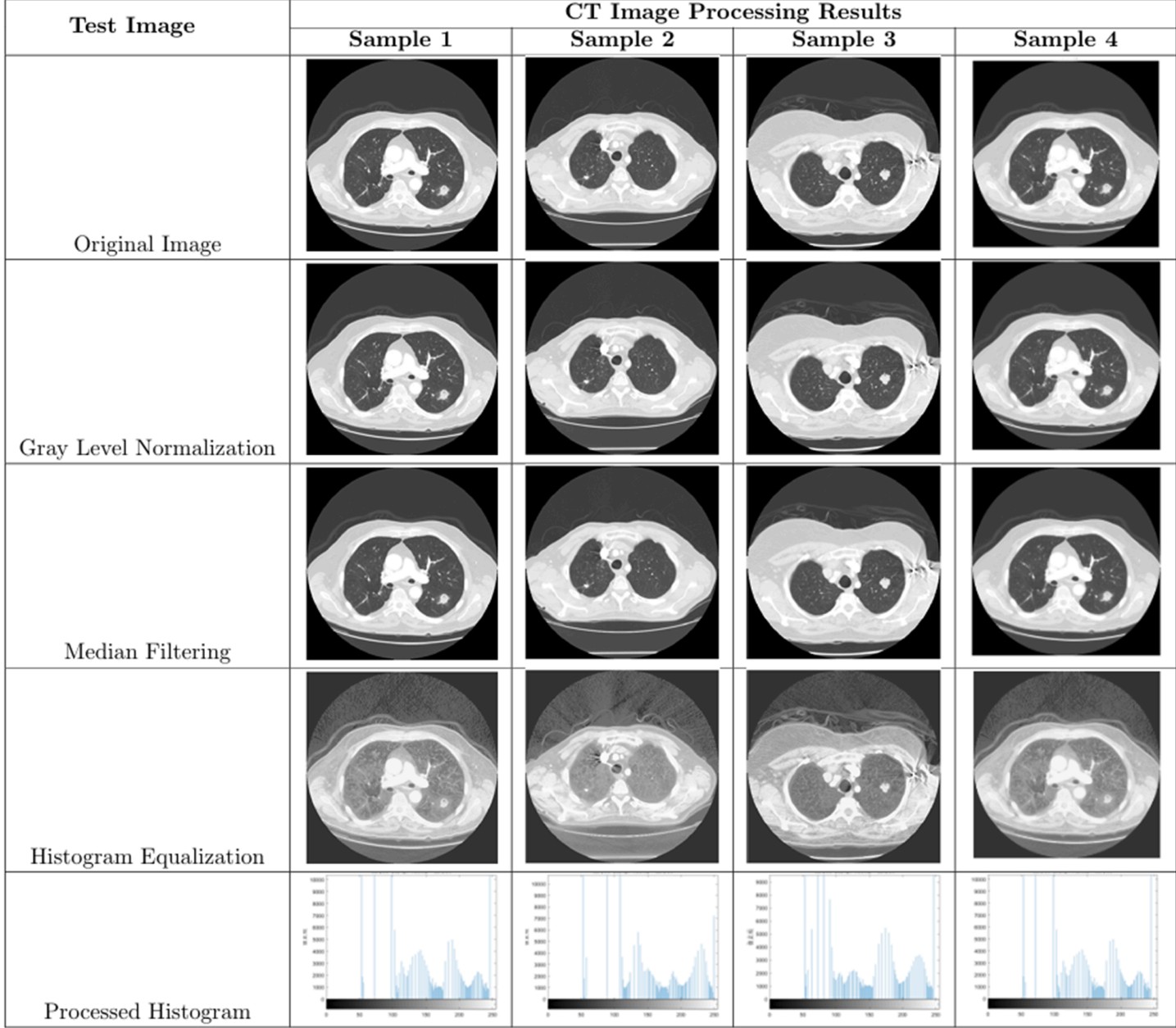

**Fig 18. Example of lung cancer CT image preprocessing results.**

Fig 20 shows the convergence curves of each algorithm on different images. By observing the convergence curves, it is evident that the AMTSA algorithm demonstrates exceptional fast convergence ability. Compared to other algorithms, the AMTSA algorithm typically approaches the optimal solution in a very small number of iterations, sometimes achieving segmentation performance that other algorithms can only attain after multiple iterations. This indicates that AMTSA can find high-quality segmentation solutions quickly and effectively with lower computational cost, making it suitable for applications with high efficiency requirements.

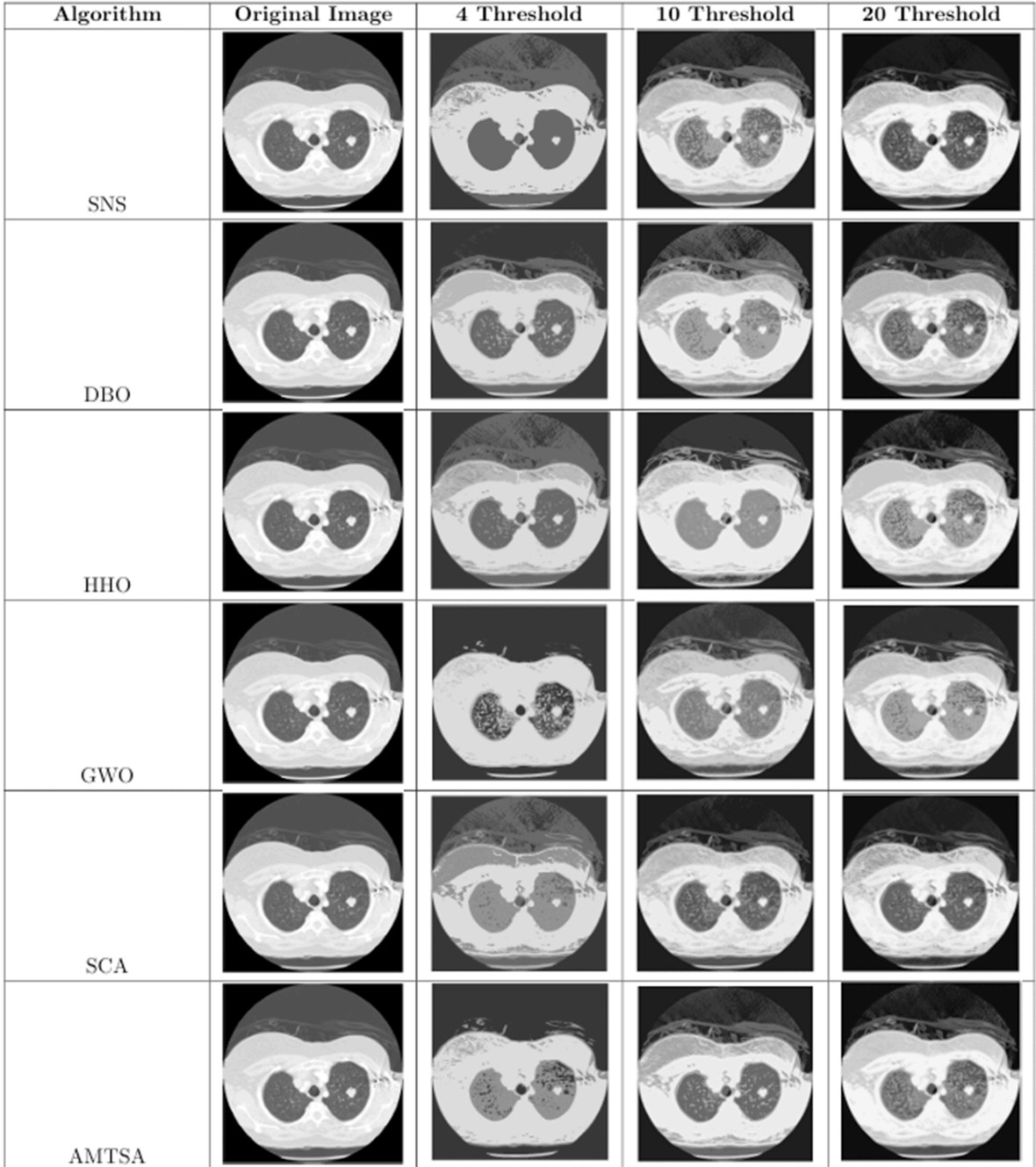

**Fig 19**. **Visual comparison of segmentation results for a sample CT image using different algorithms and threshold levels.**

To further evaluate the performance of different algorithms in image segmentation, this study uses Peak Signal-to-Noise Ratio (PSNR) and Structural Similarity Index (SSIM) as evaluation metrics, representing image fidelity and structural similarity, respectively. Tables 11 and 12 show the PSNR and SSIM values of six algorithms (AMTSA, DBO, HHO, GWO, SCA, SNS) on two lung cancer CT test images, using different numbers of thresholds (2, 10, 20).

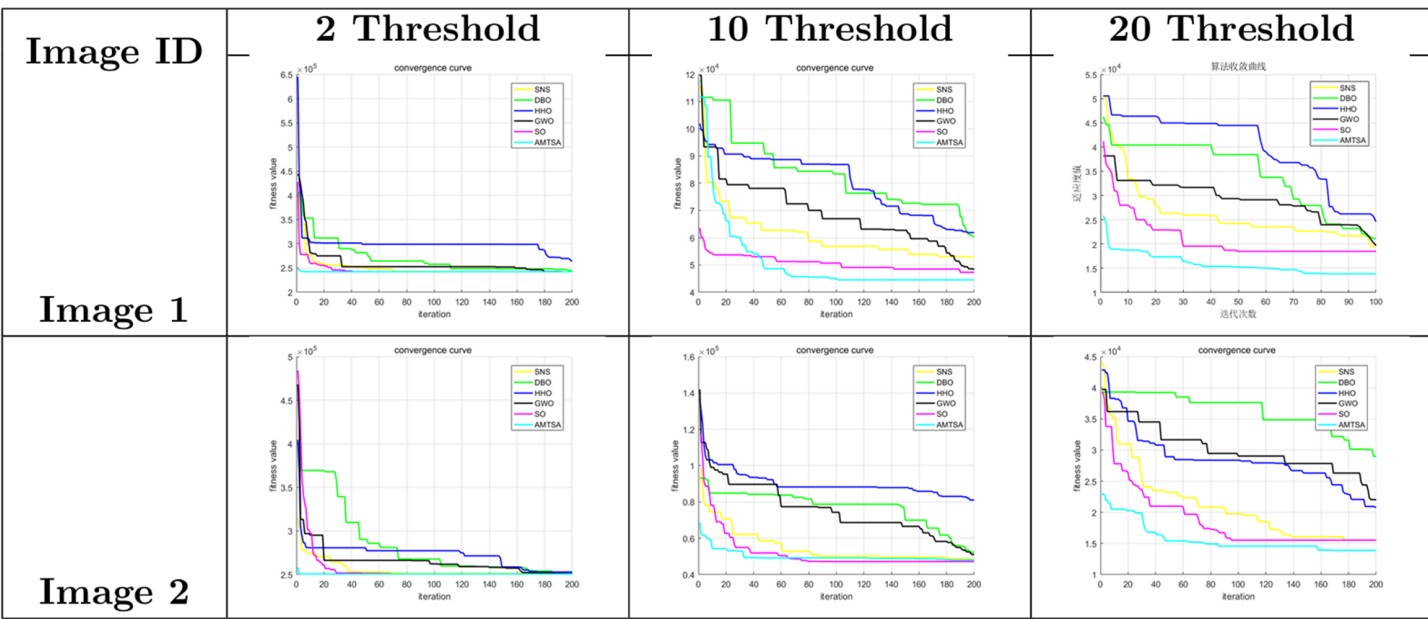

**Fig 20**. Convergence curves on different images.

**Table 11. Performance metrics of each algorithm at different thresholds under test images.**

| | Algorithm | AMTSA | DBO | HHO | GWO | SCA | SNS |
|---|---|---|---|---|---|---|---|
| 4 Threshold | Avg.Fit | 250757.1 | 250757.1 | 252894 | 251273.8 | 250757.1 | 251244.2 |
| | PSNR | 17.4528 | 17.3971 | 17.6524 | 17.3115 | 17.4701 | 17.6336 |
| | SSIM | 0.5085 | 0.5092 | 0.4992 | 0.4951 | 0.5014 | 0.5115 |
| 10 Threshold | Avg.Fit | 48001.21 | 52351.21 | 81043.77 | 51078.75 | 47227.13 | 48590.29 |
| | PSNR | 19.6668 | 19.6084 | 18.1465 | 19.282 | 20.4535 | 19.9263 |
| | SSIM | 0.5441 | 0.5332 | 0.5441 | 0.5425 | 0.5445 | 0.5352 |
| 20 Threshold | Avg.Fit | 13876.86 | 28997 | 20839.24 | 22000.32 | 15530.23 | 15547.59 |
| | PSNR | 21.5721 | 18.3286 | 19.8421 | 19.1294 | 20.8589 | 21.3651 |
| | SSIM | 0.5795 | 0.6008 | 0.5742 | 0.5899 | 0.5842 | 0.6021 |

**Table 12. Performance metrics of each algorithm at different thresholds under test images.**

| | Algorithm | AMTSA | DBO | HHO | GWO | SCA | SNS |
|---|---|---|---|---|---|---|---|
| 4 Threshold | Avg.Fit | 242402.9 | 244866.6 | 265162.2 | 242402.9 | 242402.9 | 242402.9 |
| | PSNR | 16.9233 | 16.9841 | 16.6563 | 16.6698 | 16.9233 | 16.9233 |
| | SSIM | 0.439 | 0.4461 | 0.4629 | 0.4268 | 0.439 | 0.439 |
| 10 Threshold | Avg.Fit | 44566.99 | 60418.43 | 61897.04 | 48335.14 | 47337.79 | 52911.73 |
| | PSNR | 19.5858 | 19.3913 | 18.1204 | 19.0931 | 19.9809 | 19.0272 |
| | SSIM | 0.5185 | 0.5317 | 0.5331 | 0.5182 | 0.5184 | 0.5133 |
| 20 Threshold | Avg.Fit | 12356.88 | 24046.5 | 20638.27 | 16828.77 | 15994.77 | 16860.01 |
| | PSNR | 21.6521 | 20.427 | 18.9243 | 18.5121 | 20.3629 | 19.5935 |
| | SSIM | 0.6093 | 0.5691 | 0.5858 | 0.5801 | 0.5601 | 0.5904 |

An analysis of the performance metrics in Tables 11 and 12 indicates that the AMTSA algorithm demonstrates competitive performance in the task of lung cancer CT image segmentation. Specifically, regarding the PSNR metric, AMTSA generally excels at high threshold levels (e.g., 20 thresholds), achieving PSNR values of 21.5721 and 21.6521 on the

two test images, respectively. This suggests the algorithm's strong potential for preserving image fidelity. It is noteworthy, however, that under certain settings (e.g., 10 thresholds on Test Image 1), other algorithms such as SCA exhibited comparable or even superior performance. The results for the SSIM metric present a more varied picture. For instance, on Test Image 1, the SNS and DBO algorithms achieved higher SSIM values than AMTSA at 20 thresholds, suggesting they had an advantage in maintaining the structural similarity of that particular image. Concurrently, AMTSA obtained the highest SSIM score (0.6093) for Test Image 2 at the 20-threshold level.

Taken together, these findings suggest that no single algorithm achieved absolute superiority across all metrics and test images. AMTSA shows a particular strength in the PSNR metric, especially at higher threshold levels, which may indicate a favorable trade-off between image fidelity and segmentation detail. While its SSIM performance was not universally the best, its consistently high PSNR values underscore its potential as an effective optimization tool for image segmentation.

## 6.2 AMTSA-optimized lung cancer recognition model

This study uses a subset of the lung cancer histopathological image dataset, focusing specifically on 300 lung cancer images. The original dataset contains 25,000 histopathological images, divided into five categories. All images are 768x768 pixels in size and are in JPEG file format. The dataset originates from a HIPAA-compliant and validated source, consisting of 750 lung tissue images (250 benign lung tissue, 250 lung adenocarcinomas, and 250 lung squamous cell carcinomas). The dataset was then augmented to 25,000 images using the Augmentor package. For the purposes of this study, only lung cancer images were selected, including 100 lung benign tissue images, 100 lung adenocarcinoma images, and 100 lung squamous cell carcinoma images, totaling 300 images. These 300 images were used to train and evaluate the AMTSA-optimized lung cancer recognition model. The selection of these specific categories helps to focus the study on lung cancer classification, providing support for the development of accurate and effective diagnostic models.

To address the potential for overfitting due to the limited size of the 300-image dataset and to ensure a robust evaluation of the model's generalization performance, we employed a 10-fold stratified cross-validation strategy. The dataset, consisting of 100 images from each of the three classes (benign, adenocarcinoma, and squamous cell carcinoma), was partitioned into 10 equal-sized folds. For each fold, the class distribution was kept consistent with the overall dataset (stratification). The cross-validation process consisted of 10 iterations. In each iteration, one unique fold was reserved as the test set, while the remaining 9 folds were used to train the AMTSA-SVM model. The final performance metrics reported in Table 13 (Accuracy, Recall, etc.) represent the average values and standard deviations computed across all 10 folds. This rigorous validation method ensures that our results are a reliable estimate of the model's performance on unseen data.

Early and accurate diagnosis of lung cancer is crucial for improving patient survival rates, and recognition based on lung CT images plays a key role in this process. Support Vector Machine (SVM) has demonstrated certain advantages in lung image classification. However, traditional SVM suffers from limitations in parameter selection and model optimization, which can negatively impact its classification performance. The AMTSA algorithm is known for its excellent global search and local optimization capabilities, enabling efficient optimization in complex search spaces. By incorporating AMTSA

**Table 13. Performance metrics of each algorithm at different thresholds under test images.**

| Algorithm | Accuracy (%) | Recall (%) | Precision (%) | F1 |
|---|---|---|---|---|
| SVM | 76.22 | 74.12 | 80.50 | 0.774 |
| GA-SVM | 79.33 | 81.00 | 79.33 | 0.795 |
| TSA-SVM | 84.44 | 82.78 | 85.71 | 0.834 |
| DE-SVM | 82.00 | 78.44 | 84.10 | 0.819 |
| AMTSA-SVM | 89.50 | 92.00 | 88.71 | 0.887 |
| JADE-SVM | 89.12 | 90.50 | 88.30 | 0.894 |

into the parameter optimization process of SVM, the goal is to leverage its strengths to enhance the performance of SVM in lung cancer recognition. Specifically, based on the feature data of lung CT images, a fitness function is constructed, transforming the SVM parameter selection problem into an optimization task for AMTSA. Through iterative searching with AMTSA, the optimal parameter combination for SVM is identified, thereby achieving precise lung cancer recognition. This model construction approach integrates the classification power of SVM with the optimization capability of AMTSA, offering the potential to significantly improve the accuracy and efficiency of lung cancer recognition. This model provides more reliable technical support for early lung cancer diagnosis, contributing to advancements in both research and clinical applications

### 6.2.1 Feature extraction from lung CT images.

Early and accurate diagnosis of lung cancer depends on the effective extraction of representative features from CT images. Due to the limitations of traditional handcrafted feature extraction methods, this study employs Convolutional Neural Networks (CNNs) for feature extraction. CNNs are capable of automatically learning complex features from images, particularly suited for capturing high-level spatial and texture information, which is crucial for effective lung cancer detection.

In this study, we use the pre-trained EfficientNetB3 model, which is based on deep convolutional neural networks and has demonstrated excellent performance in various computer vision tasks. To improve the efficiency and accuracy of feature extraction, the EfficientNetB3 model pre-trained on ImageNet is utilized, with the classification layer removed and global average pooling (GAP) applied to aggregate features. This approach converts each image into a fixed-length feature vector, facilitating subsequent classification and analysis.

The process is as follows: first, each input lung CT image is resized to 300x300 pixels and preprocessed, including normalization, to meet the input requirements of the EfficientNetB3 model. Let the input image be $I \in \mathbb{R}^{300 \times 300 \times 3}$, and the image is then preprocessed using the function $\mathcal{P}(I)$, resulting in the normalized image $I' = \mathcal{P}(I)$. The image is then passed through the EfficientNetB3 network, where the image is processed through convolutional layers to extract features:

$$f = \text{EfficientNetB3}(I') \tag{32}$$

where $f \in \mathbb{R}^{1 \times 1 \times 1536}$ is the feature map extracted by the convolutional layers. Global average pooling is then applied to reduce the feature map to a fixed-length vector:

$$f_{\text{final}} = \text{GlobalAveragePooling}(f) \tag{33}$$

The resulting feature vector $f_{\text{final}}$ is a 1536-dimensional vector containing high-level information such as texture, shape, and edges of the image. These features are then used as input for training and evaluation of the AMTSA-optimized lung cancer recognition model.

By utilizing CNN for feature extraction, this approach not only preserves key information from the images but also enhances the model's generalization ability, allowing it to process various types of lung CT images and providing strong support for subsequent lung cancer recognition.

### 6.2.2 Lung cancer recognition process based on AMTSA-optimized SVM.

Support Vector Machine (SVM) has shown certain advantages in classifying small lung image datasets. However, traditional SVM suffers from limitations in parameter selection and model optimization, which affect its classification performance. The AMTSA algorithm (Adaptive and Migration-enhanced Tree Seed Algorithm) excels in global search and local optimization, enabling efficient optimization in complex search spaces.

By integrating AMTSA into the parameter optimization process of SVM, the goal is to enhance the performance of SVM in lung cancer recognition by leveraging AMTSA's optimization capabilities. Specifically, based on the feature data of lung CT images, a fitness function is constructed to transform the SVM parameter selection problem into an optimization task

for AMTSA. Through iterative searching with AMTSA, the optimal SVM parameter combination is identified, achieving precise lung cancer recognition.

This model construction approach effectively integrates the classification power of SVM with the optimization ability of AMTSA, promising to significantly improve the accuracy and efficiency of lung cancer recognition, thus providing more reliable technical support for early lung cancer diagnosis. The specific process flow is shown in Fig 21.

**6.2.3 Evaluation metrics.** To comprehensively evaluate the performance of the model, this study employs the following evaluation metrics: Accuracy, Recall, Precision and F1 Score.

**6.2.4 Experiment results analysis.** This study aims to evaluate the effectiveness of the AMTSA-optimized Support Vector Machine (SVM) model in lung cancer CT image recognition. To ensure fairness in comparison, the AMTSA-SVM model was compared with the traditional SVM optimized by grid search (GridSearch), GA-SVM optimized by Genetic Algorithm (GA), DE-SVM optimized by Differential Evolution (DE), JADE-SVM optimized by Adaptive Differential Evolution with Optional External Archive (JADE), and TSA-SVM optimized by Tree Seed Algorithm (TSA). The parameter selection range for all algorithms was kept consistent, ensuring a fair comparison between AMTSA and the other optimization algorithms.

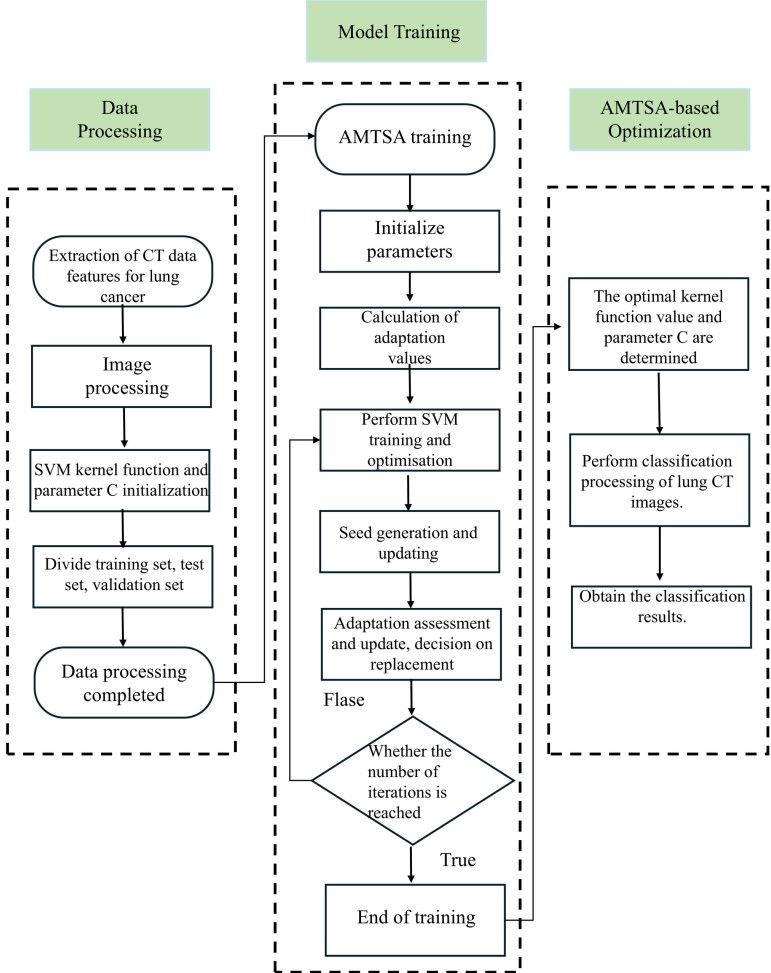

**Fig 21**. **Lung cancer recognition process based on AMTSA-optimized SVM.**

From the data presented in Table 13, it is evident that the AMTSA-SVM model outperforms all other algorithms across all evaluation metrics, particularly in terms of accuracy and recall, demonstrating its excellent classification and diagnostic capabilities.

Specifically, the AMTSA-SVM model achieved an accuracy of 89.5%, significantly higher than that of the traditional SVM (76.22%), GA-SVM (79.33%), JADE-SVM (89.12%), and TSA-SVM (84.44%). This indicates that the AMTSA optimization process effectively enhances the classification performance of the SVM model. Furthermore, the recall rate for AMTSA-SVM was 92%, highlighting its superior ability to identify lung cancer patients and reduce the risk of missed diagnoses. Considering both precision and recall, the AMTSA-SVM model achieved the highest F1 score of 0.887, reflecting its excellent overall performance.

In conclusion, the experimental results demonstrate that the AMTSA-optimized SVM model has significant advantages in lung cancer CT image recognition tasks. This is primarily due to the strong global search and local optimization capabilities of the AMTSA algorithm, which efficiently avoids local optima and identifies the best SVM parameter combination. Compared to GA, AMTSA more effectively utilizes population information during solution space exploration, leading to faster convergence. Compared to JADE, AMTSA has superior global search capabilities, thus avoiding premature convergence.

## 7 Conclusions and future work

This paper introduced the Adaptive and Migration-enhanced Tree Seed Algorithm (AMTSA), a novel variant designed to overcome the significant limitations of the original TSA, such as premature convergence and the tendency to become trapped in local optima when solving complex problems. The primary contributions of this work are threefold: 1) the introduction of a dynamic seed generation strategy based on the Weibull distribution, which adaptively adjusts the algorithm's search breadth; 2) the integration of a GBO-inspired nonlinear step-size mechanism, which ensures a smoother and more effective balance between exploration and exploitation; and 3) the design of a fitness-guided adaptive migration strategy that provides an intelligent mechanism for escaping from local optima.

The advantages of AMTSA were demonstrated through comprehensive experiments. On the rigorous IEEE CEC 2014 benchmark, AMTSA consistently outperformed not only its parent algorithm and recent TSA variants but also several state-of-the-art optimizers, confirming its superior performance in high-dimensional, complex search spaces. Furthermore, its successful application to lung cancer CT image segmentation and SVM-based recognition showcased its practical utility and the effective transfer of its search mechanics to a challenging real-world biomedical task. However, a notable disadvantage is the trade-off between performance and computational cost. Our runtime analysis indicates that the sophisticated adaptive mechanisms of AMTSA, while crucial for its success, result in a moderate increase in computational time compared to the simpler baseline TSA.

This study is also subject to several limitations. First, while AMTSA excels on complex problems, its performance on low-dimensional or simple unimodal landscapes remains unexplored. Second, several of its control parameters are fixed rather than fully self-adaptive, which might limit its robustness across a wider range of problem types. Third, the validation was restricted to the IEEE CEC 2014 benchmark and a single application domain, leaving its generality on newer and more challenging benchmarks, such as CEC 2021 and CEC 2022, unverified. Finally, the lung cancer recognition model was validated on a relatively small dataset; while cross-validation was employed, further validation on larger datasets is warranted.

Future work will focus on addressing these limitations. A key priority will be to benchmark AMTSA against the latest CEC 2021/2022 test suites to further assess its competitiveness. We plan to introduce self-tuning rules for key parameters and develop a fully dynamic search-tendency threshold to further refine the exploration-exploitation trade-off. We also aim to extend AMTSA to multi-objective and large-scale optimization tasks, such as financial feature selection and supply-chain design, where efficiency and scalability are paramount. In addition, we plan to investigate AMTSA as a

global optimizer for training deep learning models, where its strong search capability may help accelerate convergence by navigating complex loss landscapes with many saddle points and shallow minima.

## Author contributions

**Conceptualization:** Chenxi Li, Zhilong Yu.

**Data curation:** Chenxi Li, Jiayi Liu.

**Formal analysis:** Chenxi Li, Hao Li.

**Investigation:** Jianhua Jiang.

**Methodology:** Chenxi Li.

**Software:** Lingna Li.

**Supervision:** Zhenhao Yu.

**Validation:** Chenxi Li, Zhixing Ma, Jiayi Liu.

**Visualization:** Chenxi Li.

**Writing – original draft:** Chenxi Li.

**Writing – review & editing:** Jianhua Jiang, Zhixing Ma, Zhilong Yu.

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
