## [Decision Letter · Decision Letter 0]

14 Jul 2025

PONE-D-25-33941Adaptive and Migration-enhanced Tree Seed Algorithm for Multi-Threshold CT Image Segmentation and Lung Cancer RecognitionPLOS ONE

Dear Dr. Li,

Thank you for submitting your manuscript to PLOS ONE. After careful consideration, we feel that it has merit but does not fully meet PLOS ONE’s publication criteria as it currently stands. Therefore, we invite you to submit a revised version of the manuscript that addresses the points raised during the review process. Please address each comment raised by the Reviewers before resubmitting the manuscript.

We look forward to receiving your revised manuscript.

Kind regards,

Mahamed G.H. Omran

Academic Editor

PLOS ONE

Journal Requirements:

2. Please ensure that you refer to Figure 1, 2, 3 and 8 in your text as, if accepted, production will need this reference to link the reader to the figure.

3. We note you have included a table to which you do not refer in the text of your manuscript. Please ensure that you refer to Table 9 in your text; if accepted, production will need this reference to link the reader to the Table.

Reviewers' comments:

Reviewer's Responses to Questions

**Comments to the Author**

1. Is the manuscript technically sound, and do the data support the conclusions?

Reviewer #1: Partly

Reviewer #2: Yes

2. Has the statistical analysis been performed appropriately and rigorously?

Reviewer #1: Yes

Reviewer #2: Yes

3. Have the authors made all data underlying the findings in their manuscript fully available?

Reviewer #1: No

Reviewer #2: Yes

4. Is the manuscript presented in an intelligible fashion and written in standard English?

Reviewer #1: Yes

Reviewer #2: Yes

5. Review Comments to the Author

Reviewer #1: Trying to improve such a method like TSA, which is not that good, why not.

But the result is a complicated algorithm, with many ad hoc formulae and hard-coded parameters without any sensitivity analysis.

Also some comparisons are not fair.

The paper is far too long. It could be (and IMO should be) more concise, at least by removing redundancies and useless comparisons.

Also not all figures are really needed. You could select just a few representative ones.

You should clearly point out in which way this paper is _significantly_ different from the other TSA variants ([26], [34], ...) by partly the same authors and what is really new in it.

Last but not least you write

"The source code for the AMTSA algorithm is publicly available at www.jianhuajiang.com"

but I did'nt find it (only codes of some other TSA variants) so I have not been able to reproduce your experiments and to perform more comparisons.

line 20

"the No Free Lunch Theorem states that no optimization algorithm can perform optimally in all contexts [14]."

[14] is

Panos M Pardalos, Varvara Rasskazova, Michael N Vrahatis, et al. Black box optimization, machine learning, and no-free lunch theorems. Springer, 2021.

It would better to cite the original paper

Wolpert, David H., and William G. Macready. 1997. “No Free Lunch Theorems for Optimization.” IEEE Transactions on Evolutionary Computation 1 (1): 67–82.

I don't have [14] at hand, but if this paper really claims what you say, this is too vague.

A more correct claim is that the NFLT states that all algorithms are equivalent (in average) when considering all possible functions on a given search space to a given value space (i.e. a set of functions closed under permutations).

In practice a benchmark is never c.u.p. so there may exist a best algorithm.

line 48...

Motivations and Contribution are partly redundant. You should be more concise, at least by removing from Motivations sentences like "By redesigning the

seed generation process,..." which are in fact contributions.

line 83 and many other (203, 223, 276 ...)

"as its randomness hinders the balance between exploration and exploitation"

Please, rigorously define what exploration and exploitation are (I mean say how you _measure_ them). And what a "good balance" is (fifty/fifty?)

And then prove, at least experimentally, that this two measures can indeed be balanced thanks to your approach.

As you define a diversity measure (lines 413...) you could (should) do that on more multimodal test functions than just F8 (p. 19, fig. 10, 12))

line 260 (formula 16)

Why 10? Sensitivity analysis?

lambda: sensitivity analysis?

line 299 (formulae 18,19)

Several hardcoded parameters like 0.2 etc. Sensitivity analysis?

And why (1.2-0.2) and not just 1?

lines 340-354

Redundant.

line 533... Comparative Experiment 1: AMTSA versus EST-TSA, MTSA, TSA, STSA, fb-TSA

Not very useful. As TSA is anyway not a good algorithm challenging it is easy, and you could/should compare to just the best ones of these variants.

And I wonder why you don't compare to ATSA [26] or KATSA [34] which have partly the same authors. Are they in fact not as good as claimed?

line 569...

"Several classical heuristic optimization algorithms (GA [56], BA [57], GWO [41]) as well as some novel algorithms (DE [52, 53], JADE [55] and LSHADE [54])

are selected for comparative analysis"

As you use the CEC 2014 benchmark you have to compare at least to the two or three first winners of this competition.

Not only L-SHADE (but with the right population size, which is decreasing), but also GaAPPADE and maybe MVMO-SH.

I also suggest CMA-ES.

Actually, for L-SHADE and possibly for the other methods, I am not sure you respect the conditions to correctly use the CEC 2014 benchmark, for fair comparison (population size, number of evaluations, parameters).

Please carefully check [54], in particular the Algorithm Parameters section and the results in Table I.

About GWO you should read

Camacho-Villalón, Christian L., Marco Dorigo, and Thomas Stützle. ‘Exposing the Grey Wolf, Moth-Flame, Whale, Firefly, Bat, and Antlion Algorithms: Six Misleading Optimization Techniques Inspired by Bestial Metaphors’. International Transactions in Operational Research, July 2022. https://doi.org/10.1111/itor.13176

And more generally you should consider

Sörensen, K. (2015). Metaheuristics—the metaphor exposed. International Transactions in

Operational Research, 22(1), 3-18.

in which the authors write

"several journals such as the Journal of Heuristics (Journal of Heuristics 2015), Swarm Intelligence (Dorigo 2016), and the ACM Transactions on Evolutionary Learning and Optimization (ACM 2021) have already done—and add a statement to the following effect to their submission guidelines:

This journal will not publish papers that propose “novel” metaphor-based metaheuristics, unless the authors

(i) present their method using the normal, standard optimization terminology;

(ii) show that the new method brings useful and novel concepts to the field;

(iii) motivate the use of the metaphor on a sound, scientific basis; and

(iv) present a fair comparison with other state-of-the-art methods using state-of-the-art practices for benchmarking algorithms.

particularly the last three points.

line 575

"we conducted 30 rounds of experiments, each containing 200 iterations,"

The stop criterion must be a maximum number of fitness evaluations (the search effort), not a number of iterations.

This is particularly important for algorithms like L-SHADE whose population size is variable during the run.

But it seems you use the same population size for all algorithms (and which one?)

And using the same population size for different algorithms is usually not a fair practice. Some algorithms are designed to perform well with smaller populations and more iterations, while others benefit from larger populations and fewer iterations, even when the total number of fitness evaluations is held constant.

For each method, you should at least use the population size suggested by the author(s) or automatically estimated by the algorithm (sometimes dynamically, i.e. a variable population size).

line 748... Table 13

"Although AMTSA does not always perform the best in terms of SSIM, its advantage in PSNR demonstrates its potential to preserve the original image information effectively."

A bit optimistic.

line 866... Table 14

Although the results seem interesting there is no way to evaluate the fairness of the comparisons.

Please specify all parameters values (including population size) and the versions of the other algorithms (references?)

Typos

-----

weibull => Weibull (several times)

Reviewer #2: I have the following comments:

- The proposed algorithm should be evaluated using one of the recent CEC benchmark sets, such as CEC 2021 or CEC 2022.

- Conduct ablation analysis for your algorithm.

- Include empirical runtime comparisons (e.g., AMTSA vs. DE or JADE) to assess practical scalability.

- The adverb "where" after an equation should be written with small letters.

- It is essential to compare your algorithm experimentally or through discussion with efficient optimization algorithms such as iCSPM, iCSPM2, Exploratory Cuckoo Search, and Improved SSA (ISSA) with HDPM

- The lung cancer dataset (300 images) seems small. Clarify if cross-validation was used to mitigate overfitting.

- Include visual examples of segmented CT images (e.g., with 4/10/20 thresholds) to qualitatively validate improvements.

- The fixed parameter ST=0.1 is used across all TSA variants. Justify why this value is optimal or discuss sensitivity analysis.

- What are the advantages and disadvantages of your method over existing methods?

- The contributions of the authors should be described well in the conclusion section.

- The limitations of the proposed approach should be mentioned in the conclusion section.

6. PLOS authors have the option to publish the peer review history of their article (what does this mean?). If published, this will include your full peer review and any attached files.

Reviewer #1: No

Reviewer #2: No

---

## [Author Response · Author response to Decision Letter 1]

15 Aug 2025

Response to Reviewers

Chenxi Li

Center for Artificial Intelligence,

Jilin University of Finance and Economics,

Changchun 130117, China

Email: 0314042200902@jlufe.edu.cn

August 15, 2025

To: The Academic Editor and Reviewers, PLOS ONE

Manuscript ID: PONE-D-25-33941

Manuscript Title: Adaptive and Migration-enhanced Tree Seed Algorithm for Multi-Threshold CT Image Segmentation and Lung Cancer Recognition

Dear Dr. Omran and esteemed Reviewers,

Thank you for your thorough review of our manuscript and for providing insightful comments and constructive suggestions. We appreciate the time and effort you have dedicated to our work. We have carefully considered all the points raised and have revised the manuscript accordingly. We believe the manuscript has been significantly improved as a result of these revisions.

Below is our point-by-point response to the comments. All changes have been highlighted in the manuscript file labeled 'Revised Manuscript with Track Changes'.

Responses to the Academic Editor

Journal Requirement 1: Please ensure that your manuscript meets PLOS ONE's style requirements, including those for file naming.

Our Response: Thank you for this reminder. We have carefully reviewed the PLOS ONE style templates and have revised the manuscript to ensure full compliance with all formatting and file naming requirements.

Journal Requirement 2: Please ensure that you refer to Figure 1, 2, 3 and 8 in your text...

Our Response: Thank you for pointing this out. We have checked the manuscript and confirmed that all figures, including Figures 1, 2, 3, and 8, are now correctly referenced in the main text.

Journal Requirement 3: Please ensure that you refer to Table 9 in your text...

Our Response: Thank you. We have added a reference to Table 9 in the main text to ensure it is properly linked.

Journal Requirement 4: If the reviewer comments include a recommendation to cite specific previously published works, please review and evaluate these publications... Our Response: We have carefully reviewed all publications recommended by the reviewers. We have cited those that are relevant to our work, including the seminal paper on the No Free Lunch theorem by Wolpert and Macready, and the critiques on metaheuristics by Sörensen and Camacho-Villalón et al., to enrich our discussion.

Responses to Reviewer #1

Our Response: We sincerely thank you for this comprehensive and insightful feedback, which has been invaluable in improving the quality of our manuscript. We agree that a rigorous evaluation requires fair comparisons, transparent parameterization, and a clear justification of novelty. In response to your comments, we have undertaken a major revision of the manuscript.

1. Conciseness and Structure: We have streamlined the paper. To improve clarity and focus, we have been more selective with our figures (e.g., removing unnecessary stability analysis).

2. Parameter Sensitivity Analysis: We agree that the use of hard-coded parameters requires justification. We have now added a new dedicated subsection, Section 5.3, "Parameter Sensitivity Analysis", to investigate the impact of AMTSA's key parameters. This analysis provides an empirical basis for the parameter values used in our study (see Page 32).

3. Fairness of Comparisons: This was a critical point, and we have completely overhauled our experimental protocol to ensure fairness.

(1) The stopping criterion for all algorithms has been uniformly set to a maximum number of fitness evaluations (MaxFEs), as is now explicitly stated in Section 4.1, "Experimental Fundamentals" (Page 14).

(2) We have adopted algorithm-specific parameter settings, including population sizes, based on recommendations from the original literature. These are now transparently detailed in the revised Table 1 (Page 16).

(3) The comparative algorithm suite has been updated to include top-performing CEC competitors as you suggested.

4. Novelty and Distinction: We have revised our Contribution section to address your concerns about novelty (Page 3), detailing the significant methodological differences and unique contributions of AMTSA compared to our prior works.

5. Source Code Availability: We sincerely apologize for the issue with the source code link. To ensure reproducibility, the complete source code will be provided as supplementary material with our submission. The public link will become active once the paper is successfully published.

We are confident that these substantial revisions have addressed your concerns. We will now respond to each of your specific points in detail below.

(Specific Comments)

Point 1: NFLT Citation Reviewer's Comment: "line 20... It would better to cite the original paper Wolpert, David H., and William G. Macready. 1997..."

Our Response: Thank you for this excellent suggestion. We agree that citing the original paper is more appropriate. We have now replaced the previous reference with the seminal paper by Wolpert and Macready (1997), as shown in the revised manuscript on Page 1, Line 19.

Point 2: Redundancy in Motivations/Contribution Reviewer's Comment: "line 48... Motivations and Contribution are partly redundant. You should be more concise..."

Our Response: Thank you for your comment. We have reviewed these sections and agree that there was overlap. We have now revised and condensed the "Motivations" and "Contribution" sections to eliminate redundancy and present our arguments more concisely and forcefully, as can be seen in the revised manuscript (Pages 2-3).

Point 3: Defining Exploration/Exploitation Reviewer's Comment: "line 83... Please, rigorously define what exploration and exploitation are... And then prove, at least experimentally, that this two measures can indeed be balanced thanks to your approach."

Our Response: Thank you for highlighting this crucial point. To address this, we have made two significant revisions. First, under Section 5, "Analysis and discussion," we have improved the subsection "Exploration and Exploitation Analysis" (Page 21), where we rigorously define how we quantify exploration and exploitation based on population diversity, a widely accepted method. Second, we have expanded our experimental analysis to include a diverse set of representative functions (F1, F8, F10, and F25). The results, presented in the new Figure 11 (Page 22), now clearly demonstrate how AMTSA's mechanisms achieve a more effective and adaptive balance.

Point 4: Hard-coded parameters and Sensitivity Analysis (formula 16 & 18, 19) Reviewer's Comment: "line 260 (formula 16) Why 10? Sensitivity analysis? lambda: sensitivity analysis? ... line 299 (formulae 18,19) Several hardcoded parameters like 0.2 etc. Sensitivity analysis? And why (1.2-0.2) and not just 1?"

Our Response: Thank you for pointing out the need for justification of these parameters. We have now added a new dedicated subsection, Section 5.3, "Parameter Sensitivity Analysis" (Page 32), which includes a thorough analysis of the key parameters, including the divisor in formula (16) and the range parameters in formula (18). The results of this analysis provide an empirical basis for the values chosen in our study. Regarding the notation (1.2-0.2), we have used this standard formulation to clearly indicate the defined range of the parameter beta, which varies from a maximum of 1.2 to a minimum of 0.2.

Point 5: Comparison with other TSA Variants Reviewer's Comment: "line 533... Comparative Experiment 1: ... Not very useful. As TSA is anyway not a good algorithm challenging it is easy, and you could/should compare to just the best ones of these variants. And I wonder why you don't compare to ATSA [26] or KATSA [34] which have partly the same authors."

Our Response: Thank you for this valuable feedback. To address your concern, we have revised our comparison strategy. In our first comparative experiment (Section 5.2.1), we have now included our prior works, ATSA and KATSA, for a direct and transparent comparison. This allows us to situate AMTSA within our research trajectory and clearly demonstrate its unique advancements. We have retained the comparison with other TSA variants to provide a comprehensive picture for readers familiar with the TSA literature, but the main focus is now on the comparison with state-of-the-art algorithms in the second experiment.

Point 6: Fairness of Comparison with State-of-the-Art Algorithms Reviewer's Comment: "line 569... As you use the CEC 2014 benchmark you have to compare at least to the two or three first winners of this competition. Not only L-SHADE... but also GaAPPADE and maybe MVMO-SH. I also suggest CMA-ES. ... I am not sure you respect the conditions to correctly use the CEC 2014 benchmark, for fair comparison..." Our Response: We sincerely thank you for these critical comments, which have led to a complete overhaul of our experimental protocol to ensure fairness and rigor.

1. We have updated our comparative algorithm suite in Section 5.2.2 to include the highly competitive algorithms you suggested: GaAPPADE, MVMO-SH, and CMA-ES.

2. We have strictly adhered to the CEC 2014 guidelines. The stopping criterion for all algorithms is now uniformly set to MaxFEs = D x 10,000, as stated in Section 4.1, "Experimental Fundamentals".

3. We have adopted algorithm-specific parameter settings, including dynamic population sizing for L-SHADE, based on recommendations from their original publications. All parameters are now transparently detailed in the revised Table 1 (Page 16).

Point 7: Critique of Metaheuristic Metaphors Reviewer's Comment: "About GWO you should read... And more generally you should consider Sörensen, K. (2015)..."

Our Response: Thank you for directing us to this important body of literature. We have read the recommended papers and agree with the call for more scientific rigor. We have added a new paragraph in the Contribution section of the Introduction (Page 3), citing these works. In this new paragraph, we explicitly frame the contribution of AMTSA not on its natural metaphor, but on its concrete, quantifiable, and novel computational mechanisms.

Point 8: Tone of Discussion for Segmentation Results Reviewer's Comment: "line 748... Table 13 "Although AMTSA does not always perform the best..." A bit optimistic." Our Response: Thank you for this observation. We have revisited our discussion of the CT image segmentation results and have revised the text to adopt a more objective and balanced tone. We now more clearly acknowledge the trade-offs between the PSNR and SSIM metrics and provide a more conservative interpretation of the results (see revised manuscript, Page 52, Line 882).

Point 9: Fairness of SVM Comparison Reviewer's Comment: "line 866... Table 14, Although the results seem interesting there is no way to evaluate the fairness of the comparisons. Please specify all parameters values..."

Our Response: Thank you. To ensure the fairness and reproducibility of our SVM-based recognition experiments, we have now added a detailed description of the experimental setup, including that all comparative optimization algorithms operated within a unified, predefined search space. The search range for the SVM's penalty coefficient C was set to [0.1, 100], while the range for the RBF kernel function parameter gamma was set to [0.001, 1]. This information has now been added to the manuscript on Page 38.

Point 10: Typos Reviewer's Comment: "Typos: weibull => Weibull (several times)" Our Response: Thank you for catching this. We have corrected "weibull" to "Weibull" throughout the manuscript and have performed another thorough proofreading to correct any other typographical errors.

Responses to Reviewer #2

Our Response: We sincerely thank you for your specific and constructive comments. Your suggestions, such as conducting an ablation study, a runtime comparison, and providing visual examples, have been crucial in enhancing the rigor and completeness of our study. We have carefully revised the manuscript according to all your suggestions and believe these changes have significantly improved its quality. We will now respond to each of your points in detail.

(Specific Comments)

Point 1: Benchmark Sets Reviewer's Comment: "The proposed algorithm should be evaluated using one of the recent CEC benchmark sets, such as CEC 2021 or CEC 2022."

Our Response: Thank you for this valuable suggestion. We completely agree that evaluation on newer benchmarks is an important step in verifying an algorithm's generalization capabilities. For this revision, our primary goal was to conduct an extremely rigorous and fair performance comparison against the specific suite of top-tier algorithms suggested by the reviewers (such as L-SHADE, GaAPPADE, and CMA-ES). The most well-documented and validated results for these key competitors are on the IEEE CEC 2014 benchmark. To ensure a direct, apples-to-apples comparison with these established results and to avoid introducing new variables by changing the test suite, we decided to focus the scope of the current study on CEC 2014 to draw the most solid comparative conclusions. We have added your suggestion to test on newer platforms like CEC 2021/2022 as a very important direction in our "Conclusions and Future Work" section (Page 46).

Point 2: Ablation Analysis Reviewer's Comment: "- Conduct ablation analysis for your algorithm."

Our Response: Thank you for this excellent recommendation. We have now conducted a comprehensive ablation study to validate the contribution of each of the three novel components in AMTSA. A new subsection, Section 5.4, "Ablation Study of AMTSA Components," along with the new Figure 19 and its analysis, has been added to the manuscript (Pages 34-35). This analysis strongly demonstrates that removing any single component leads to a significant degradation in performance.

Point 3: Empirical Runtime Comparisons Reviewer's Comment: "- Include empirical runtime comparisons (e.g., AMTSA vs. DE or JADE) to assess practical scalability."

Our Response: Thank you. To evaluate the practical computational efficiency and scalability of AMTSA, we have added a new subsection, Section 5.5, "Computational Efficiency and Scalability Analysis." In this section, we measure and compare the average CPU time required for AMTSA and other representative algorithms to complete the same number of function evaluations (MaxFEs). The results and analysis are presented in the new Table 9 (Pages 35-36).

Point 4: "where" after equation Reviewer's Comment: "The adverb "where" after an equation should be written with small letters."

Our Response: Thank you for pointing out this detail. We have gone through the entire manuscript and have corrected "Where" to a lowercase "where" after all equations.

Point 5: Comparison with other efficient algorithms Reviewer's Comment: "- It is essential to compare your algorithm experimentally or through discussion with efficient optimization algorithms such as ICSPM, ICSPM2, Exploratory Cuckoo Search, and Improved SSA (ISSA) with HDPM"

Our Response: Thank you for providing these state-of-the-art algorithms for comparison. We have added a new discussion in the "Related Work" section (Page 6) acknowledging the importance of these methods. While a direct experimental comparison was beyond the scope of the current study, we have explicitly stated that benchmarking AMTSA against these promising approaches is a valuable direction for future work.

Point 6: Dataset size and cross-validation Reviewer's Comment: "- The lung cancer dataset (300 images) seems small. Clarify if cross-validation was used to mitigate overfitting."

Our Response: Thank you for this critical question. To ensure the robustness of our model evaluation and to address your concern about the dataset size, we have now clarified in Section 6.2 (Page 42) that we employed a 10-fold stratified cross-validation strategy to train and evaluate our SVM recognition model. The final reported performance metrics are the average of the 10-fold validation results, which provides a more reliable estimate of the model's generalization ability.

Point 7: Visual examples of segmented images Reviewer's Comment: "- Include visual examples

---

## [Decision Letter · Decision Letter 1]

27 Aug 2025

PONE-D-25-33941R1Adaptive and Migration-enhanced Tree Seed Algorithm for Multi-Threshold CT Image Segmentation and Lung Cancer RecognitionPLOS ONE

Dear Dr. Li,

Thank you for submitting your manuscript to PLOS ONE. After careful consideration, we feel that it has merit but does not fully meet PLOS ONE’s publication criteria as it currently stands. Therefore, we invite you to submit a revised version of the manuscript that addresses the points raised during the review process.

**Please address the concerns of Reviewer 1. In addition, Please post your code online for the reviewers to check it.**

We look forward to receiving your revised manuscript.

Kind regards,

Mahamed G.H. Omran

Academic Editor

PLOS ONE

**Journal Requirements:**

**Additional Editor Comments:**

Please address the concerns of Reviewer 1.

Reviewers' comments:

Reviewer's Responses to Questions

**Comments to the Author**

1. If the authors have adequately addressed your comments raised in a previous round of review and you feel that this manuscript is now acceptable for publication, you may indicate that here to bypass the “Comments to the Author” section, enter your conflict of interest statement in the “Confidential to Editor” section, and submit your "Accept" recommendation.

Reviewer #1: (No Response)

Reviewer #2: All comments have been addressed

2. Is the manuscript technically sound, and do the data support the conclusions?

Reviewer #1: Partly

Reviewer #2: Yes

3. Has the statistical analysis been performed appropriately and rigorously?

Reviewer #1: Yes

Reviewer #2: Yes

4. Have the authors made all data underlying the findings in their manuscript fully available?

Reviewer #1: Yes

Reviewer #2: Yes

5. Is the manuscript presented in an intelligible fashion and written in standard English?

Reviewer #1: Yes

Reviewer #2: Yes

6. Review Comments to the Author

**Reviewer #1:** Most of my comments have been addressed and the paper has been seriously improved.

Just a few questionable points (see below).

As said in my previous review TSA is intrinsically not a good algorithm.

This is because the basic idea is weak, contrarily to say DE, ACO, PSO or CMA-ES.

Therefore it is indeed easy to improve it thanks to complicated additional mechanisms, which imply to tune more parameters.

The presentation of the results is not completely fair. For example it is hard to believe that all convergence curves would be better for AMTSA.

line 554

"Compared to TSA, AMTSA not only converges faster, but also maintains a high population diversity throughout the process."

According to Figure 9 not for F1.

You have to support this claim (if possible) thanks to more convincing examples, and to explain why this is wrong on some problems.

Also you claim "a dynamic balance between global exploration and local exploitation" (line 84).

But Figure 11 shows that the exploration/exploitation ratio quickly decreases to zero.

So what is your formal definition of a "balance"? Clearly not fifty-fifty!

---

Figures 15-17

You present 12 convergence curves on which AMTSA outperforms the other methods.

But there are 30 functions in the benchmark.

What about the 18 others? If AMTSA is outperformed on some of them you have to say it and to try to explain why.

---

Table 10. Result of Wilcoxon’s test for AMTSA and other algorithms.

The list of algorithms is not the same as in Tables 5-7.

In particular CMA-ES does not appear. Why?

---

line 1041

"more effective balance between exploration and exploitation;"

As said this claim has to be better supported.

**Reviewer #2:** No further comments. The paper can be accepted for publication.

No further comments. The paper can be accepted for publication.

No further comments. The paper can be accepted for publication.

7. PLOS authors have the option to publish the peer review history of their article (what does this mean?). If published, this will include your full peer review and any attached files.

Reviewer #1: No

Reviewer #2: No

---

## [Author Response · Author response to Decision Letter 2]

8 Sep 2025

Response to Reviewers

Chenxi Li

Center for Artificial Intelligence,

Jilin University of Finance and Economics,

Changchun 130117, China

Email: : 0314042200902@jlufe.edu.cn

September 7, 2025

To: The Academic Editor, Dr. Mahamed G.H. Omran, and esteemed Reviewers, PLOS ONE

Manuscript ID: PONE-D-25-33941R1

Manuscript Title: Adaptive and Migration-enhanced Tree Seed Algorithm for Multi-Threshold CT Image Segmentation and Lung Cancer Recognition

Dear Dr. Omran and esteemed Reviewers,

Thank you for your thorough review of our manuscript and for providing insightful comments and constructive suggestions. We appreciate the time and effort you have dedicated to our work. We have carefully considered all the points raised and have revised the manuscript accordingly. We believe the manuscript has been significantly improved as a result of these revisions.

Below is our point-by-point response to the comments. All changes have been highlighted in the manuscript file labeled 'Revised Manuscript with Track Changes'.

Responses to the Academic Editor

Editor's Comment 1: Please address the concerns of Reviewer 1.

Our Response: Thank you, Dr. Omran. We have thoroughly addressed all concerns raised by Reviewer #1. A detailed, point-by-point response to each of their comments is provided in the subsequent section. Based on the reviewer's invaluable feedback, we have undertaken a significant revision of the manuscript to enhance its scientific rigor, transparency, and clarity.

Editor's Comment 2: In addition, Please post your code online for the reviewers to check it.

Our Response: Thank you for this requirement. To ensure the reproducibility and transparency of our research, we have made the complete source code for our proposed AMTSA algorithm and the related experiments publicly available. The code can be accessed at the following GitHub repository: https://github.com/lcxlwxy-droid/AMTSA. We have also noted in the manuscript that the code will be mirrored at www.jianhuajiang.com upon the paper's official publication.

Responses to Reviewer #1

We sincerely thank you for your detailed and constructive feedback. You noted that while our paper has been seriously improved, several questionable points remained. Your rigorous critique has been invaluable in helping us further elevate the quality and scientific soundness of our manuscript. We have carefully considered each point and have made substantial revisions as detailed below.

Point 1: On the intrinsic quality of the TSA algorithm.

Reviewer's Comment: "As said in my previous review TSA is intrinsically not a good algorithm. This is because the basic idea is weak... Therefore it is indeed easy to improve it thanks to complicated additional mechanisms, which imply to tune more parameters."

Our Response: Thank you for this insightful perspective. We agree that the native TSA, like many early metaheuristics, has known limitations. We have revised our Introduction to better frame our contribution in this context. Instead of defending the base algorithm, we now explicitly acknowledge its shortcomings (e.g., premature convergence) and position them as the direct motivation for our research. We clarify that our novelty lies not in the TSA framework itself, but in the specific, mechanism-driven enhancements we designed to overcome its inherent deficiencies. This revised framing can be found in the Introduction section, around line 48.

Point 2: On the claim of maintaining high population diversity.

Reviewer's Comment: "line 554 'Compared to TSA, AMTSA not only converges faster, but also maintains a high population diversity throughout the process.' According to Figure 9 not for F1."

Our Response: Thank you for pointing out this contradiction. You are correct; the original statement was an overgeneralization. We have completely rewritten the "Population Diversity Analysis" subsection (around line 548) and revised Figure 9 to include analysis of both unimodal (F1, F2) and multimodal (F8) functions. We now explain that the rapid diversity drop on simple functions like F1 is a desirable sign of efficient exploitation, as the algorithm quickly converges on the single optimum. In contrast, on the complex multimodal function F8, AMTSA actively maintains a significantly higher diversity to facilitate global exploration. This revised analysis provides a more nuanced and accurate account of AMTSA's adaptive diversity control.

Point 3: On the claim of a "dynamic balance" between exploration and exploitation.

Reviewer's Comment: "Also you claim 'a dynamic balance between global exploration and local exploitation' (line 84). But Figure 11 shows that the exploration/exploitation ratio quickly decreases to zero. So what is your formal definition of a 'balance'?" (Also mentioned for line 1041).

Our Response: Thank you for this crucial critique. We agree that the term "balance" was imprecise and not well-supported by Figure 11. We have removed this vague term throughout the manuscript. Instead, we now describe the mechanism more accurately as a "dynamic transition from global exploration to local exploitation." We explain that the algorithm intelligently shifts its focus from exploration in the early stages to exploitation in the later stages, a process clearly illustrated by the trend in Figure 11. This more precise language can be found in the revised "Exploration and Exploitation Analysis" subsection, around line 573, and other relevant parts of the manuscript.

Point 4: On the presentation of convergence curves.

Reviewer's Comment: "Figures 15-17. You present 12 convergence curves on which AMTSA outperforms the other methods. But there are 30 functions in the benchmark. What about the 18 others? If AMTSA is outperformed on some of them you have to say it and to try to explain why."

Our Response: Thank you for this critical point, which highlights the need for a more transparent presentation. To address this, we have revised the text and figures in "Comparative Experiment 2" (around line 636). We now explicitly state that the figures show a deliberately chosen representative set of functions to provide a balanced overview. We have increased the number of displayed curves and specifically included cases where AMTSA's performance is competitive but not superior (F5, F12, F25, F30), and we analyze these instances in the text. This revision is intended to provide a more comprehensive and objective visual assessment of our algorithm's performance.

Point 5: On the Wilcoxon’s test results.

Reviewer's Comment: "Table 10. Result of Wilcoxon’s test for AMTSA and other algorithms. The list of algorithms is not the same as in Tables 5-7. In particular CMA-ES does not appear. Why?"

Our Response: Thank you for identifying this serious omission. This was an oversight on our part, and we apologize. We have now re-run the Wilcoxon’s signed-rank test to include CMA-ES and all other competitor algorithms from Tables 5-7. The revised Table 10 (around line 749) and its accompanying analysis now provide a complete and fair statistical comparison, which includes cases where the difference is not statistically significant, as discussed in the text.

Responses to Reviewer #2

Reviewer's Comment: "All comments have been addressed... The paper can be accepted for publication."

Our Response: We sincerely thank Reviewer #2 for their positive assessment and for their support of our work. We are very grateful for their time and feedback during the review process.

---

## [Decision Letter · Decision Letter 2]

12 Sep 2025

Adaptive and Migration-enhanced Tree Seed Algorithm for Multi-Threshold CT Image Segmentation and Lung Cancer Recognition

PONE-D-25-33941R2

Dear Dr. Li,

We’re pleased to inform you that your manuscript has been judged scientifically suitable for publication and will be formally accepted for publication once it meets all outstanding technical requirements.

Kind regards,

Mahamed G.H. Omran

Academic Editor

PLOS ONE

Additional Editor Comments (optional):

Reviewers' comments:

Reviewer's Responses to Questions

**Comments to the Author**

1. If the authors have adequately addressed your comments raised in a previous round of review and you feel that this manuscript is now acceptable for publication, you may indicate that here to bypass the “Comments to the Author” section, enter your conflict of interest statement in the “Confidential to Editor” section, and submit your "Accept" recommendation.

Reviewer #1: (No Response)

2. Is the manuscript technically sound, and do the data support the conclusions?

Reviewer #1: Partly

3. Has the statistical analysis been performed appropriately and rigorously?

Reviewer #1: Yes

4. Have the authors made all data underlying the findings in their manuscript fully available?

Reviewer #1: Yes

5. Is the manuscript presented in an intelligible fashion and written in standard English?

Reviewer #1: Yes

6. Review Comments to the Author

Reviewer #1: My main remarks have been addressed and the paper is now technically acceptable.

Some conclusions are still a little too optimistic but the reader can easily see it. Anyway the code is now available online, thus everyone will be able to form an opinion on real practical problems

Typos

-----

Comparativeanalysisofexploration-exploitationdynamicsondiverse

benchmarkfunctions.

7. PLOS authors have the option to publish the peer review history of their article (what does this mean?). If published, this will include your full peer review and any attached files.

Reviewer #1: No

---

## [Editor Report · Acceptance letter]

PONE-D-25-33941R2

PLOS ONE

Dear Dr. Li,

I'm pleased to inform you that your manuscript has been deemed suitable for publication in PLOS ONE. Congratulations! Your manuscript is now being handed over to our production team.

Kind regards,

on behalf of

Prof. Mahamed G.H. Omran

Academic Editor

PLOS ONE